# Improving soil moisture and runoff simulations at 3 km over Europe using land surface data-assimilation

Bibi S. Naz[1,2], Wolfgang Kurtz[7], Carsten Montzka[1], Wendy Sharples[2,3], Klaus Goergen[1,2], Jessica Keune[4], Huilin Gao[5], Anne Springer[6], Harrie-Jan Hendricks Franssen[1,2], Stefan Kollet[1,2]

[1]Research Centre Jülich, Institute of Bio- and Geosciences: Agrosphere (IBG-3), Jülich 52425, Germany
[2]Centre for High-Performance Scientific Computing in Terrestrial Systems, Geoverbund ABC/J, Jülich 52425, Germany
[3] Research Centre Jülich, Jülich Supercomputing Centre, Jülich 52425, Germany
[4]Laboratory of Hydrology and Water Management, Ghent University, Ghent 9000, Belgium
[5]Zachry Department of Civil Engineering, Texas A&M University, College Station, TX 77843, USA
[6]Institute of Geodesy and Geoinformation, Bonn University, Nussallee 17, Bonn 53115, Germany
[7]Leibniz Supercomputing Centre, Environmental Computing Group, Boltzmannstr. 1, 85748 Garching, Germany

*Correspondence to*: Bibi S. Naz (b.naz@fz-juelich.de)

**Abstract.** Accurate and reliable hydrologic simulations are important for many applications such as water resources management, future water availability projections and predictions of extreme events. However, the accuracy of water balance estimates is limited by the lack of large scale observations, model simulation uncertainties and biases related to errors in model structure and uncertain inputs (e.g. hydrologic parameters and atmospheric forcings). The availability of long-term and global remotely sensed soil moisture offers the opportunity to improve model estimates through data assimilation with complete spatio-temporal coverage. In this study, we assimilated the European Space Agency (ESA) Climate Change Initiative (CCI) derived soil moisture (SM) information to improve the estimation of continental-scale soil moisture and runoff. The assimilation experiment was conducted over a time period 2000-2006 with the Community Land Model, version 3.5 (CLM3.5) integrated with the Parallel Data Assimilation Framework (PDAF) at spatial resolution of 0.0275° (~3 km) over Europe. The model was forced with the high-resolution reanalysis COSMO-REA6 from the Hans-Ertel Centre for Weather Research (HErZ). The performance of assimilation was assessed against open-loop model simulations and cross-validated with independent ESA CCI derived soil moisture (CCI-SM) and gridded runoff observations. Our results showed improved estimates of soil moisture, particularly in the summer and autumn seasons when cross-validated with independent CCI-SM observations. The assimilation experiment results also showed overall improvements in runoff, although some regions were degraded, especially in central Europe. The results demonstrated the potential of assimilating satellite soil moisture observations to produce downscaled and improve high-resolution soil moisture and runoff simulations at the continental scale, which is useful for water resources assessment and monitoring.

# 1 Introduction

Soil moisture (SM) is a key variable of the hydrologic cycle playing an important role in major processes related to infiltration and runoff generation, root water uptake and plant transpiration, and evaporation (Vereecken et al., 2016). Thus, soil moisture strongly influences the partitioning of incoming radiative energy into latent and sensible heat and significantly affects the land surface energy and water budgets. Consequently, accurate estimates of large scale SM are needed for detection of long-term trends in hydrological states and fluxes, for example in the context of a land surface reanalysis; hydrologic predictions such as discharge forecasts for large river basins (Western et al., 2004) and water resource management and planning (e.g. groundwater recharge, mitigation of droughts) (Andreasen et al., 2013; Dobriyal et al., 2012; Sridhar et al., 2008); identifying regions susceptible to extreme events such as droughts and floods (Seneviratne et al., 2010); lower boundary condition for numerical weather predictions (Drusch, 2007); and irrigation management and agriculture practices (Bolten et al., 2010; Shock et al., 1998). At continental space and inter-annual time scales, SM typically exhibits large variability (Brocca et al., 2010), depending on rainfall distribution, topography, soil physical properties, vegetation characteristics, and human impacts, such as irrigation. Monitoring this variability is a major challenge due to the scarcity of *in situ* SM observations networks. Recent advancements in satellite-based sensors offer great potential to monitor SM over large scales for continental water resources assessment, particularly in areas where ground observation networks are sparse (Mohanty et al., 2013). Conventionally, satellite observations have been used in global water balance studies to provide information on the water cycle components, such as precipitation, evapotranspiration (ET), soil moisture, water storage and runoff (Kiehl and Trenberth, 1997; Running et al., 2004; Trenberth et al., 2007; Vinukollu et al., 2011). However, sparse data coverage in satellite observations limits their ability to provide spatially and temporally consistent time series of water balance estimates.

Another approach to facilitate studies at a regional to global scale is to estimate water budget components using land surface models forced with precipitation and other atmospheric data (such as the Community Land Model (CLM) (Lawrence et al., 2011), the Variable Infiltration Capacity (VIC) model (Liang et al., 1994, 1996), or the Joint UK Land Environment Simulator (JULES) (Best et al., 2011; Clark et al., 2011). Simulated soil moisture distributions from the land surface models provide spatially and temporally continuous information, yet their accuracy is limited by model deficiencies, and uncertainties in both model parameters and atmospheric forcing variables (Chen et al., 2013; Draper et al., 2009). Therefore these uncertainties need to be characterized in hydrologic predictions, in order to provide useful hydrologic data and information for water resource management. In order to improve model predictions and simultaneously honor observation and model uncertainties, remotely sensed soil moisture has been merged with model predication using data assimilation (DA) (Chen et al., 2013; De Lannoy and Reichle, 2016; Kumar et al., 2008; Lahoz and De Lannoy, 2014; Lievens et al., 2016; Liu and Gupta, 2007; Moradkhani et al., 2005; Nie et al., 2011). Using DA approaches, previous studies also investigated the impact of uncertainties in both parameters and state variables of a hydrologic model based on joint state–parameter estimation approach (Cammalleri and Ciraolo, 2012; DeChant and Moradkhani, 2012; Gharamti et al., 2015; Liu

et al., 2016; Pathiraja et al., 2016; Rafieeinasab et al., 2014; Xie and Zhang, 2010). For example, Han et al. (2014) evaluated the joint state and parameter estimation method at catchment scale for the coupled CLM and Community Microwave Emission Model (CMEM) (De Rosnay et al., 2009) through assimilation of synthetic microwave brightness temperature data. Similarly, Samuel et al. (2014) studied the ensemble based DA with dual state-parameter estimation to evaluate the streamflow forecast and variations in soil moisture. However, many of these studies mainly focused on using data assimilation approaches for improved predictions at the watershed scales. Fewer studies demonstrated the potential of assimilating satellite observations into land surface models to improve soil moisture and runoff estimates at regional and global scales (e.g. Crow and Ryu, 2009; De Lannoy and Reichle, 2016; Lievens et al., 2015; Liu and Mishra, 2017; Lopez et al., 2016; Pan et al., 2008; Rains et al., 2017; Reichle and Koster, 2005; Renzullo et al., 2014). For instance, Rains et al. (2017) assimilated SMOS data into CLM over Australia for drought monitoring purposes. Similarly, Liu and Mishra, (2017) also assimilated satellite SM data at the global scale to evaluate the performance of the community land surface model (CLM4.5) in simulating hydrologic fluxes such as SM, ET and runoff at 0.5° (~50 km) spatial resolution. They found that assimilating satellite SM data into the CLM4.5 model improved the soil moisture simulations, which also led to better representation of other hydro-meteorological variables in the model, such as ET and runoff. At the continental scale, several studies have explored the role of soil moisture assimilation over Europe, in different modeling frameworks (e.g. Albergel et al., 2017; Brocca et al., 2010; De Rosnay et al., 2013; Draper et al., 2009; Ni-Meister et al., 2006). Using NASA's global Catchment Land Model (CLSM), Ni-Meister et al. (2006) improved simulated soil moisture over small Eurasia catchments through assimilation of near surface soil moisture derived from Scanning Multichannel Microwave Radiometer (SMMR). Using Extended Kalman Filter (EKF), Draper et al. (2009) demonstrated the usefulness of assimilating near-surface soil moisture observations from C-band Advanced Microwave Scanning Radiometer (AMSR-E) in the land surface model ISBA (Interactions between Soil, Biosphere and Atmosphere) at 9 km resolution over continental Europe. More recently, Albergel et al. (2017) showed the potential of using the satellite-derived soil moisture data from European Space Agency (ESA) Climate Change Initiative (CCI) over Europe and Mediterranean domain to sequentially assimilate soil moisture and leaf-area index product into the ISBA land surface model at 0.5°(~50 km) resolution. They found significant improvements in the surface soil moisture but little improvements of discharge estimates when compared to the open loop (i.e. no assimilation) simulations.

In these global to regional scale studies, despite these important advancements, the data assimilation systems were employed at fairly coarse spatial resolution (i.e. at 50 to 25 km scale), which is too coarse to provide locally relevant information (Bierkens et al., 2015; Wood et al., 2011). For example, predicting water cycle processes for scientific and applied assessment of the terrestrial water cycle requires a high-resolution modeling framework on the order of $10^0$-$10^1$ km horizontal grid spacing. While most of the global remote sensing observations are available at relatively low resolution (i.e. at 50 to 25 km scale), data assimilation systems can be used as an effective downscaling tool by merging the remote sensing information in space and time with high resolution models. In turn, data assimilation frameworks that include higher resolution meteorological, land cover, and soil texture information can be used to constrain coarse resolution remotely

sensed soil moisture observations. However, in both cases, the spatial mismatch between coarse-resolution satellite data and high-resolution hydrologic models constitutes a great challenge. To address this issue, the scale disparity between observations and modeling approaches needs to be taken into account either in the data assimilation algorithm (De Lannoy et al., 2012; Sahoo et al., 2013), or through pre-processing of satellite products to match the model resolution (Merlin et al.,

2013; Verhoest et al., 2015). Another challenge is the availability of computational resources, because the computational burden increases (non-) linearly with increasing model resolution, with the number of ensemble members in the data assimilation system and with the increasing complexity of simulated processes.

In this work, we assimilated the coarse resolution ESA CCI-SM data over Europe from January 2000 to December 2006 into the 3 km high-resolution CLM model using joint state and parameter estimation approach and evaluate its impacts both on

surface soil moisture and other hydrologic variables such as surface and subsurface runoff. A number of soil moisture retrievals from other missions such as the Soil Moisture and Ocean Salinity (SMOS, launched in 2009) (Kerr et al., 2001; Mecklenburg et al., 2016) and SMAP mission (Soil Moisture Active Passive, launched in 2015) (Entekhabi et al., 2010) have been used in assimilation studies (e.g. Lievens et al., 2015, 2016). These recent high-resolution data products are only available for less than 10 years and cannot be used to apply soil moisture information in a land surface reanalysis for

extended time periods. Recently, a number of studies highlighted the quality and stability of ESA-CCI product (e.g. Dorigo et al., 2017; McNally et al., 2016; Wagner et al., 2012) and its potential use in data assimilation studies (Albergel et al., 2013, 2017; Liu et al., 2018). We selected ESA CCI-SM data because of its availability at longer time scales, which also provides a basis for evaluating the feasibility to derive a land surface reanalysis, conditioned to satellite information, for Europe over longer time scales, and allows to assess the potential impact of assimilating longer-term soil moisture

observations on hydrologic simulations.

The main goal of this study is twofold. Firstly, it investigates the value of coarse-resolution remotely sensed soil moisture data in improving soil moisture and runoff modeling and to provide higher spatial resolution, downscaled estimates of the surface soil moisture profile and hydrological fluxes with complete spatio-temporal coverage over Europe. Secondly, it aims at exploring the potential of long-term remotely sensed products such as ESA-CCI SM for downscaling of soil moisture to

high spatial resolution at the continental scale via data assimilation. In this study, the analysis also focused on the performance of the Community Land Model (v3.5) to simulate surface and subsurface runoff as result of assimilation updates restricted to soil moisture for upper soil layers. In addition, the study also interrogates whether assimilation of satellite-derived surface soil moisture will improve the skill of the simulated discharge, in gauged and ungauged regions. Therefore, assimilating satellite-derived information into land surface models may have an important added value for regions

where in situ measurements are not available.

In order to obtain the assimilated product of near surface soil moisture, we used CLM3.5 coupled to the Parallel Data Assimilation Framework library (PDAF) (Nerger and Hiller, 2013). PDAF is computationally efficient due to its parallelization of data assimilation routines and in-memory exchange of data. Therefore, PDAF is suitable for applications at large spatial scales and high-resolution over longer time periods. The coupled CLM-PDAF setup and the experimental

design are described in Sec. 2. The results, including model validation and analysis of simulated soil moisture and runoff are documented in Sec. 3; while the discussion and conclusions are presented in Sec. 4 and Sec. 5, respectively.

## 2 Methods and Data

### 2.1 Model Description

In this study, the Community Land Model version CLM3.5 (Oleson et al., 2004) was applied to represent land surface processes such as soil moisture evolution, evaporation from soil and vegetation, transpiration and interception of precipitation by vegetation canopy, throughfall and infiltration, surface and subsurface runoff and snow. Specifically, runoff is parameterized using a simple TOPMODEL-based scheme (SIMTOP; Niu et al., 2005). Soil water is calculated by solving the one-dimensional Richards equation (Zeng and Decker, 2009). An operation groundwater table depth and recharge to

groundwater from the soil column is updated dynamically using the algorithm described in Niu et al. (2007). The snow model in CLM explicitly simulates multilayer snow depending on the total snow depth, and includes processes such as snow-melting, surface frost and sublimation, liquid water retention and thawing-freezing processes (Dai et al., 2003; Dickinson et al., 2006; Stöckli et al., 2008). Total runoff is calculated as the sum of the subsurface runoff, surface runoff and runoff generated from lakes, glaciers, and wetlands (Oleson et al., 2004).

CLM3.5 has been widely applied at continental and global scales to understand how land processes and anthropogenic impacts affect climate (e.g. Dickinson et al., 2006). The CLM model parameterizes most of the land surface processes (such as infiltration, evaporation, surface runoff, subsurface drainage, canopy and snow processes) using the water and energy balance equations. CLM3.5 offers significant improvements in estimating the components of the terrestrial water cycle compared to earlier versions (Oleson et al., 2008), including improvements in soil water availability and resistance terms to

reduce the soil evaporation which was overestimated in earlier versions (Niu et al., 2005; Oleson et al., 2008;Yang and Niu, 2003). Compared to CLM3.0, Oleson et al.,(2008) showed that CLM3.5 exhibits more realistic partitioning of ET into its components (i.e. transpiration, ground evaporation, and canopy evaporation), which resulted in overall improvements in the representation of the annual cycle of total water storage. Previous studies also showed that CLM3.5 produces too high soil moisture with too low variability compared to root zone soil moisture modelled by later CLM versions (4.0 and 4.5) (e.g.

Lawrence et al., 2011 and Niu et al., 2011). In order to reduce these biases, Li and Ma, (2015) introduced a factor to describe soil porosity and increase recharge water from the soil column to the aquifer in the newer CLM leading to improved estimates of soil moisture and biogeochemical processes. However, Lawrence et al. (2011) showed that the differences between CLM3.5 and new versions of CLM with respect to soil moisture variability remained small when compared to observations.

In addition, CLM3.5 is designed for coupling with climate models and is also part of the fully coupled Terrestrial Systems Model Platform (TerrSysMP; Shrestha et al., 2014) that simulates the full terrestrial hydrologic cycle including feedbacks between atmosphere, land-surface and subsurface compartments of the water cycle. Moreover, the CLM model can

efficiently run for large model domains and at high spatial resolution. Since, we performed our simulations at high spatial resolution and at continental scale, we selected the TerrSysMP-PDAF modelling framework (Kurtz et al., 2016) which is design for high performance computing infrastructures and can efficiently cope with the high computational burden of ensemble-based data assimilation. Kurtz et al. (2016) showed the efficient use of parallel computational resources by

TerrSysMP-PDAF, which is needed to simulate predicted states and fluxes over large spatial domains and long simulations. In this study, we used the CLM-PDAF setup, in which PDAF is coupled with the stand-alone CLM3.5 for soil moisture assimilation. Readers are referred to Kurtz et al. (2016) for technical descriptions of coupling and model performance.

## 2.2 Data assimilation framework

The Parallel Data Assimilation Framework (PDAF) (Nerger and Hiller, 2013) was used to assimilate satellite soil moisture

into CLM3.5. PDAF provides data assimilation methods such as the EnKF (Burgers et al., 1998; Evensen, 2003) and the local ensemble transform Kalman filter (LETKF) (Hunt et al., 2007). In this study, the EnKF-algorithm was used for data assimilation, which is a relatively efficient and robust technique for assimilating satellite data into land surface models (e.g. Brocca et al., 2012; Crow et al., 2017; Draper et al., 2011; Matgen et al., 2012; Mohanty et al., 2013; Pauwels et al., 2001, 2002). It uses ensembles of model simulations to approximate the model state and parameter error covariance matrix in order

to optimally merge model predictions with observations.

In this study, the joint state parameter update of soil moisture and soil texture in CLM with the EnKF was used:

$$\theta_t^i = f_t\left(\theta_{t-1}^i, q_{t-1}^i, p_{t-1}^i\right) \quad (1)$$

where the state variable soil moisture $\left(\theta_t^i\right)$ is a vector containing soil moisture values within a soil layer for each grid cell and can be described with a non linear model (CLM in our case) operator $f_t(\cdot)$ at time step $t$ for realization $i$ using the forcing data $q$ and model parameter $p$.

The state-parameter vector $x^i$ for realization $i$ was calculated using the perturbed soil texture (% sand and % clay) and perturbed precipitation as follows:

$$x^i = \begin{pmatrix} \theta^i \\ \%Sand^i \\ \%Clay^i \end{pmatrix} \quad (2)$$

The EnKF then calculates the ensemble of updated states-parameter vector $x_t^a$ at daily timestep $t$ of the model estimated state-parameter variable $x_t$ for each ensemble member $i$, as:

$$x_t^a = x_t^i + \mathbf{K}_t[y + \varepsilon_i - \mathbf{H}_t x_i^t] \quad (3)$$

where $y_t$ is the perturbed observation vector and $\epsilon$ is a perturbation vector of the measurement error with values drawn from

a normal distribution with a mean of zero and a standard deviation corresponding to the assigned measurement error of $0.02 \; m^3 m^{-3}$ for soil moisture and $\mathbf{H}$ is the measurement operator. $\mathbf{K}$ is the Kalman gain matrix defined as:

$$\mathbf{K}_t = \mathbf{P}_t \mathbf{H}_t^T (\mathbf{R}_t + \mathbf{H}_t \mathbf{P}_t \mathbf{H}_t^T)^{-1} \quad (4)$$

where $\mathbf{H}_t^T$ is the transpose matrix of the measurement operator at time $t$, $\mathbf{R}_t$ is the measurement error matrix, which is defined a priori based on the expected measurement error of the ESA CCI soil moisture product. $\mathbf{P}_t$ is the state error covariance matrix of the model predictions calculated as:

$$\mathbf{P_t} = \frac{\sum_{i=1}^{N}(x_i-\bar{x})(x_i-\bar{x})^{\mathrm{T}}}{N-1} \quad (5)$$

where $\bar{x}$ is the vector which contains the ensemble average soil moisture contents for the different grid cells and $N$ is the number of ensemble members.

In the DA scheme, the updated states (soil moisture) were kept in reasonable physical ranges (between residual soil moisture and porosity) to yield physically meaningful estimates of soil moisture water content, energy fluxes and water budget. For

the soil moisture update, the values of the updated soil moisture were restricted to values between zero and saturated soil water content. For the soil texture update, a value of 1% was assigned to sand and clay percentages in case the updated values are less than zero. In case the updated sum of the sand and clay are greater than 100%, the values were constrained to the normalized sum of updated soil and clay percentages. Other soil parameters such as the soil hydraulic and thermal parameters were adjusted after the soil texture update using pedo-transfer functions.

**2.3  Data**

**2.3.1 Land surface data and atmospheric forcing**

The land surface static input data used in this study consisted of topography, soil properties, plant functional types, and physiological vegetation parameters (Fig. 1). Digital elevation model (DEM) data were acquired from the 1 km x 1 km Global Multi-resolution Terrain Elevation Data 2010 (GMTED2010) (Danielson and Gesch, 2011) as shown in Fig. 1a. In

CLM, each grid cell consists of five landunits (i.e. vegetation, wetland, lakes, glaciers and urban) covering a certain percentage of the total grid cell area. The vegetation landunit is further subdivided into Plant Functional Types (PFTs) defined by fractional areas with respect to the entire grid cell (Bonan et al., 2002). In the current study, the land cover information for each PFT was based on the Moderate Resolution Imaging Spectroradiometer (MODIS) MCD12Q1 (version 5) land cover product (Friedl et al., 2002), which contains a classification of the dominant land cover (Figure 1b). The

dominant land cover information from MODIS was first aggregated to the model resolution, calculating the percentage of all 500 m pixels per 3 km grid cell. The aggregated land cover information was then transferred to the CLM-prescribed PFTs on the basis of WorldClim climate data (Hijmans et al., 2005).

Monthly leaf area index (LAI) values for each PFT were computed based on the 1 km Global Land Surface Satellite (GLASS) Leaf Area Index (LAI) product (1981-2012). GLASS contains of 1 km x 1 km global maps of LAI provided every

8 days. The product was derived from time-series of MODIS (MOD09A1) and AVHRR reflectance data using general regression neural network method (Xiao et al., 2014). To derive a monthly climatology over the assimilation period (2000 – 2006), the 1 km 8-day improved GLASS LAI for each year was used to calculate a mean monthly LAI that was then

aggregated to the model resolution. The monthly LAI values for each PFT were then determined by mapping the 3 km pixels to the 3 km aggregated PFT values within each grid cell. This approach provides spatially distributed and temporally continuous LAI data within each PFT for the considered time period of 2000-2006. To account for annual variability in LAI, yearly model runs were performed where the LAI information was updated at the start of each year run. It should be noted that CLM3.5 only allows to specify monthly average LAI values for each PFT.

Additional properties of each of the sub-grid land fractions, such as the stem area index, and the monthly heights of each PFT, were calculated based on the global CLM3.5 surface data set (Oleson et al., 2008). To provide soil texture data in the model (Fig. 1c and 1d), sand and clay percentages were prescribed based on pedotransfer functions (Schaap and Leij (1998) for 19 soil classes derived from the FAO/UNESCO Digital Soil Map of the World (Batjes, 1997).

[**Figure 1**]

For the time period of 2000-2006, the high-resolution atmospheric reanalysis COSMO-REA6 dataset (Bollmeyer et al., 2015) from the Hans-Ertel Centre for Weather Research (HErZ; Simmer et al., 2016) was used as the atmospheric forcing for CLM3.5. We preferred to use this data over other datasets, because of its high spatial resolution in comparison to other commonly used forcing datasets such as the European gridded data set (E-OBS) (Haylock et al., 2008) and Interim ECMWF Reanalysis (ERA-Interim; Dee et al., 2011) at 25 and 80 km resolution, respectively. We used data from 2000-2006 which were available at the beginning of this study. The COSMO-REA6 is only now publicly available for a longer time period of 1995-2015. The essential meteorological variables applied in this study, such as barometric pressure, precipitation, wind speed, specific humidity, near surface air temperature, downward shortwave radiation and downward longwave radiation were downloaded from the German Weather Service (DWD; ftp://ftp-cdc.dwd.de/pub/REA/). The COSMO-REA6 reanalysis is based on the COSMO model and available at 0.055° (~6 km) covering the European CORDEX domain (Gutowski Jr et al., 2016). COSMO-REA6 was produced through the assimilation of observational meteorological data using the existing nudging scheme in COSMO with boundary conditions from ERA-Interim data. Bollmeyer et al. (2015) compared the COSMO-REA6 precipitation data with the precipitation data from Global Precipitation Climatology Centre and showed that COSMO-REA6 performed well compared to observations with small underestimations of precipitation in mid and southern Europe and overestimation of precipitation in Scandinavia, Russia and along the Norwegian coast. Additionally, Springer et al. (2017) assessed the closure of the water budget in the 6 km COSMO-REA6 and compared to global reanalyses (ERA-Interim, Modern-Era Retrospective Analysis for Research and Applications, Version 2 (MERRA-2) for major European river basins. Springer et al. (2017) found that the COSMO-REA6 closes the water budget within the error estimates whereas the global reanalyses underestimate the precipitation minus evapotranspiration surplus in most river basins. A more comprehensive assessment of the precipitation of the HErZ reanalysis can be found in Wahl et al. (2017) albeit based on the 2 km data product, only available for central Europe.

### 2.3.2 ESA CCI microwave soil moisture

The European Space Agency (ESA) Climate Change Initiative (CCI) program provides daily soil moisture (CCI-SM) at 0.25° spatial resolution for approximately the top few millimeters to centimeters of soil from 1978 to 2016. The daily CCI-SM product (v03.2) is produced at 0.25° spatial resolution from the microwave retrieved surface soil moisture data and is

merged from multiple sensors (Dorigo et al., 2017; Liu et al., 2011, 2012; Wagner et al., 2012; http://www.esa-oilmoisture-cci.org). For the study period of 2000 to 2006, the CCI-SM data are based on passive microwave observations (i.e. DMSP SSM/I, TRMM TMI, Aqua AMSR-E and Coriolis WindSat; Owe et al., 2008), whereas the active data products are based on observations from the C-band scatterometers on board of the ERS-1 and ERS-2 (Bartalis et al., 2007; Wagner et al., 2013) satellites. In this product, the absolute soil moisture was re-scaled against the 0.25° land surface modeling soil moisture

(GLDAS-NOAH, Rodell et al., 2004) using cumulative density function matching. The soil type–specific hydraulic parameters in Noah are obtained from the pedotransfer function (PTF) provided in Cosby et al. (1984), which was also adopted by CLM. The underlying soil classification in our setup is based on data from FAO soil map (Batjes, 1997), which was the basis for the GLDAS derived soil parameters used in GLDAS-Noah and employed to derive the ESA CCI-SM product (e.g. Dorigo et al., 2012). The setup and the parameterizations of Noah and our CLM model should hence be fairly

consistent.

 In this study, we used the merged product of active and passive soil moisture data which showed better accuracy than either of the passive or active data alone (Liu et al., 2011). To match the spatial resolution of our CLM3.5 setup, the original SM values were re-sampled and re-gridded to 0.0275° using the first-order conservative interpolation method (Jones, 1999) which is based on the ratio of source cell area overlapped with the corresponding destination cell area. The conservative

regridding scheme preserves the physical flux fields between the source and destination grid. The CCI-SM dataset showed large data gaps over the European continent during the four seasons (December-February (DJF; Winter), March-May (MAM; Spring), June-August (JJA; Summer), and September-November (SON; Autumn); Fig. 2b). According to Fig. 2b, the temporal coverage (i.e. the ratio between the number of days and the total number of days in a season) was generally low during the winter and spring seasons, ranging from less than 30% (Scandinavian regions) to about 60% in southern Europe.

SM observations showed the highest temporal coverage during the summer and autumn. Due to the sparseness of the SM data at daily temporal resolution, 100 grid cells were randomly selected covering the complete model domain (Fig. 2a). The satellite CCI-SM daily soil moisture data at these locations were assimilated. However, the number of observations for each day ranged between 2 and 75 depending on the availability of the daily CCI-SM data. As shown in Fig. 2c, there is a higher level of noise in the CCI-SM data for the first two years (2000 and 2001) probably related to the absence of data from other

sensors like AMSRE-E and Windsat in those first two years. Moreover, availability of selected observations was lower during winter and spring, while summer soil moisture was well covered during years 2003 to 2006. This seasonal difference in data availability is related to the occurrence of soil freezing events and snow cover.

Furthermore, in land surface modelling systematic differences between the model climatology and the observation data climatology are commonly corrected before assimilation, to ensure that data assimilation is applied under conditions of no systematic bias. Previous studies used different procedures to correct for biases, such as the estimation of a single constant bias value, seasonal dependent bias or CDF-matching (e.g. (Drusch et al., 2005; Reichle and Koster, 2004). The procedure

has some important limitations: (i) the polynomial fit during CDF matching cannot provide perfect agreement because the introduced noise changes the random difference between both data sets, (ii) the bias is only partially corrected or over-corrected; (iii) the bias in the DA-procedure is not assigned to the model or measurement data, but after the assimilation it is implicitly assumed that the systematic bias is related to the bias in the measurements (model states are not corrected for a systematic bias). A priori bias correction is a specific approach taken in land surface data assimilation in case of large

mismatches between modeled and measured values, for example, when the observations are located outside of the ensemble spread. We argue that for this dataset, we see systematic biases between model and data, yet these are small enough. In addition, data assimilation is able to remove biases besides the random component. A further argument for not following this approach was that spatial patterns could be altered and thereby some of the independent information provided by the satellite may be removed. We, therefore, did not perform any bias correction of the ESA CCI-SM data by rescaling of the

observations to model climatology to retain as much of the independent satellite information as possible.

[**Figure 2**]

### 2.3.3 Observational gridded monthly runoff

In order to evaluate the potential of improving runoff estimates by assimilating soil moisture observations, the non-routed observational gridded monthly runoff data from Gudmundsson and Seneviratne (2016) (E-RUN version 1.1) were used as

independent dataset. The E-RUN product provides monthly pan-European runoff estimates from 1950 to 2015 at 0.5°(~50 km) resolution. The monthly runoff rates were generated using a collection of streamflow observations from small catchments combined with gridded precipitation and temperature data using a machine learning approach (Gudmundsson and Seneviratne, 2016). Monthly runoff was estimated using a regression model, which was trained with a subset of observed runoff rates and E-OBS precipitation and temperature. The fitted model was subsequently applied to all grid cells

of the E-OBS data to derive pan-European estimates of monthly runoff (Gudmundsson and Seneviratne, 2016). Using this cross-validation method, Gudmundsson and Seneviratne (2016) reported higher accuracy in central and western Europe, while accuracy was lower in other regions due to low density of available stations. For model validation, we preferred to use this dataset over the discharge observations at different gauge stations, because the non-routed gridded runoff product has distinct advantage to evaluate the impact of soil moisture assimilation on runoff at every grid cell within a spatial domain.

Using gridded runoff is also useful to evaluate model structure errors in representation of runoff generation in the model. In addition, in the CLM3.5, the river routing module is implemented at 0.5° where the discretization of river routing elements is based on a grid method in which the grid for river routing is independent of the grid for runoff simulation. Therefore, a coarse spatial resolution river network can lead to unrealistic flow accumulation and an adequate validation of the results is

not possible. However, our comparison of aggregated runoff using E-RUN data for few watersheds with monthly discharge observed at station and obtained from Global Runoff Data Center (GRDC; Global Runoff Data Center, 2011) in Europe showed a good agreement with observed discharge (Fig. S1). In the current study, the half degree monthly runoff rates were resampled and re-gridded to 0.0275° using the first-order conservative interpolation method for comparison with the CLM3.5 simulated total runoff.

## 2.4 CLM-PDAF experimental design and analyses

The joint state and parameter assimilation experiments were performed for the time period of January 2000 to December 2006. The model spinup was performed by simulating the time period of 2000-2006 five times in order to obtain equilibrium initial state variables. The initial state variables from the spin-up were then used as initial condition for the ensemble runs as described below. In this study, we implemented CLM3.5 for the EURO-CORDEX domain with a spatial resolution of 0.0275° (3 km), inscribed into the official EUR-11 grid. The model was run with 1h time step and the time window for soil moisture updates was set to 1 day. In this study, we assumed a spatially uniform observational error of 0.02 $mm^3/mm^3$ for CCI-SM in the CLM-PDAF setup.

The outputs of a land surface model are sensitive to both atmospheric forcings and soil characteristics. To account for uncertainties in atmospheric forcing and soil texture, precipitation and soil texture (%sand and %clay) were perturbed in this study (Fig. S2 and Fig. S3). Log-normally distributed, spatially homogeneous and temporally uncorrelated multiplicative perturbations were added to precipitation. The mean and standard deviation of the applied perturbation factors for precipitation were equal to 0.1 and 0.15, respectively. Sand and clay content were perturbed with random noise drawn from spatially uniform distribution (±10%). In order to avoid unphysical values of the soil parameters, the sum of the sand and clay content were constrained to have a value not larger than 100%. The initial ensemble size was set to 20 for the precipitation and soil texture in the simulation/assimilation experiment to update the volumetric soil water content (SWC) of the top soil layer (~ 2cm). Previous studies (e.g. De Lannoy et al., 2012; Kumar et al., 2008; Pan and Wood, 2010; Yin et al., 2015) showed that the performance of EnKF relies on the ensemble size. For example, Yin et al. (2015) indicated that when the ensemble size is close to 12, it may lead to efficient DA updating process, while Pan and Wood, (2010) suggested 20 ensemble members. Our initial investigation showed slightly improved correlation ($R^2$) between simulated and CCI-SM soil moisture for 20 ensemble members compared to 12 ensemble members (as shown in Fig. S4). In addition to ensemble size, systematic biases can also be attributed to erroneous model parameter values, which is one of the main sources of error and uncertainty in land surface model predictions. To account for biases in soil parameters, the joint state and parameter assimilation framework was used to estimate the model states and model parameters jointly by updating the soil water content and soil texture properties such as %sand and %clay. Although this approach has also significant limitations, related to the fact that we do not know well enough the relative importance of systematic model errors and systematic errors in the measurement data, an advantage is that we correct for possible systematic model bias by modifying soil texture parameters.

Our main experiment consisted of two CLM-PDAF simulations: (a) an open-loop simulation (no data assimilation, CLM-OL) and (b) an ensemble simulation with data assimilation of ESA CCI-SM data (CLM-DA) at 100 random locations (Fig. 2a). We evaluated the results of both simulations by a cross-validation with ESA CCI-SM data that were not assimilated. The soil moisture validation of the CLM-DA and CLM-OL simulations used all the available CCI-SM data in the time

period of 2000 to 2006. This approach not only allowed us to independently cross-validate the SM values over grid cells that were not used in the data assimilation, but also to produce updated soil moisture contents at other locations (at the European scale), based on spatial correlations, and to investigate whether soil moisture characterization between measurement locations could also be improved, and its impacts on runoff characterization. For SM comparison, the average of simulated SWC in the top two layers (i.e. at 0.007 and 0.03 m depth) was used. Additionally, the monthly runoff dataset E-RUN as

described in Sec. 2.3 was used to validate runoff as simulated by CLM-OL and CLM-DA.

To assess the skill of the assimilation experiments, statistical evaluation including mean absolute error (MAE), the root mean square error (RMSE), percentage bias (PBIAS) and correlation coefficient (R) were used as validation measures. For runoff validation, Nash–Sutcliffe coefficient of efficiency (NSE) and Kling–Gupta efficiency (KGE) indices were also used which are typically used to evaluate model performance for runoff and river flow. These measures are expressed as follows:

$$MAE = \frac{1}{n}\sum_{i=1}^{n}\left(|Y_i - Y_{obs,i}|\right) \qquad (6)$$

$$RMSE = \sqrt{\frac{1}{n}\Sigma_{i=1}^{n}\left(Y_i - Y_{obs,i}\right)^2} \qquad (7)$$

$$PBIAS = 100 \times \left(\Sigma_{i=1}^{n}\left(Y_i - Y_{obs,i}\right) / \Sigma_{i=1}^{n}\left(Y_{obs,i}\right)\right) \qquad (8)$$

$$R = \frac{\left(\Sigma_{i=1}^{n}\left(Y_{obs,i} - \bar{Y}_{obs}\right)(Y_i - \bar{Y})\right)}{\left(\Sigma_{i=1}^{n}\left(Y_{obs,i} - \bar{Y}_{obs}\right)\right)\left(\Sigma_{i=1}^{n}(Y_i - \bar{Y})\right)} \qquad (9)$$

$$NSE = 1 - \frac{\Sigma_{i=1}^{n}\left(Y_{obs,i} - Y_i\right)^2}{\Sigma_{i=1}^{n}\left(Y_{obs,i} - \bar{Y}_{obs}\right)^2} \qquad (10)$$

$$KGE = 1 - \sqrt{(cc - 1)^2 + \left(\frac{\sigma_{sim}}{\sigma_{obs}} - 1\right)^2 + \left(\frac{\mu_{sim}}{\mu_{obs}} - 1\right)^2} \qquad (11)$$

where cc is Pearson correlation coefficient calculated as:

$$cc = \frac{\frac{1}{n}\Sigma_{i=1}^{n}\left(Y_{obs,i} \times Y_i\right) - \mu_{sim} \times \mu_{obs}}{\sigma_{sim} \times \sigma_{obs}} \qquad (12)$$

where $n$ is the total number of time steps; $Y_i$ and $Y_{obs,i}$ represent the simulated ensemble mean and observation values at time step $i$, respectively. $\mu_{sim}$ and $\mu_{obs}$ represent mean values, while $\sigma_{sim}$ and $\sigma_{obs}$ represent standard deviation for the simulated

and observed data for the whole modelled time period. For NSE and KGE in Eq. (10) and Eq. (11), a value equal to one represents perfect agreement between simulated and observed runoff, while a value less than 0 indicate that the observed mean is a better predictor than the model.

In addition to these measures, a normalized error reduction index (NER) was also used to evaluate the improvement of the data assimilation approach. NER is calculated as:

$$NER_\% = 100 \times \left(1 - \frac{E_{DA}}{E_{OL}}\right) \qquad (13)$$

where $E_{DA}$ and $E_{OL}$ represent the data assimilation and open loop model runs. $E$ in Eq. (13) represents the statistical error index for both RMSE and MAE in this study. NER values range between negative infinity and 100%. Positive NER values indicate improvement as result of data assimilation relative to open loop, while NER < 0 indicates a degradation in assimilation results.

## 3 Results

In this section, the impact of assimilating the ESA CCI-SM data at selected locations into CLM3.5 using the joint state-parameter estimation on the terrestrial hydrologic cycle was analyzed focusing on soil moisture and runoff. The results were presented for the complete EURO-CORDEX domain and for 8 pre-defined analysis regions from the "Prediction of Regional scenarios and Uncertainties for Defining European Climate change risks and Effects" (PRUDENCE) project (Christensen and Christensen, 2007) as shown in Fig. 1a. We referred to these regions as the "PRUDENCE" regions.

### 3.1 Impacts of assimilation on soil moisture

### 3.1.1 Regional and seasonal mean comparison

Figure 3 showed a comparison of the seasonal mean volumetric SWC ($mm^3/mm^3$) in the upper soil layer from the CLM3.5 experiments (CLM-OL, CLM-DA) with the seasonal mean of satellite CCI-SM data. The CLM-OL simulation exhibited slightly higher SWC in all seasons over most part of Europe (Fig. 3b) compared to the CLM-DA simulations (Fig. 3c). The difference between CLM-OL and CCI-SM were larger than the difference between CLM-DA and CCI-SM, which indicates that assimilation of CCI-SM minimizes the overestimation of SWC in CLM-OL. Overall, the mean difference between measured and simulated SWC was reduced from 0.11 $cm^3/cm^3$ (CLM-OL) to 0.06 $mm^3/mm^3$ (CLM-DA) over most parts of Europe (Table 1). This illustrated the efficiency of CCI-SM assimilation to improve simulated SWC by CLM. Seasonally, the upper soil layer SWC difference between CLM-OL and CCI-SM was larger for spring season than for other seasons, and this overestimation was reduced in CLM-DA (i.e. from 0.11 $mm^3/mm^3$ to 0.08 $mm^3/mm^3$; Table 1). CCI-SM assimilation also improved SWC characterization in other seasons, with the differences between CCI-SM and CLM-OL for winter, summer and autumn seasons were around 0.09 $mm^3/mm^3$ and differences between CCI-SM and CLM-DA in these seasons were reduced to the magnitude lower than 0.05 $mm^3/mm^3$.

[**Figure 3**]

[Table 1]

Figure 4 showed the comparison of 2000-2006 temporally averaged SM estimated by CLM-OL and CLM-DA with the CCI-SM dataset over PRUDENCE regions. Generally, CLM-OL overestimated the SWC values for all sub-regions and in all seasons. However, using data assimilation, this overestimation was reduced consistently in all sub-regions, as can be seen from the CLM-DA results. Noticeably, assimilation also helped to reduce the spatial variability, as indicated by the narrow spread of CLM-DA estimated SWC quartiles compared to CLM-OL in Fig. 4. Validating the simulations with CCI-SM data, the improvements of the CLM-DA varied within PRUDENCE regions and seasons. Improvements were more prominent for British Island, France, and Central Europe (for all seasons), while for other regions SWC was slightly overestimated in spring (Fig. 4b) and underestimated in summer and autumn (Fig. 4c and Fig. 4d). The underestimation of SWC was particularly pronounced over the Iberian Peninsula and Scandinavia regions in the summer (Fig. 4c).

[**Figure 4**]

The goodness of fits, including PBIAS, RMSE MAE, and correlation coefficient (R), between simulated SWC according CLM-OL or CLM-DA and CCI-SM (for the surface layer) are provided in Table 2. These statistical measures were calculated over PRUDENCE region and for each season on the basis of cross-validation with CCI-SM data that were not used in the data assimilation in order to independently evaluate the impact of data assimilation on improving soil moisture characterization. Note, that for calculating these statistics, model data were only used for the days when satellite data were available. CLM-OL showed higher PBIAS, RMSE and MAE values and lower R values than CLM-DA with CCI-SM assimilation over EU and all PRUDENCE regions (Table 2). However, the CLM-DA simulations compared well with the CCI-SM data based on the decreased of PBIAS, RMSE and/or MAE values combined with a slightly improved R values over these regions.

[Table 2]

In order to validate the skill of CLM-DA relative to CLM-OL, the NER index was applied to show the improvement with CCI-SM data assimilation in terms of RMSE and MAE using daily values of surface layer SWC for each PRUDENCE region and each season, as shown in Fig. 5. As described in Sec. 2.4, the positive NER signals indicate improvements while the negative NER signal presents degradations in the assimilation performance. The NER of RMSE (Fig. 5a) and MEA (Fig. 5b) were mostly positive over most regions indicating improvements in surface SWC estimates through assimilation of CCI-SM data. Negative NER-values (for both RMSE and MAE) were found over Scandinavia, reflecting a negative impact of CCI-SM data assimilation on SWC characterization. This might be because of uncertainties related to assimilated CCI-SM over this region due to limited amount of data because of longer winter with frozen or snow cover conditions or larger measurement errors as indicated by Dorigo et al. (2017). CLM-DA showed higher positive NER-values in the summer and autumn seasons, and lower NER-values in the winter season, related to comparatively small SWC improvements (Fig. 5a and Fig. 5b).

[**Figure 5**]

### 3.1.2 Daily validation

The long-term (January 2000 to December 2006) daily SM averaged over PRUDENCE regions in Europe, as simulated by CLM-OL and CLM-DA, and observed by CCI-SM are shown in Fig. 6. The assimilated CCI-SM data improved the simulations of daily surface soil moisture in CLM-DA. The daily soil moisture patterns simulated by CLM-DA compared well with the CCI-SM observations, with peaks and troughs generally coinciding for all regions and over the European domain except for the years 2000 and 2001. The CCI-SM observations showed increased variability and drier soil moisture values for the years 2000 and 2001 compared to the full period. This can be explained by the strong contribution of the X-band passive microwave data of SSM/I and TRMM to the final CCI-SM product. Wang (1987) showed that X-band data have a shallow soil penetration depth of a few millimeters and are sensitive to vegetation cover. After implementing the C-band radiometer data of AMSR-E in 2002 and Windsat in 2003 into CCI-SM, noise level and bias were reduced (Dorigo et al., 2017). Regionally, the daily soil moisture values estimated by the CLM-DA showed a slightly better agreement with the CCI-SM data for Central Europe regions (i.e. Iberian Peninsula, France and Mid-Europe) than Scandinavia, the Alpine, Mediterranean and Eastern Europe regions where the winter season bias was more pronounced. The overall small improvements in surface soil moisture as result of data assimilation in these regions might be due to the limited amount of CCI-SM data in the winter season (Fig. 2b), dense vegetation, frozen soil (e.g. in the Scandinavian regions) and/or CLM3.5 model errors related to simulating soil moisture in colder regions (Oleson et al., 2008; Zeng and Decker, 2009). Additionally, the magnitudes of the bias and variance of the CCI-SM observational error could be important. As indicated by Dorigo et al. (2017), the CCI-SM error variance is low where the satellite track density increases and the error variance is high in areas with more data gaps. Note that the setup of CLM-DA in this study assumed a spatially uniform observational error for CCI-SM.

[**Figure 6**]

### 3.2 Impact of soil moisture assimilation on runoff

The non-routed gridded runoff observation data from E-RUN product were used to evaluate simulated surface and subsurface runoff estimates. In order to compare with E-RUN runoff data, the total runoff was calculated as the sum of the surface and subsurface runoff for each grid cell. Figure 7 showed runoff estimates of the two experiments, i.e. CLM-OL and CLM-DA, compared to the E-RUN data. CLM-OL simulated higher magnitudes of runoff (on average 1.16 mm/day) over most parts of Europe compared to CLM-DA (on average 0.76 mm/day) in all seasons. Compared to CLM-OL (Fig. 7b), regional runoff patterns simulated by CLM-DA (Fig. 7c) compared better with runoff observations (Fig 7a). The increasing difference in runoff between E-RUN and CLM-OL simulations was more pronounced in spring and summer seasons (Fig. 7d). CLM-DA reduced this bias over most areas with respect to the E-RUN runoff data (Fig. 7e), but underestimated runoff in winter and spring particularly in central Europe. Overall, the difference in runoff between CLM-OL and E-RUN was, on average, 0.44 mm/day over Europe, which reduced to 0.03 mm/day (Table 3). At the seasonal scale, however, the difference

in winter runoff between CLM-DA and E-RUN was -0.63 mm/day and higher than the differences between CLM-OL and E-RUN (on average -0.15 mm/day). Compared to the open loop, the deviation in other seasons runoff in CLM-DA was reduced with respect to E-RUN over most part of Europe with the exception of Scandinavia and the Alpine region where negative differences became larger in all seasons (Table 3).

[Figure 7]

[Table 3]

The temporally averaged runoff for all grid cells over all PRUDENCE regions for both CLM-OL and CLM-DA experiments and comparison with E-RUN observation data is presented in the boxplots in Fig. 8. The box plots reflect the distribution of runoff in which quartile and median are marked by solid lines. In comparison with the E-RUN data in the winter season, the CLM-DA simulations underestimated runoff over most regions, while open loop simulations showed better agreement with E-RUN runoff over most of the grid cells (Fig. 8a). However, the overestimation of runoff in CLM-OL was more obvious in the spring season over all regions (Fig. 8b), while assimilating CCI-SM data minimized this overestimation but introduced a dry bias as suggested by lower values for CLM-DA runoff with respect to CLM-OL and E-RUN observations. This underestimation of runoff as a result of soil moisture assimilation was more pronounced over British Island in all seasons and over Scandinavia in summer and autumn seasons (Fig 8c and 8d).

[Figure 8]

The time series of monthly runoff, as illustrated in Fig. 9, showed that CLM-OL compared well with runoff observations over British Island, Iberian Peninsula, France and the Mid-Europe regions, but overestimated the magnitude of runoff in the Mediterranean. Scandinavia, Alpine and Eastern Europe regions. When compared to open loop, CLM-DA performed better than CLM-OL (compared to E-RUN) in Mid-Europe, Scandinavia, Alpine and Eastern Europe in capturing peaks and low runoff, while in other regions such as the British Island, Iberian Peninsula, France and the Mediterranean, peak runoff in winter was underestimated whereas low runoff in summer was in correspondence with observed monthly runoff data. The uncertain performance of soil moisture assimilation on peak runoff simulations mainly lies in relatively weak dependence of runoff generation on antecedent soil moisture because during high flow periods, the soil moisture is nearly saturated and the runoff is largely controlled by precipitation. These results were consistent with those of previous research; for example the studies of Albergel et al. (2017) and Liu et al. (2018) showed that assimilated ESA CCI satellite-derived soil moisture data into the land surface models improved the surface soil moisture but found little improvements in discharge compared to the open loop simulations.

[Figure 9]

In terms of statistical measures, the runoff simulation based on CLM-OL showed higher PBIAS than CLM-DA over EU and most PRUDENCE regions, except British Island, Scandinavia and Alpine regions, where higher negative percentage biases were observed for CLM-DA with the magnitude of -60, -54 and -58% bias in runoff, respectively (Table 4). Additionally, the NSE and KGE values over these regions showed low positive to negative values for the CLM-DA scenario. This indicated poor performance of the CLM model in simulating runoff in spite of soil moisture assimilation. To better illustrate

the impact of assimilating soil moisture on model estimates of runoff, the NER index of both RMSE and MAE showed positive values for Iberian Peninsula, France, the Mediterranean and East Europe regions, indicating improvements in runoff (Fig. 10). However, negative signals in NER were observed in winter for all regions except in East Europe. Negative NER values were mainly located over British Island, Scandinavia and Alpine regions in all seasons. The negative impact of ESA CCI SM assimilation on runoff simulation over Scandinavia and Alpine regions is probably related to their large proportions of the dense forest and complex topography. Both dense forest coverage and complex topography reduce the data quality of remotely sensed SM retrievals, thus impeding its performance in DA.

[**Table 4**]

[**Figure 10**]

**4 Discussion**

This study demonstrated that the assimilation of coarse-scale satellite CCI soil moisture data is beneficial and improves the high-resolution CLM model simulations of soil moisture and runoff over a large spatial domain. This study also highlighted the added value of merging coarse resolution satellite observations through data assimilation with a land surface model to generate higher spatial resolution, downscaled estimates of the surface soil moisture profile with complete spatio-temporal coverage and with a higher accuracy than that of the open loop model estimates. To the best of our knowledge, this is the first study to provide a downscaled daily soil moisture product over seven years and at 3 km resolution over Europe.

An important challenge in the assimilation was the difference in spatial resolution between CCI-SM data and model states. In this study, the coarser resolution CCI-SM data was rescaled to the model resolution (3 km). To examine whether the rescaling of the ESA CCI SM data to model resolution may introduce any bias in the data, we compared the original 0.25° against 0.0275° ESA CCI soil moisture. Only small differences between the two resolutions were visible particular for the time period of 2003-2006 (Fig. S5). We found some differences in the first two years (2000 and 2001) and in the regions where the temporal coverage of the ESA CCI data is less than 30%. However, for the time period and regions with a good coverage of ESA CCI soil moisture data, the differences in the resolution were not significant. A further possibility is the multiscale assimilation of the CCI-SM data, which would allow to update various model grid cells covered by a satellite observation (Montzka et al., 2012). In multiscale assimilation, the average soil moisture content for the group of grid cells covered by the satellite measurement is compared with the satellite-based soil moisture content which may result in slightly improved CLM simulation results, but was beyond the scope of this study. In addition to discrepancies at the spatial scale, uncertainties in soil moisture estimations may result from data gaps in satellite soil moisture retrievals, which are limited in regions of pronounced topography, standing water, areas of dense vegetation and snow covered areas and frozen soil. Additionally, CCI-SM is a merged product from a variety of sensors leading to inconsistencies due to differences in viewing angle, sensor characteristics and soil moisture retrieval algorithms (Dorigo et al., 2017). In future, more observations are needed to independently validate model and assimilation experiments. In this work, the CCI-SM dataset was also used for

verification over grid cells that were not used in the data assimilation. However, it would be preferable to validate with another independent dataset at the continental scale. The problem is that at the model grid scale only very limited independent (*in situ*) soil moisture data are available. Furthermore, it can be difficult to compare the point-based observation with the average value of coarse resolution model grid cell.

Another challenge to implement the integrated hydrologic and data assimilation framework at 3 km resolution was the high computational cost associated with the EnKF which relies on an ensemble of realizations to estimate model uncertainty. We evaluated the impact of the number of ensembles members (i.e. 12 and 20) on the performance of EnKF by comparing the surface soil layer SWC simulated by CLM-DA for one test year (i.e. 2006). While, some improvements were observed in the simulated soil moisture when using 20 instead of 12 ensemble members, in general the simulated soil moisture from the DA-

runs with 12 and 20 ensemble members are quite close to the observed values (Fig. S5). It should be noted that using an increased number of ensemble size is a big challenge for such a large-scale high-resolution model because of the memory and storage requirements, and to a lesser degree also because of the computational burden. For example, one year of model run with 20 ensemble members required 680GB of computer storage per output variable (i.e. equivalent to 5TB of storage for 7 years of simulations per variable at daily time scale) and resulted in the use of 76,800 CPU core-hours (compare to

46,000 core-hours with 12 ensemble members).

The assimilation framework used in this study explicitly accounted for uncertainty in the model forcing data (e.g. precipitation) and soil texture properties (%sand and % clay) and used the joint state and parameter estimation to reduce parameter uncertainty. While parameter updating is expected to correct part of the systematic model bias, there is a potential that other water balance terms like evapotranspiration and also runoff may be degraded through data assimilation to

compensate for model structural and/or input data errors. To investigate this, we evaluated the modeled total runoff (surface and subsurface) with non-routed gridded runoff observations. While we found overall improvement in percent bias over EU-CORDEX domain (i.e., from 60% to 4% over Europe, Table 4) and smaller improvements over different PRUDENCE regions after the assimilation, there were some degradation of runoff estimates after soil moisture assimilation (Fig. 10) for many regions, The differences in runoff between CLM-DA and CLM-OL were small, since the fluxes were often reproduced

reasonably well in CLM-OL with an average correlation of 0.9 over the EU-CORDEX domain (Table 3). The degradation in runoff over some regions and overall marginal improvements might be due to several factors, including a lack of analysis updates of water flux terms, model structure error (e.g., weak coupling of runoff processes with water table dynamics and soil water storage), and observations errors in the monthly data evaluation. From a comparison of individual components of total runoff between CLM-OL and CLM-DA, we found that the assimilation of soil moisture into CLM model has greater

impacts on subsurface runoff than on surface runoff (Fig. 11). As shown in Fig. 11, assimilation of CCI-SM data resulted in an underestimation of subsurface runoff over all regions, but has less impact on the surface runoff. This underestimation might be related to the CLM model limitations to correctly represent processes controlling the partitioning of subsurface and surface flow in the model and/or to the exponential form of runoff parameterization. For example in the CLM model, the overall saturation status of the soil column is the controlling factor for both surface and subsurface runoff generation. Thus

any changes in the surface layer soil moisture will also have more profound effects on the subsurface flow. In addition, the surface runoff generation is based on the assumption of saturation excess runoff, meaning that the water table needs to intersect the surface before the surface runoff is generated. This assumption is problematic at the large spatial scales, especially in arid and semi-arid regions. In dry regions, assimilation of soil moisture data may result in reduction of soil moisture values close to the residual water content values, which may lead to small surface runoff generation. For example, (Sheng et al., 2017) also found that CLM exhibited limitations in water-limited areas where surface runoff is determined by groundwater dynamics and identified that the saturation excess surface runoff assumption as the main cause of these limitations.. Additionally, the assumption of topographically controlled surface runoff generation in the CLM model is problematic in areas with flat topography, thick soils, or deep groundwater (Li et al., 2011). Another reason may be uncertainties of E-RUN runoff data used in this study, which were derived from gridded atmospheric variables and flow observations at coarser resolution (0.5° x 0.5° grid resolution). In the future, additional observational data need to be explored in assimilation experiments.

This study only considered uncertainty with respect to soil texture parameters, while other soil and ecosystem parameters were assumed deterministic. In data assimilation, it is preferable to account for additional model parameter uncertainties that show a high sensitivity towards runoff. Alternatively, prior model calibration can be considered to constrain model parameters better and reduce systematic biases and uncertainties in CLM3.5 before applying the assimilation framework. For improvement of hydrologic predictions, joint assimilation of additional datasets such as river discharge and snow data may also be considered in future research.

## 5 Conclusions

A soil moisture data assimilation framework at the continental scale was applied to generate daily soil moisture and runoff estimates as part of a terrestrial systems monitoring framework for Europe at 3 km resolution for the years 2000 to 2006. An ensemble was generated by perturbing precipitation and soil texture properties. These ensembles were used as input in the CLM-PDAF data assimilation framework (Kurtz et al., 2016) and used to assimilate CCI-SM soil moisture data. The impact of satellite soil moisture assimilation on daily soil moisture and monthly runoff was evaluated and cross-validated with CCI-SM data and gridded runoff from E-RUN observations at regional and seasonal scales. Using this high-resolution CLM-PDAF setup, the conclusions of this study are:

1. Assimilation of satellite SM improved the soil moisture simulations over most parts of Europe relative to open-loop simulations. Open loop simulations overestimated SM in most parts of Europe and in all seasons. For the study domain, on average, the RMSE for near surface SWC was reduced from 0.10 $mm^3/mm^3$ in the open-loop simulations to 0.07 $mm^3/mm^3$ with SM assimilation.

2. Regionally, significant improvements were achieved for soil moisture across most regions, except over Scandinavia. The low performance of CLM-DA in these regions might be due to the lack of data in space and

time, as caused by satellite track changes, radio-frequency interference, dense vegetation, snow and frozen soil limiting the assimilation of soil moisture data in land surface processes simulations. Analogously, CLM-DA performed poorly for years 2000 and 2001, which appears to be related to large data gaps and higher noise levels in the CCI-SM satellite data in these years. This indicates suitability of ESA CCI-SM for data assimilation studies from 2002 onwards, whereas the accuracy and noise level of earlier data is not appropriate for this purpose.

3. At the seasonal time scale, the CLM-DA simulations performed better in the summer and autumn seasons than in the winter and spring seasons. This might be again related to large data gaps in the winter season or model limitations to correctly represent complex cold region processes such as frozen soil.

4. The assimilation of CCI-SM data into CLM3.5 led to an overall marginal improvement in the simulated total runoff over Europe. Improvements in runoff were more prominent over the Iberian Peninsula, Mediterranean and East-Europe regions, where CLM-DA, on average, minimized the difference to E-RUN from 0.65 mm/day, 0.54 mm/day and 0.66 mm/day, to 0.25 mm/day, 0.14 mm/day and 0.25 mm/day, respectively (Table 3). The improvements over other regions, such as the British Island, France, Mid-Europe, Scandinavia and Alpine, were comparatively small. These findings indicated the potential of satellite soil moisture assimilation in CLM3.5 to improve other terrestrial components of the water cycle as a basis for more accurate water balance analyses.

The results from this study are not only useful as a standalone high-resolution reanalysis product over Europe, but can also be used as an independent dataset for validation of other land surface models. In this study, the soil moisture estimates with improved spatial resolution obtain via data assimilation offer a new product for monitoring soil water content with distinct benefits over the original CCI-SM data. In addition, by selecting the ESA CCI soil moisture product for assimilation, the potential impact of the long term soil moisture observations on hydrologic simulations can be assessed for climate change studies. Recently, with the availability of COSMO-REA6, the time period can be extended to 2000-2015 in future studies using the proposed methodology to derive a land surface reanalysis at 3km resolution for continental Europe. Moreover, CLM3.5 is also part of the fully coupled Terrestrial Systems Model Platform (TerrSysMP) (Gasper et al., 2014; Keune et al., 2016; Shrestha et al., 2014) that simulates the full terrestrial hydrologic cycle including feedbacks between atmosphere, land-surface and subsurface compartments of the water cycle. The impact of satellite soil moisture assimilation on other water cycle variables across the soil-vegetation-atmosphere system using TerrSysMP and its effects on the accuracy of model simulations at the continental scale remains to be explored.

**Acknowledgements**

The authors gratefully acknowledge the computing time granted by the JARA-HPC Vergabegremium on the supercomputer JURECA at Forschungszentrum Jülich. This work was supported by the Energy oriented Centre of Excellence (EoCoE),

grant agreement number 676629, funded within the Horizon2020 framework of the European Union. C. Montzka was funded by BELSPO (Belgian Science Policy Office) in the framework of the STEREO III programme – HYDRAS+ (SR/00/302) and the ERA-PLANET/GEOEssential project (Horizon2020, grant agreement number 689443). W. Kurtz was supported by SFB-TR32 "Patterns in soil–vegetation–atmosphere systems: monitoring, modelling and data assimilation" funded by the German Science Foundation (DFG). We thank the anonymous reviewers and the editor for their constructive comments and suggestions that helped us to improve the paper significantly.

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

**Tables and Figures**

**Table 1. Difference in CLM-OL and CLM-DA simulated mean seasonal SWC (mm$^3$/mm$^3$) with CCI-SM data for all PRUDENCE regions and all seasons, i.e. winter (DJF), spring (MAM), summer (JJA) and autumn (SON).**

| Regions | CLM-OL minus CCI-SM | | | | CLM-DA minus CCI-SM | | | |
|---|---|---|---|---|---|---|---|---|
| | Winter | Spring | Summer | Autumn | Winter | Spring | Summer | Autumn |
| **BI** | 0.03 | 0.05 | 0.02 | 0.04 | 0.01 | 0.02 | -0.02 | 0.00 |
| **IP** | 0.07 | 0.09 | 0.11 | 0.09 | 0.04 | 0.06 | 0.07 | 0.05 |
| **FR** | 0.03 | 0.06 | 0.06 | 0.05 | 0.01 | 0.03 | 0.02 | 0.02 |
| **ME** | 0.04 | 0.07 | 0.06 | 0.05 | 0.02 | 0.04 | 0.01 | 0.02 |
| **SC** | 0.07 | 0.08 | 0.02 | 0.03 | 0.04 | 0.05 | -0.02 | 0.00 |
| **AL** | 0.03 | 0.05 | 0.04 | 0.02 | 0.01 | 0.02 | 0.00 | -0.01 |
| **MD** | 0.05 | 0.07 | 0.08 | 0.06 | 0.03 | 0.04 | 0.04 | 0.03 |
| **EA** | 0.07 | 0.08 | 0.07 | 0.05 | 0.05 | 0.05 | 0.03 | 0.02 |
| **EU** | 0.09 | 0.11 | 0.09 | 0.09 | 0.07 | 0.08 | 0.05 | 0.06 |

**Table 2. Evaluation performance criteria for comparing CLM-OL and CLM-DA with CCI-SM (spatially averaged SWC, surface layer, EU and PRUDENCE regions)**

| | Soil Moisture (CLM-OL) | | | | | | | | |
|---|---|---|---|---|---|---|---|---|---|
| | EU | BI | IP | FR | ME | SC | AL | MD | EA |
| **PBIAS (%)** | 54.1 | 16.4 | 50.7 | 24.7 | 25.6 | 23.0 | 16.4 | 33.4 | 34.8 |
| **RMSE (mm$^3$/mm$^3$)** | 0.10 | 0.05 | 0.10 | 0.07 | 0.07 | 0.06 | 0.05 | 0.07 | 0.08 |
| **MAE (mm$^3$/mm$^3$)** | 0.09 | 0.04 | 0.09 | 0.06 | 0.06 | 0.05 | 0.04 | 0.07 | 0.07 |
| **R** | 0.48 | 0.41 | 0.75 | 0.60 | 0.51 | -0.14 | 0.55 | 0.80 | 0.40 |
| | Soil Moisture (CLM-DA) | | | | | | | | |
| | EU | BI | IP | FR | ME | SC | AL | MD | EA |
| **PBIAS (%)** | 36.4 | 3.1 | 33.2 | 10.1 | 11.0 | 9.0 | 3.0 | 18.1 | 19.0 |
| **RMSE (mm$^3$/mm$^3$)** | 0.07 | 0.03 | 0.07 | 0.04 | 0.04 | 0.05 | 0.03 | 0.04 | 0.06 |
| **MAE (mm$^3$/mm$^3$)** | 0.06 | 0.03 | 0.06 | 0.03 | 0.03 | 0.04 | 0.03 | 0.04 | 0.05 |
| **R** | 0.51 | 0.40 | 0.76 | 0.61 | 0.54 | -0.12 | 0.56 | 0.80 | 0.42 |

**Table 3. Monthly mean bias (CLM minus E-RUN) in mean seasonal runoff (mm/day) for CLM-OL and CLM-DA for all PRUDENCE regions and all seasons, i.e. winter (DJF), spring (MAM), summer (JJA) and autumn (SON).**

| | CLM-OL minus E-RUN | | | | CLM-DA minus E-RUN | | | |
|---|---|---|---|---|---|---|---|---|
| **Regions** | Winter | Spring | Summer | Autumn | Winter | Spring | Summer | Autumn |
| **BI** | -1.68 | -0.19 | 0.09 | -1.26 | -2.19 | -0.79 | -0.18 | -1.60 |
| **IP** | 0.35 | 0.94 | 0.75 | 0.57 | -0.12 | 0.37 | 0.49 | 0.27 |
| **FR** | -0.26 | 0.52 | 0.62 | 0.33 | -0.77 | -0.08 | 0.36 | 0.02 |
| **ME** | -0.02 | 0.63 | 0.51 | 0.36 | -0.50 | 0.04 | 0.25 | 0.05 |
| **SC** | -0.11 | -0.27 | -0.98 | -0.83 | -0.58 | -0.86 | -1.25 | -1.15 |
| **AL** | -0.25 | -0.72 | -1.04 | -0.96 | -0.72 | -1.33 | -1.32 | -1.29 |
| **MD** | 0.14 | 0.88 | 0.70 | 0.45 | -0.33 | 0.29 | 0.44 | 0.14 |
| **EA** | 0.66 | 0.90 | 0.54 | 0.56 | 0.19 | 0.30 | 0.27 | 0.24 |
| **EU** | 0.39 | 0.68 | 0.38 | 0.30 | -0.08 | 0.09 | 0.12 | -0.01 |

**Table 4. Evaluation performance criteria for comparing CLM-OL and CLM-DA with E-RUN (spatially averaged runoff, EU and PRUDENCE regions).**

| Total Runoff (CLM-OL) | | | | | | | | | |
|---|---|---|---|---|---|---|---|---|---|
| | **EU** | **BI** | **IP** | **FR** | **ME** | **SC** | **AL** | **MD** | **EA** |
| **PBIAS** | 59.7 | -38.6 | 130.8 | 34.2 | 46.1 | -30.9 | -37.4 | 87.5 | 130.3 |
| **RMSE (mm/day)** | 0.5 | 1.3 | 0.8 | 0.7 | 0.6 | 0.9 | 0.9 | 0.7 | 0.7 |
| **MAE (mm/day)** | 0.4 | 0.9 | 0.7 | 0.6 | 0.5 | 0.8 | 0.8 | 0.6 | 0.7 |
| **NSE** | -4.0 | -0.2 | -1.8 | 0.3 | -0.3 | -0.4 | -0.4 | -1.6 | -10.2 |
| **KGE** | -0.1 | 0.1 | -0.4 | 0.3 | 0.4 | 0.1 | 0.3 | 0.0 | -0.6 |
| **R** | 0.9 | 0.5 | 0.6 | 0.7 | 0.6 | 0.3 | 0.7 | 0.5 | 0.6 |
| **Total Runoff (CLM-DA)** | | | | | | | | | |
| | **EU** | **BI** | **IP** | **FR** | **ME** | **SC** | **AL** | **MD** | **EA** |
| **PBIAS** | 4.4 | -60.2 | 50.5 | -13.3 | -5.1 | -54.1 | -58.9 | 21.8 | 48.9 |
| **RMSE (mm/day)** | 0.3 | 1.6 | 0.6 | 0.7 | 0.5 | 1.2 | 1.3 | 0.5 | 0.4 |
| **MAE (mm/day)** | 0.2 | 1.2 | 0.5 | 0.5 | 0.4 | 1.0 | 1.2 | 0.5 | 0.3 |
| **NSE** | -0.7 | -0.9 | -0.6 | 0.1 | -0.1 | -1.2 | -1.8 | -0.6 | -2.6 |
| **KGE** | 0.2 | -0.1 | 0.0 | 0.2 | 0.3 | 0.1 | 0.1 | 0.1 | 0.0 |
| **R** | 0.6 | 0.4 | 0.2 | 0.4 | 0.3 | 0.5 | 0.6 | 0.1 | 0.4 |

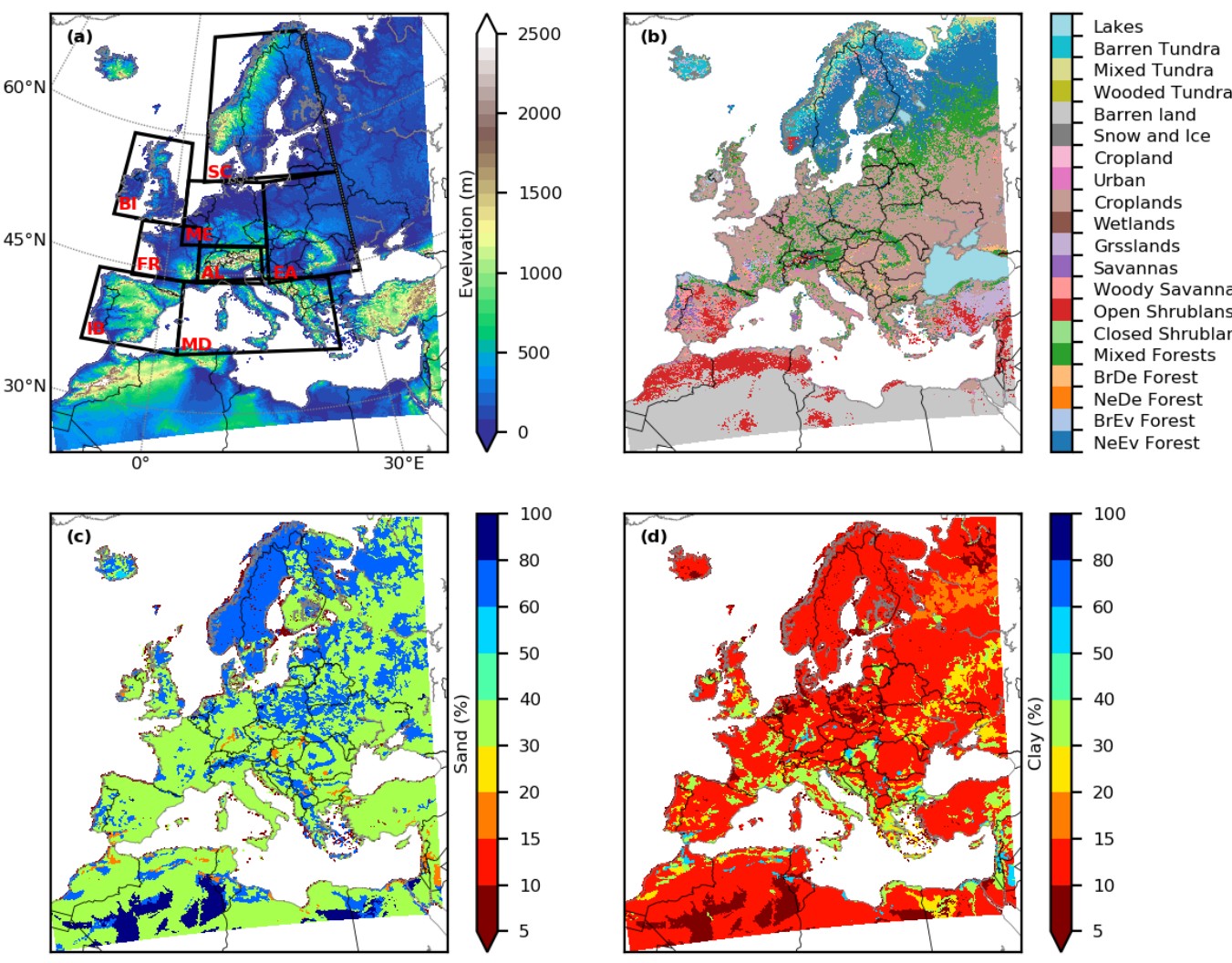

Figure 1. Model surface input data: a) USGS GMTED2010 DEM, b) dominant land use type based on MODIS data, c) percent sand content, and d) percent clay content based on global FAO soil database. The inner boxes in (a) show the boundaries of the PRUDENCE regions (FR: France, ME: mid-Europe, SC: Scandinavia, EA: Eastern Europe, MD: Mediterranean, IP: Iberian Peninsula, BI: British Islands, AL: Alpine region; Christensen and Christensen, 2007).

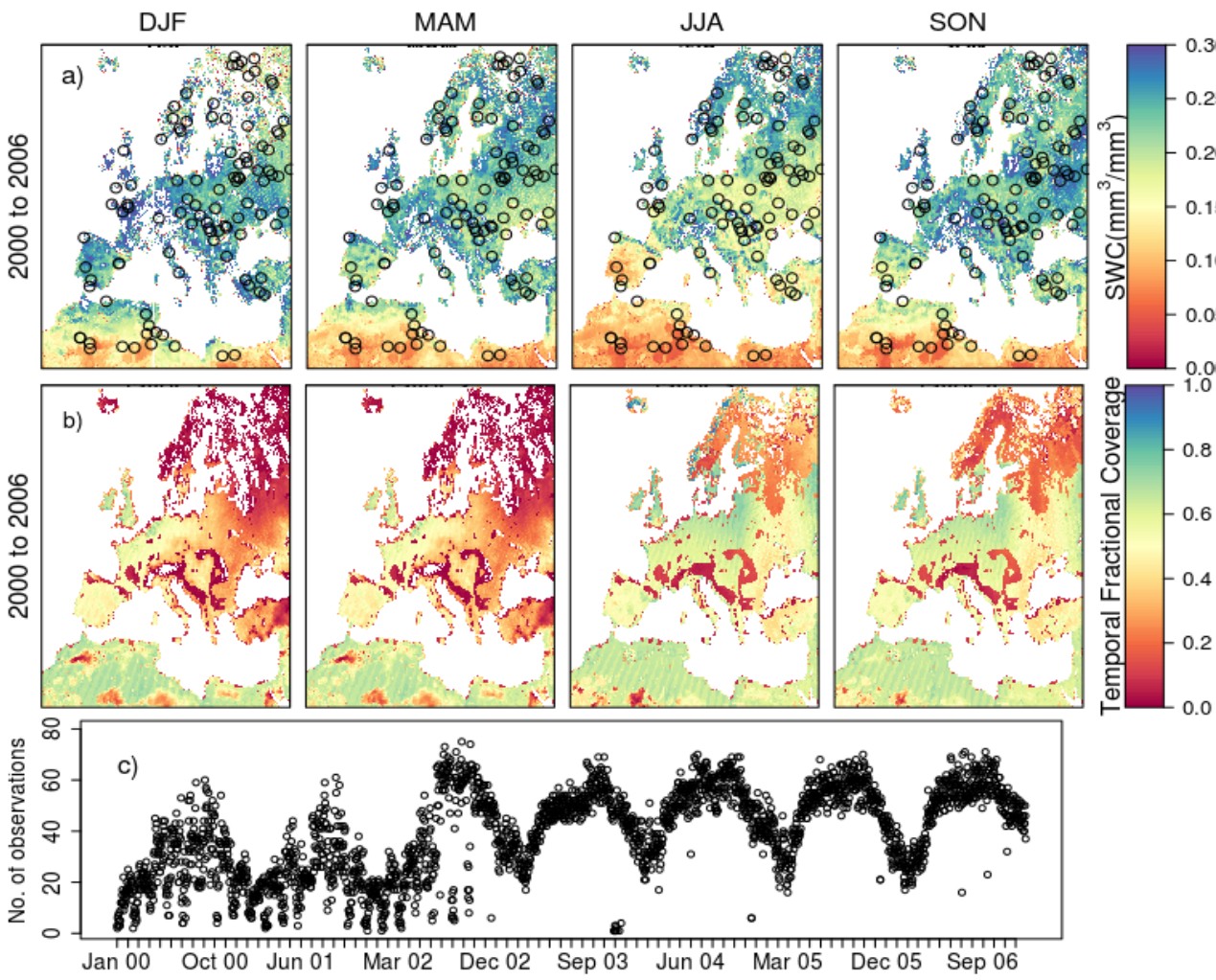

**Figure 2. Satellite ESA-CCI soil moisture data resampled to 3 km resolution for the time period of 2000 to 2006 over EU-CORDEX. (a) Temporally averaged soil moisture content for different seasons, (b) fraction of days that soil moisture observations were reported during different seasons, and (c) number of selected observations with valid data for the respective day over the 2000-2006 period used for assimilating soil moisture in the data assimilation experiment. Black circles in (a) indicate the location of grid cells selected for data assimilation.**

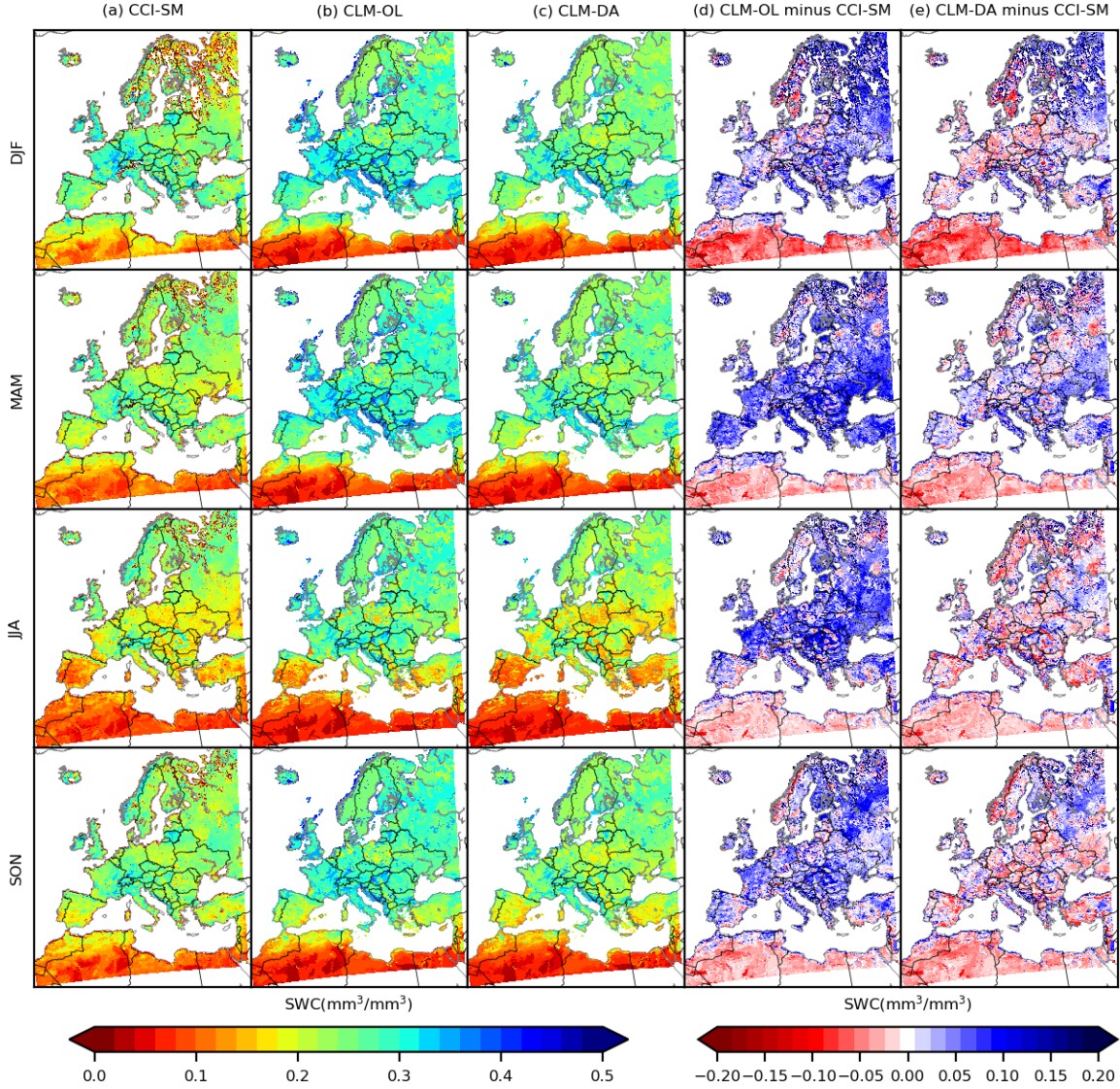

**Figure 3. Temporally averaged soil moisture (mm$^3$/mm$^3$) content over the 2000 – 2006 period for a) CCI-SM, b) CLM-OL c) CLM-DA, and difference between d) CLM-OL and CCI-SM and d) CLM-DA and CCI-SM for DJF (December, January and February), MAM (March, April, May), JJA (June, July and August) and SON (September, October, November) seasons.**

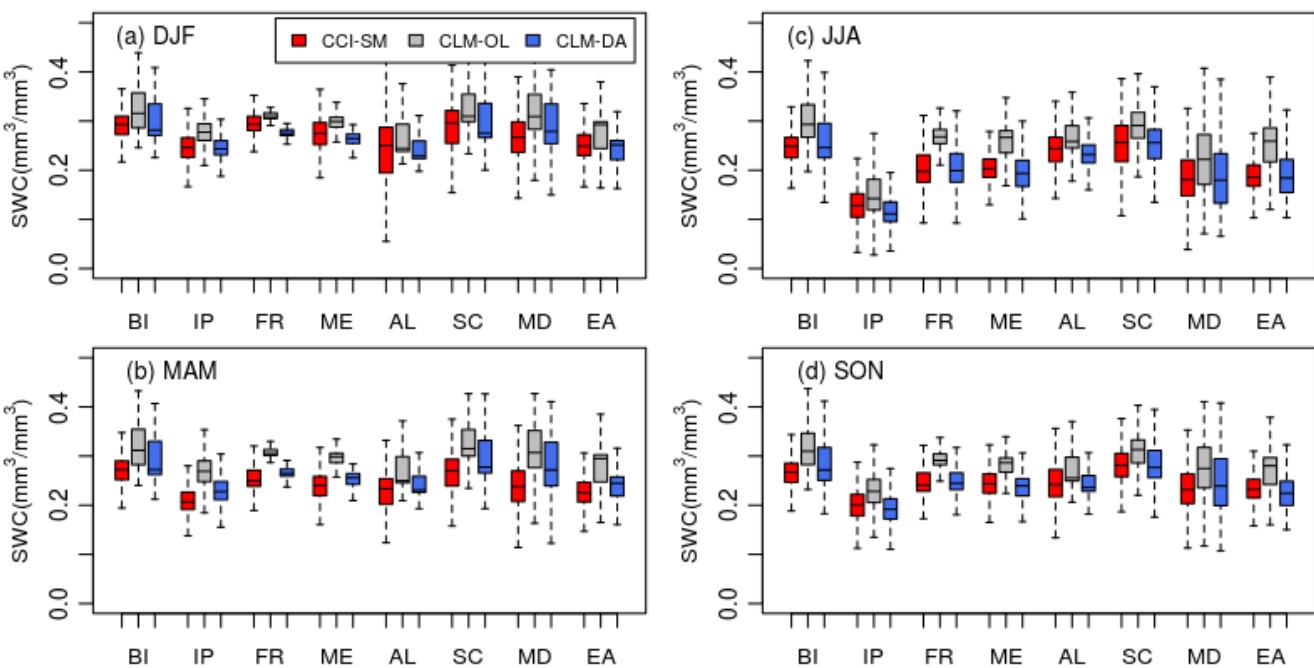

Figure 4. Box plots showing the spread of seasonally averaged soil water content (mm³/mm³) over the 2000 – 2006 time period and in the PRUDENCE regions for (a) DJF, (b) MAM, (c) JJA and (d) SON seasons. The boxplots illustrate the spatial distribution of SWC with quartiles, median and extreme values marked by solid lines.

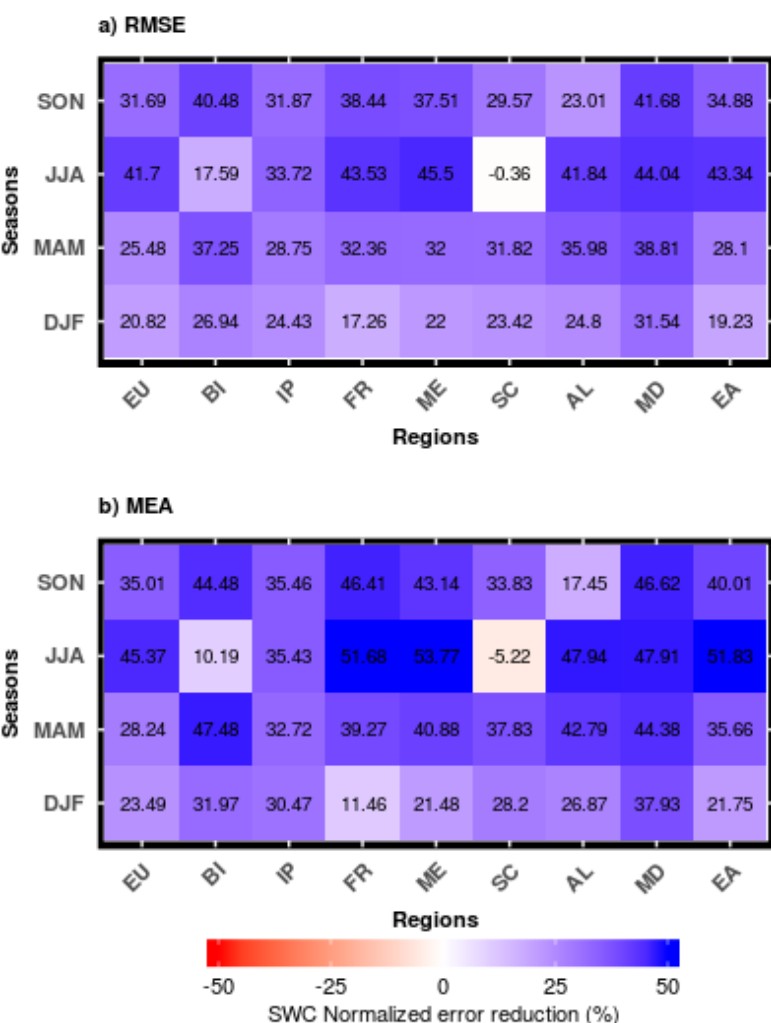

**Figure 5. Normalized error reduction (NER) index of (a) RMSE and (b) MEA for daily soil water content over different seasons and PRUDENCE regions using CLM-OL and CLM-DA simulations over the years 2000 – 2006.**

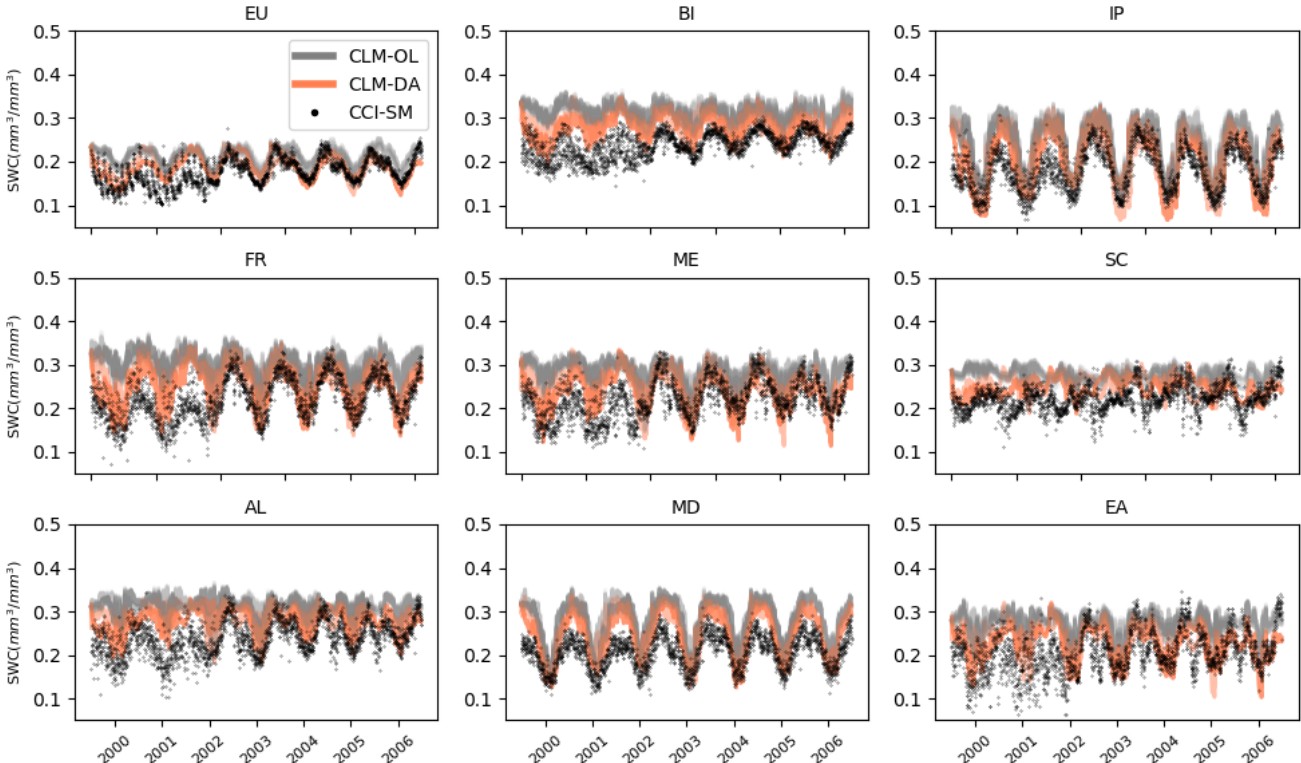

**Figure 6. Spatially averaged daily soil water content (SWC) simulated with CLM-DA and CLM-OL and compared with CCI-SM data for the years 2000 – 2006 over Europe and the PRUDENCE regions. The orange and gray lines are the CLM-DA and CLM-OL 20 ensemble members, respectively.**

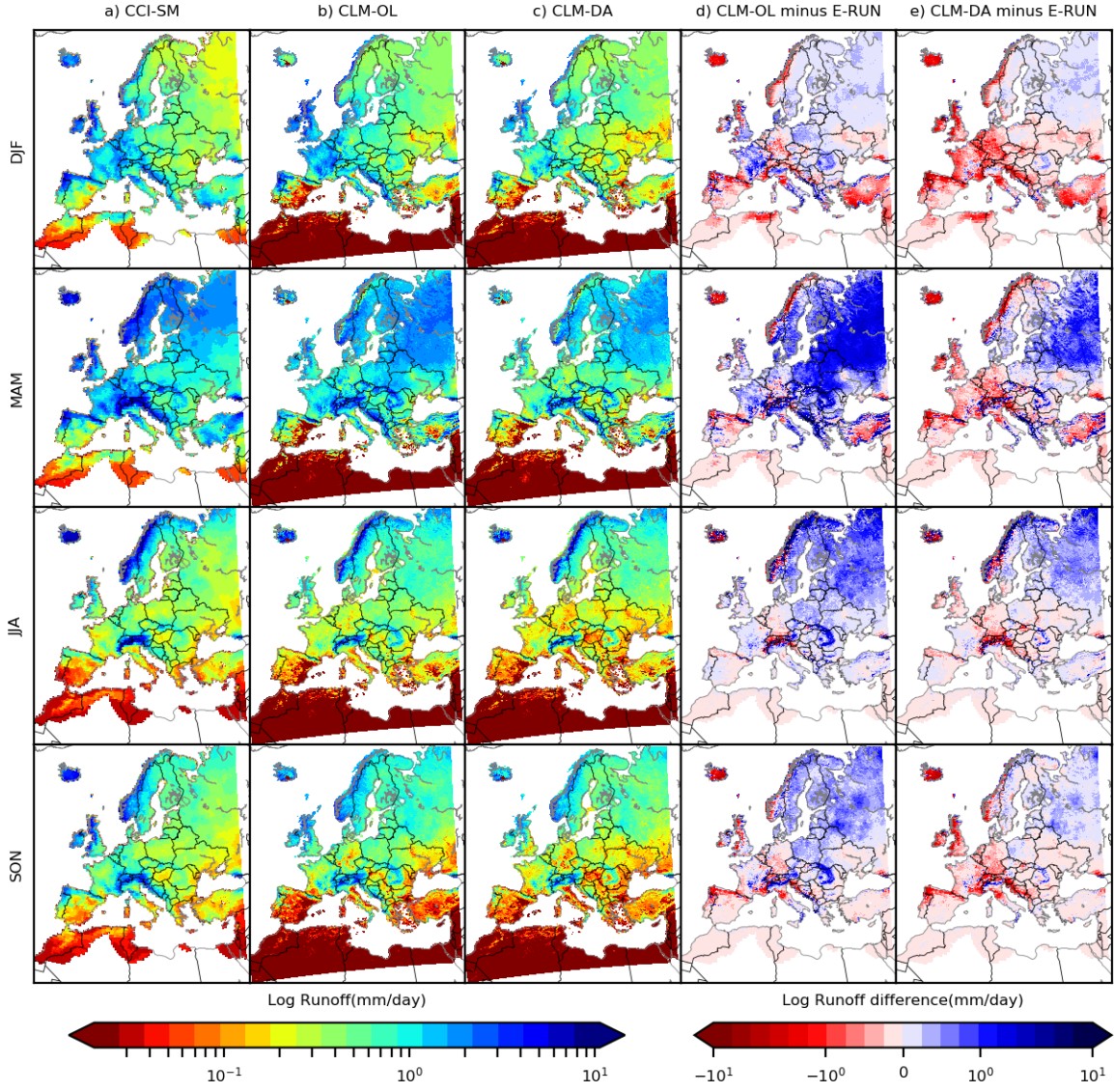

**Figure 7. Temporally averaged monthly runoff (mm/day) at log scale over the 2000 – 2006 period for a) E-RUN, b) CLM-OL c) CLM-DA, and difference between d) CLM-OL and E-RUN and d) CLM-DA and E-RUN for DJF, MAM, JJA and SON seasons.**

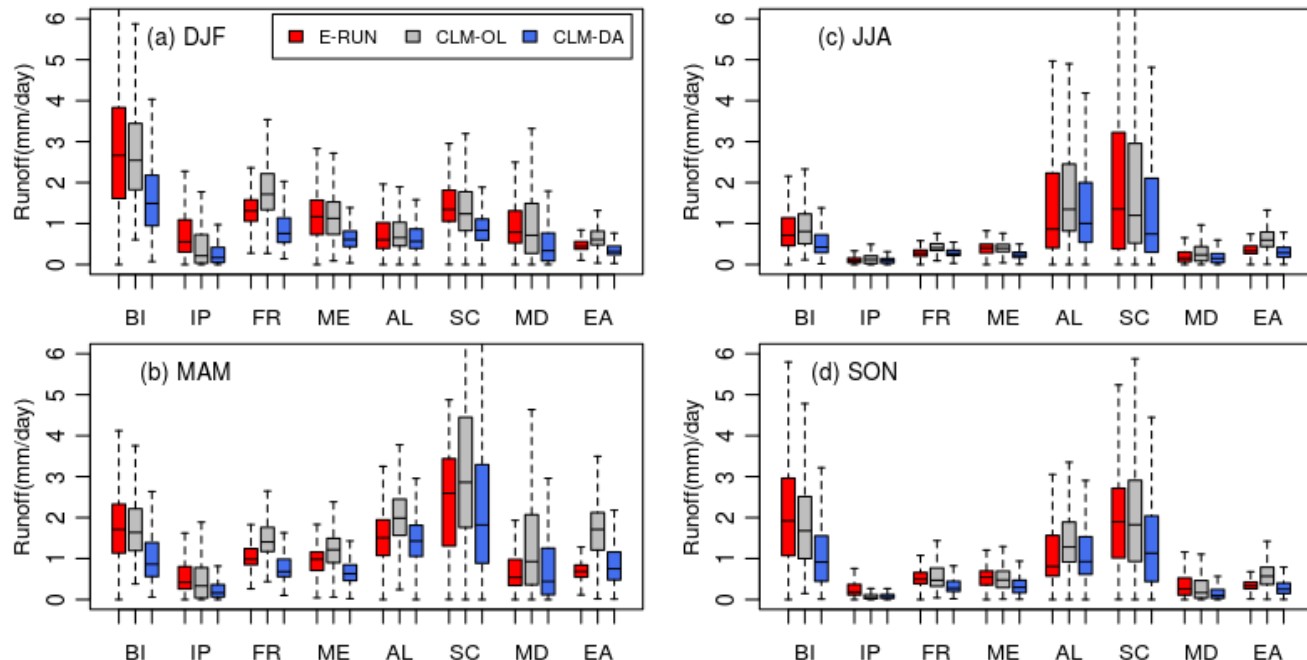

**Figure 8. Boxplots of temporally averaged runoff (mm/day) over the years 2000 – 2006 for all PRUDENCE region and seasons, i.e. (a) DJF, (b) MAM, (c) JJA and (d) SON. The boxplots indicate the spatial distribution of monthly averaged runoff over each region.**

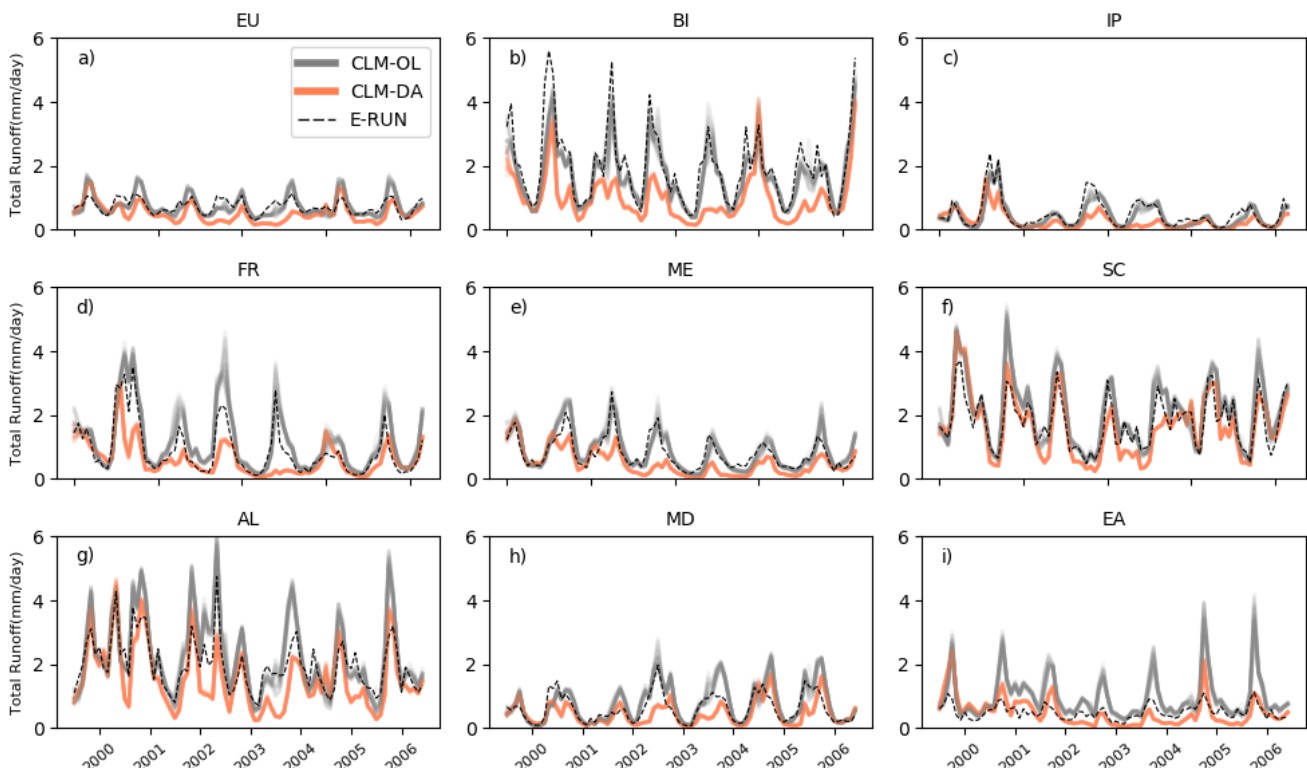

**Figure 9. Monthly time series of runoff from CLM-DA and CLM-OL simulation and compared with E-RUN runoff observation data for the years 2000 – 2006 over Europe and PRUDENCE regions. The orange and gray lines are the CLM-DA and CLM-OL 20 ensemble members, respectively.**

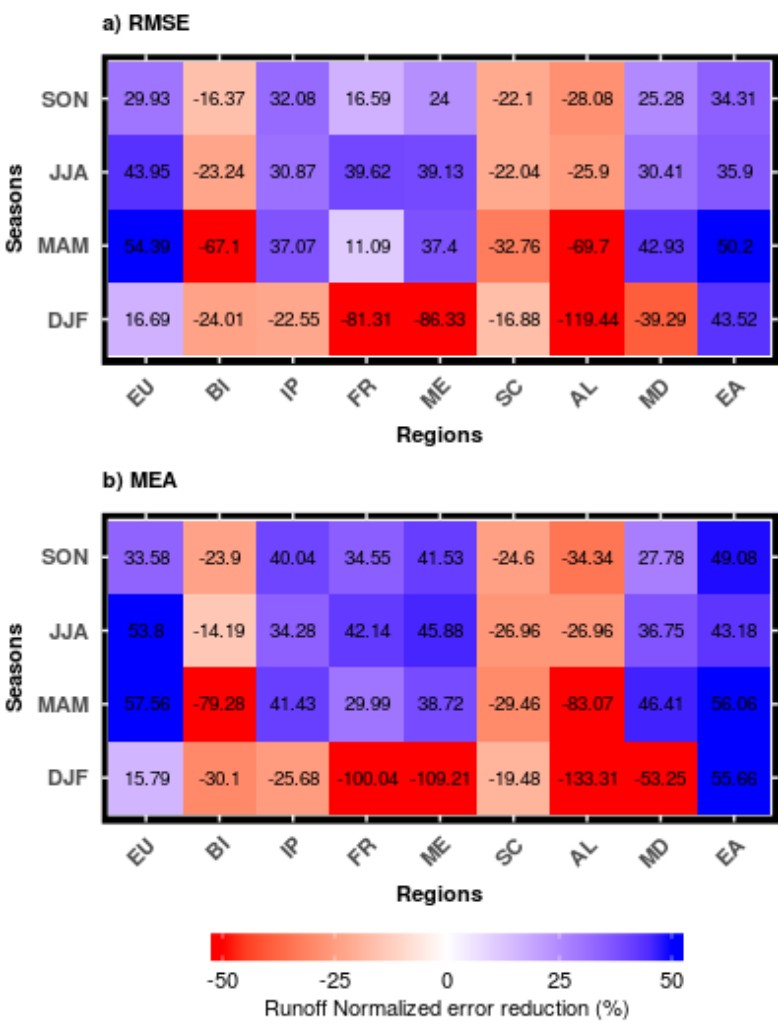

**Figure 10. Normalized error reduction (NER) index of (a) RMSE and (b) MEA for runoff over different seasons and PRUDENCE regions using CLM-OL and CLM-DA simulations over the years 2000 – 2006.**

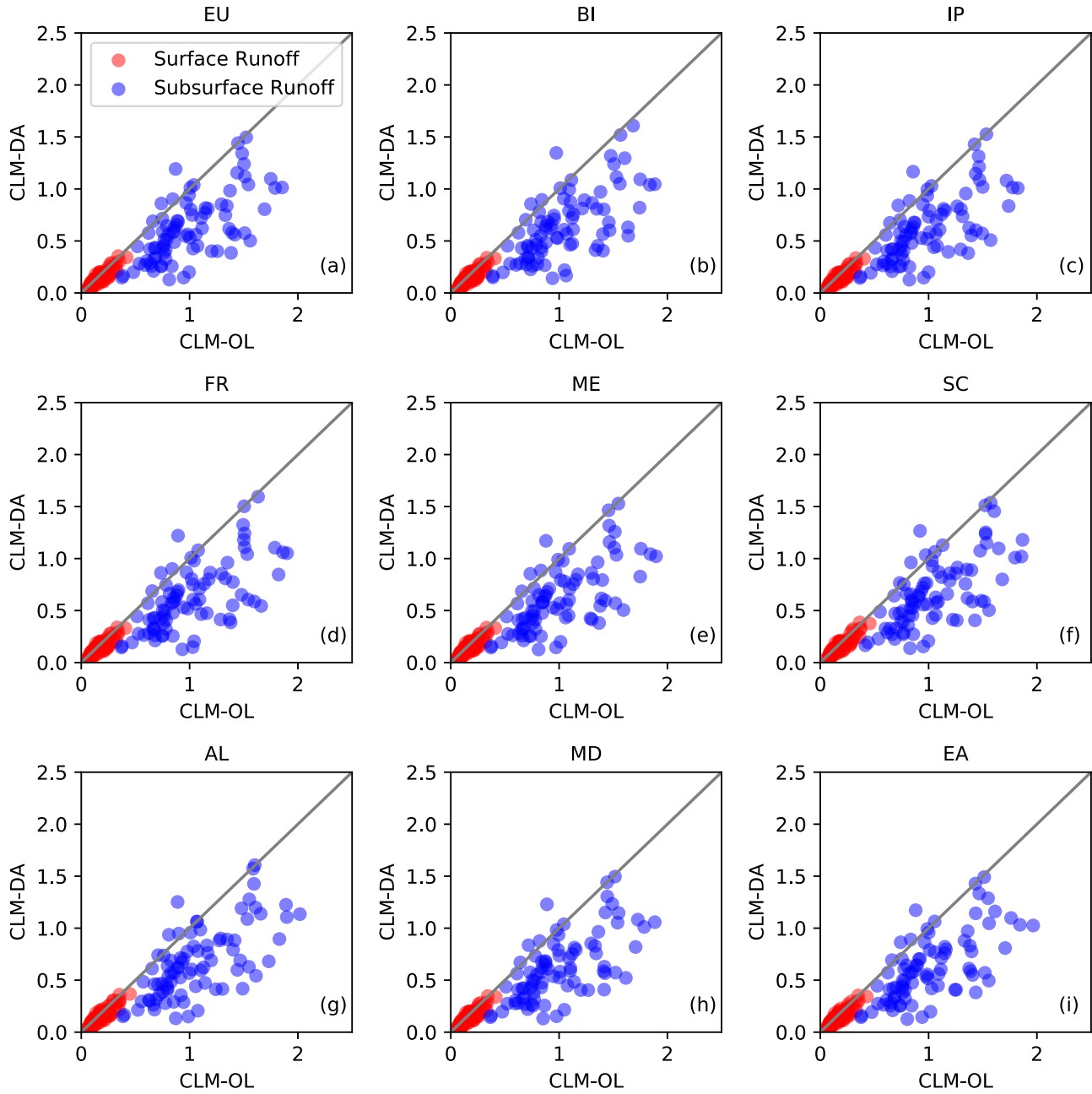

**Figure 11. Comparison of spatially-averaged monthly simulated runoff components (surface and subsurface runoff) between CLM-OL and CLM-DA over Europe and PRUDENCE regions for the time period of 2000 − 2006. Gray line represents 1:1 line.**