# Peer review of "Improving soil moisture and runoff simulations at 3 km over Europe using land surface data-assimilation"

_Hydrology and Earth System Sciences, 2018_

## Referee Comment (RC1) · Anonymous Referee #1 · 27 Apr 2018

General comments The manuscript aims to demonstrate that a high resolution data-assimilation modelling framework allows improving soil moisture and runoff simulations at a continental scale. Thus, it addresses a question within the scopes of the journal. Aims of the work are overall clearly outlined and supported by references. I suggest to better justify the choices of models and datasets and temporal domain (2000-2006). Data-assimilation results are compared to open-loop simulations to quantitatively assess this improvement basing on root mean square error and mean bias error estimates with respect to CCI SM data. Overall results are well supported by figures and graphs. However, I would suggest the Authors to give a more detailed explanation for differences in overestimate and underestimate between the regions and between the

seasons (particularly in par. 3.1.1). It seems that this part has been deepened more for runoff than for soil moisture. Besides I would suggest the Authors to better explain if and how results can be affected by spatial resolution differences in the data. The advancements described in this study would benefit from a quantitative or qualitative comparison with other studies claiming the use of data assimilation for improving soil moisture and runoff simulations, to assess if the obtained results are satisfying. My recommendation is to accept the manuscript with minor review.

Specific comments Page 2 Line 32 and Page 6 Line 2: I would suggest to add spatial resolution in km, as done is other sections of the text Page 2 Lines 18-27: I think sentences here are a bit contradictory. I would suggest the Authors to better clarify what is commonly done in the state of the art and what is rarely done (and eventually why it is rarely done), in order to better highlight element of novelties of this work. Figure 1b: I would suggest to use a discretized legend as it represents different classes Page 4 Lines 2-3: the Authors state that "CLM3.5 was used in this study, instead of its most recent version, to keep the modelling framework consistent to Kurtz et al. (2016)." Would it be possible to hypothesize some advantages or disadvantages in using CLM most recent versions? Page 5 Line 9: I would suggest to briefly explain how this conversion is done or at least provide a reference? Page 6 Lines 12-15: Would it be possible to perform the inverse resampling (from $0.0275°$ to $0.25°$) and compare results with the one from $0.0275°$ to $0.25°$? The same could be said for runoff rates (paragraph 2.3.3) Page 7 Line 27: Is it possible that Authors refer to another figure? Page 10 Line 3: "CLM-DA reduces the runoff bias compared to CLM-OL". Evidence of this is in Figure 9, not in Figure 7. It would suggest the author to better clarify this point or remove it (as it is already stated in line 20) I suggest to add a table similar to Table 1 also for soil moisture.

Technical corrections Page 1 Line 31: Western et al., 2002 has to be changed with Western et al., 2004 according to references Page 2 Line 16: bracket missing after "...Clark et al., 2011)" Page 2 Line 25: López et al., 2016 is missing in the references

and there is a repetition of the name in the text Page 2 Line 28: Rains et al., 2017 is missing in the references Page 3 Line 28: remove comma before brackets Page 4 Line 9: remove the semicolon before bracket Page 4 Lines 12-27: I think this sentence is too long, thus I would split it into different sentences maybe one for each formula Page 5 Lines 30-31: Wahl et al. 2017 is missing in the references Page 6 Line 13: Jones, 1999 is missing in the references Page 6 Line 19: remove full stop after brackets Page 9 Line 28: replace Decker and Zeng, 2009 with Zeng and Decker, 2009, as for references Page 11 Lines 32-33: I think that Authors mean "it is preferable to account for additional model parameter uncertainties that shows a high sensitivity towards runoff" instead of "it is preferable to account for additional model parameter uncertainties towards runoff that shows a high sensitivity" Page 12 Line 12: Remove "This study showed that", as it is a repetition of the above line Page 12 Line 30: "The improvement in peak runoff could be OF particular importance in the management of extreme events such as flooding" Page 14 Line 7: reference missing in the text Page 15 Line 9: reference missing in the text Page 17 Line 18: reference missing in the text Page 19 Line 31: replace 998 with 1998 Page 20 Line 34: year of publication should be after doi, as for the other references Page 21 Line 8: reference missing in the text Figure 2c: better if months initials are in English

---

## Referee Comment (RC2) · Anonymous Referee #2 · 12 May 2018

**General:**

The submitted manuscript of Naz et al. (2018) evaluates the effect of soil moisture assimilation (namely ESA-CCI product) into the CLM 3.5 over Europe during 2000-2006. The Ensemble Kalman Filter is used for the model analysis, the observations are sampled using 100 randomly located points across the entire domain, while the remaining locations are used for independent evaluation. The CLM model operates at 3km spatial resolution, while the assimilation product is available at coarser, approx. 25km (0.25degree) resolution. Additionally, the gridded (monthly) runoff product available at 0.5 degree is used for evaluation of the model runs. Results are presented for the open loop (OL) and data assimilation (DA) runs. Results are presented in terms of the RMSE and relative bias per nine PRUDENCE regions. I find this topic relevant for HESS, however, according to my opinion, the manuscript is not suitable for publication in the present form:

1. As correctly stated in Section 3.3, before applying any DA methods, the modeler should better parameterize and constrain the model parameters, reduce systematic biases etc. I am afraid you cannot apply DA after seeing those strong biases in the open loop estimates (Figures 6 or 10) at all. I encourage authors to pay attention to proper model calibration before DA analysis.

2. This is not much of surprise when soil moisture gets assimilated into a model that model simulations at the analysis step get more close to the "observation-based" product, as much as the prescribed observation errors allow (given there exists spatial correlation between the 100 assimilated locations and remaining "withheld" observations). By assimilating the ESA-CCI product, the authors claim to improve the initial conditions of the model. That's all . . . I would welcome then the added value/implications of the improved initial model (wetness) conditions (e.g. with respect to some longer lead-times): OL vs. DA.

3. Additionally, using EnKF the authors modify the internal model states and thus introduce some numerical instability (against internal physical constraints for the model), which was not discussed at all. How do you handle this issue after the analysis step?

4. Hardly any discussion for (OL and DA) results is done with respect to other SM data assimilation/modeling studies over Europe . . . which could be used as a benchmark(?) Applying CLM over EU is indeed challenging, but there are other models already able to simulate SM and the choice of CLM is not well described either.

[Figure]

5. The authors have "high-resolution" in their title. I strongly encourage them to eliminate this from the title, especially if they use such coarse scale data to assimilate.

6. Why the authors did not use "high-resolution" discharge data for independent model evaluation? There are thousands of gauges with daily time step over Europe, if the routing would be enabled. I am afraid that using monthly gridded runoff is not sufficient for a "high-resolution" study.

7. The authors could have easily run their model at the resolution of the data and save their larger efforts in computer resources.

8. Another limitation of this study is the limited ensemble size. 12 members are way too low (this number is stated on p. 7, l. 23). Also, the ad-hoc construction of the perturbations needs better reasoning and clarifications!

9. The uncertainty in the time series figures is for the 12 ensemble members?

10. Numerous papers mentioned in the text are not included in the reference list!!!

Technical:
Spell-out ESA-CCI in the abstract.
p.1, line 14: remove "and the . . . due to"
p.5, line 28: remove "In their study"
p. 6: "this product" => which product you refer to here?
p. 6, line 19: missing space after parenthesis
p.10, l. 13: "UK" => "BI"
p.12, line 9: runoff => "monthly runoff"
Figs. 5 and 9, caption: "a,c" => "a,b"

---

## Referee Comment (RC3) · Anonymous Referee #3 · 17 May 2018

General comments:

The objective of this work is to assimilate ESA-CCI satellite-derived soil moisture estimates in CLM over Europe, from 2000 to 2006. The content of the paper is disappointing. The data assimilation experiment boils down to a sensitivity study illustrating possible model biases. Independent validation is missing. No justification is given for the choice of the 2000-2006 time period. Most recent satellites are missing in this period. It is not clear whether or not interannual variability of the vegetation is accounted for in the model. The authors do not use the scores usually used in hydrology. The improvement of the assimilation on water discharge is not convincing at all. The paper is poorly organized (no Discussion section). This work cannot be published in the present form.

Recommendation: reject.

Specific comments:

- P. 1, L. 21 (independent CCI-SSM observations): do you mean observations independent from CCI-SM? They should be!

- P. 3, L. 3: 10**0km?

- P. 4, L. 3 (CLM): how is vegetation represented in this version of the model?

- P. 5, L. 5 (LAI): does this mean that LAI for a given month is the same from one year to another? Given the marked impact of LAI on evapotranspiration, this might introduce marked soil moisture biases.

- P. 5, L. 20 (6 km): it is written in the Abstract and in Section 2.3.2 that the assimilation was made at a spatial resolution of 3 km. Why such a mismatch with the spatial resolution of atmospheric forcing?

- P. 7, L. 4: E-OBS was not defined before.

- P. 11, L. 13 (this study demonstrates): I am not convinced, there is no demonstration.

- P. 11, L. 32 (soil texture): Absolute CCI-SM values depend on pedotransfer functions and texture of the NOAH model. They are not "observed" and they should not be taken for granted. This is not a good way of doing data assimilation.

- P. 22, Table 1: what about other key hydrological scores such as KGE or NSC? Given what can be seen in Fig. 10, I doubt these scores are improved by the assimilation.

---

## Author Comment (AC1) · 22 Jun 2018

**Response to comments by Anonymous Referee #1**

General comments: The manuscript aims to demonstrate that a high resolution data-assimilation modelling framework allows improving soil moisture and runoff simulations at a continental scale. Thus, it addresses a question within the scopes of the journal. Aims of the work are overall clearly outlined and supported by references.

We would like to thank the anonymous reviewer for his/her comments and constructive suggestions, which we believe have led to an improved manuscript.

I suggest to better justify the choices of models and datasets and temporal domain (2000-2006).

RESPONSE: We selected CLM because it is one of the most complete land surface model at the moment and part of a large community effort (Community Earth System Model; http://www.cesm.ucar.edu/). It has been widely applied at continental and global scales to understand how land processes and anthropogenic impact on land states affect climate (e.g. Bonan et al., 2002; Dickinson et al., 2006). The CLM model parameterizes most of the land surface processes (such as infiltration, evaporation, surface runoff, subsurface drainage, canopy and snow processes) using the water and energy balance equations. In addition, CLM was designed for coupling with climate models and is also part of the fully coupled Terrestrial Systems Model Platform (TerrSysMP; Shrestha et al., 2014) that simulates the full terrestrial hydrologic cycle including feedbacks between atmosphere, land-surface and subsurface compartments of the water cycle. For upcoming studies, it is planned to use TerrSysMP including the parallel data assimilation framework (PDAF) to assess the impacts of satellite soil moisture assimilation on other water cycle variables across the soil-vegetation-atmosphere system and its effects on the accuracy of model simulations at the continental scale. Moreover, the CLM model can efficiently run for large model domains and at high spatial resolution. Since we performed our simulations at high spatial resolution and at continental scale, we selected the TerrSysMP -PDAF modeling framework which can be run on high performance computational infrastructure and can efficiently cope with the high computational burden of ensemble-based data assimilation framework.

With respect to the time span simulated, we used a high-resolution reanalysis system dataset (COSMO-REA6 from Hans-Ertel Centre for Weather Research (HErZ); Simmer et al., 2016), which is only now publicly available for a longer time period of 1995 – 2015. However, this recent dataset was not fully publicly available at the beginning of our study. We preferred to use this data over other datasets, because of its high spatial resolution in comparison to other commonly used forcing datasets such as E-OBS and ERA-Interim at 25 and 80km resolution, respectively. In addition, the COSMO-REA6 dataset was produced through the assimilation of observational meteorological data and showed good performance particularly for precipitation when compared to the observations (Wahl et al., 2017).

We now added additional motivation for the model selection, time period and forcings datasets in the revised manuscript.

Data-assimilation results are compared to open-loop simulations to quantitatively assess this improvement basing on root mean square error and mean bias error estimates with respect to CCI SM data. Overall results are well supported by figures and graphs. However, I would suggest the Authors to give a more detailed explanation for differences in overestimate and underestimate between the regions and between the seasons (particularly in par. 3.1.1). It seems that this part has been deepened more for runoff than for soil moisture.

RESPONSE: We included an extended discussion of the soil moisture differences with respect to other data assimilation studies over Europe. Please see our response below to the specific comment 2.

Besides I would suggest the Authors to better explain if and how results can be affected by spatial resolution differences in the data.

RESPONSE: See our response and Figure R1 below to the specific comment 6 on resolution and interpolation effects.

The advancements described in this study would benefit from a quantitative or qualitative comparison with other studies claiming the use of data assimilation for improving soil moisture and runoff simulations, to assess if the obtained results are satisfying. My recommendation is to accept the manuscript with minor review.

Specific comments

Page 2 Line 32 and Page 6 Line 2: I would suggest to add spatial resolution in km, as done is other sections of the text

RESPONSE: It has been modified in the revised manuscript.

Page 2 Lines 18-27: I think sentences here are a bit contradictory. I would suggest the Authors to better clarify what is commonly done in the state of the art and what is rarely done (and eventually why it is rarely done), in order to better highlight element of novelties of this work.

RESPONSE: We appreciate the suggestion. We added several references of other assimilation/modeling studies over Europe in the introduction section of the revised manuscript. For instance, many studies have explored the role of soil moisture assimilation in different modeling frameworks over Europe (e.g. Albergel et al., 2017; de Rosnay et al., 2013; Brocca et al., 2010; Draper et al., 2009; Ni-Meister et al., 2006). Albergel et al. (2017) applied a global land data assimilation system at 0.5° over Europe and Mediterranean domain to sequentially assimilate ESA CCI satellite-derived soil moisture data and leaf-area index product into the ISBA (Interactions between Soil,

Biosphere and Atmosphere) land surface model. They found more improvements in the surface soil moisture and particularly in the summer and autumn than in the winter and spring but found little improvements to the discharge when compared to the open loop (i.e. no assimilation) simulations.

In our study, we highlighted the added value of merging coarse resolution satellite observations through data assimilation with a land surface model to generate higher spatial resolution, downscaled estimates of the surface soil moisture profile with complete spatio-temporal coverage and with a higher accuracy than that of the open loop model estimates. To the best of our knowledge, this is the first study of its kind to provide a downscaled daily soil moisture product at 3km resolution over Europe

We discuss the added value of our data assimilation experiment in the introduction section of the revised manuscript to clarify the objectives of the study.

Figure 1b: I would suggest to use a discretized legend as it represents different classes

RESPONSE: We appreciate this comment. The figure has been modified in the revised manuscript.

Page 4 Lines 2-3: the Authors state that "CLM3.5 was used in this study, instead of its most recent version, to keep the modelling framework consistent to Kurtz et al. (2016)." Would it be possible to hypothesize some advantages or disadvantages in using CLM most recent versions?

RESPONSE: We added the following text to highlight the differences in different versions of the CLM model.

"The early version of CLM used a simplified bucket runoff model. While the saturated and unsaturated runoff assumptions based on the TOPMODEL concept were later introduced into the runoff scheme in CLM 2.0 (Bonan et al., 2002) and the Common Land Model (Dai et al., 2003), these models used a constant value to describe topography. In an ensuing step, a simplified topography representation, the SIMTOP scheme, was employed in CLM 3.5 (Niu et al., 2005). Previous studies showed that CLM 3.5 produces higher soil moisture and lower variability than observations in the root zone (e.g. Lawrence et al., 2011 and Niu et al., 2011). In order to reduce these biases, Li and Ma (2015) introduced a factor to describe soil porosity and increase recharge water from the soil column to the aquifer in the newer CLM versions (4.0 and 4.5) which offer improved solutions related to soil moisture and biogeochemical processes. However, Lawrence et al. 2011 showed that the differences between CLM3.5 and new versions of CLM with respect to soil moisture variability remain low when compared with observations (Lawrence et al. 2011). Based on assessment of the applicability of CLM 3.5 across the observation-based datasets in other studies (Li and Ma 2010; Li et al., 2011) and its coupling with current PDAF setup, we decided to use CLM 3.5 in this study."

Page 5 Line 9: I would suggest to briefly explain how this conversion is done or at least

provide a reference?

RESPONSE: We prescribed vegetation as the fractional coverage of different vegetation types according to the MODIS land cover dataset for the year 2001, which contains 21 land cover types defined by the International Geosphere-Biosphere Program (IGBP) (Friedl et al., 2002). This information is added in the revised manuscript.

Page 6 Lines 12-15: Would it be possible to perform the inverse resampling (from 0.0275° to 0.25°) and compare results with the one from 0.0275° to 0.25°? The same could be said for runoff rates (paragraph 2.3.3)

RESPONSE: We compared the original 0.25° against 0.0275° ESA CCI soil moisture. Only small differences between the two resolutions were visible particular for the time period of 2003 – 2006 (Fig. R1). However, we do see some differences in the first two years and the regions where the temporal coverage of the ESA CCI data is less than 30%. However, for the time period and regions with a good coverage of ESA CCI soil moisture data, we believe that our results are not significantly affected by the difference in the resolution. Similarly, for the model outputs, we don't see any significant difference (e.g. R-squared value in both cases is 0.56) in our results between 0.0275° and 0.25° when compared to the original CCI data at 0.25° resolution as shown in Fig. R2 as an example for one region.

[Figure]

Figure R1: Comparison of spatially averaged 2000 – 2006 daily ESA CCI soil moisture data between original 0.25° (black) and interpolated 0.0275° (red) over PRUDENCE regions.

[Figure]

Figure R2: Scatter plot showing the comparison of spatially averaged daily soil moisture between ESA CCI (0.25°) and CLM at 0.25° and 0.0275° resolution over the PRUDENCE region France.

We now discussed the effects of resolution and interpolation on the results in the discussion section of the revised manuscript.

Page 7 Line 27: Is it possible that Authors refer to another figure?

RESPONSE: This has been now corrected in the revised manuscript.

Page 10 Line 3: "CLM-DA reduces the runoff bias compared to CLM-OL". Evidence of this is in Figure 9, not in Figure 7. It would suggest the author to better clarify this point or remove it (as it is already stated in line 20) I suggest to add a table similar to Table 1 also for soil moisture.

RESPONSE: We appreciate this comment. We removed this sentence and added a table for soil moisture (similar to Table 1) in the revised manuscript.

Technical corrections

Page 1 Line 31: Western et al., 2002 has to be changed with Western et al., 2004 according to references

RESPONSE: We appreciate this comment. This has been now corrected in the revised manuscript.

Page 2 Line 16: bracket missing after ". . .Clark et al., 2011)"

RESPONSE: This has been corrected in the revised manuscript.

Page 2 Line 25: López et al., 2016 is missing in the references and there is a repetition of the name in the text

RESPONSE: We appreciate this comment. This has been corrected in the revised manuscript.

Page 2 Line 28: Rains et al., 2017 is missing in the references

RESPONSE: The missing reference has been added in the revised manuscript.

Page 3 Line 28: remove comma before brackets Page 4 Line 9: remove the semicolon before bracket

RESPONSE: This has been corrected in the revised manuscript.

Page 4 Lines 12-27: I think this sentence is too long, thus I would split it into different sentences maybe one for each formula

RESPONSE: This has been modified in the revised manuscript for clarity.

Page 5 Lines 30-31: Wahl et al. 2017 is missing in the references

RESPONSE: The missing reference has been added in the revised manuscript.

Page 6 Line 13: Jones, 1999 is missing in the references

RESPONSE: The missing reference has been added in the revised manuscript.

Page 6 Line 19: remove full stop after brackets

RESPONSE: This has been corrected in the revised manuscript.

Page 9 Line 28: replace Decker and Zeng, 2009 with Zeng and Decker, 2009, as for references

RESPONSE: This has been corrected in the revised manuscript.

Page 11 Lines 32-33: I think that Authors mean "it is preferable to account for additional model parameter uncertainties that shows a high sensitivity towards runoff" instead of "it is preferable to account for additional model parameter uncertainties towards runoff that shows a high sensitivity"

RESPONSE: We appreciate this comment. This has been modified in the revised manuscript for clarity.

Page 12 Line 12: Remove "This study showed that", as it is a repetition of the above line

RESPONSE: This has been modified in the revised manuscript for clarity.

Page 12 Line 30: "The improvement in peak runoff could be OF particular importance in the management of extreme events such as flooding"

RESPONSE: This has been modified in the revised manuscript for clarity.

Page 14 Line 7: reference missing in the text

RESPONSE: The missing reference has been corrected in the revised manuscript.

Page 15 Line 9: reference missing in the text

RESPONSE: The missing reference has been corrected in the revised manuscript.

Page 17 Line 18: reference missing in the text

RESPONSE: The missing reference has been corrected in the revised manuscript.

Page 19 Line 31: replace 998 with 1998

RESPONSE: We appreciate this comment. This has been corrected in the revised manuscript.

Page 20 Line 34: year of publication should be after doi, as for the other references

RESPONSE: This has been corrected in the revised manuscript.

Page 21 Line 8: reference missing in the text

RESPONSE: The missing reference has been corrected in the revised manuscript.

Figure 2c: better if months initials are in English

RESPONSE: This has been modified in the revised manuscript.

References:

Albergel, C., Munier, S., Leroux, D. J., Dewaele, H., Fairbairn, D., Barbu, A. L., Gelati, E., Dorigo, W., Faroux, S., Meurey, C., Le Moigne, P., Decharme, B., Mahfouf, J.-F., and Calvet, J.-C.: Sequential assimilation of satellite-derived vegetation and soil moisture products using SURFEX_v8.0: LDAS-Monde assessment over the Euro-Mediterranean area, Geosci. Model Dev., 10, 3889-3912, https://doi.org/10.5194/gmd-10-3889-2017, 2017.

Bonan, Gordon B., Keith W. Oleson, Mariana Vertenstein, Samuel Levis, Xubin Zeng, Yongjiu Dai, Robert E. Dickinson, and Zong-Liang Yang. "The land surface climatology

of the Community Land Model coupled to the NCAR Community Climate Model." Journal of climate 15, no. 22 (2002): 3123-3149.

Brocca, L., Melone, F., Moramarco, T., Wagner, W., Naeimi, V., Bartalis, Z., and Hasenauer, S.: Improving runoff prediction through the assimilation of the ASCAT soil moisture product, Hydrol. Earth Syst. Sci., 14, 1881-1893, https://doi.org/10.5194/hess-14-1881-2010, 2010.

Dai, Y., Zeng, X., Dickinson, R.E., Baker, I., Bonan, G.B., Bosilovich, M.G., Denning, A.S., Dirmeyer, P.A., Houser, P.R., Niu, G. and Oleson, K.W., 2003. The common land model. *Bulletin of the American Meteorological Society*, *84*(8), pp.1013-1023.
Niu, G.Y., Yang, Z.L., Dickinson, R.E. and Gulden, L.E., 2005. A simple TOPMODEL-based runoff parameterization (SIMTOP) for use in global climate models. *Journal of Geophysical Research: Atmospheres*, *110*(D21).

de Rosnay, P., Drusch, M., Vasiljevic, D., Balsamo, G., Albergel, C., and Isaksen, L.: A Simplified Extended Kalman Filter for the global operational soil moisture analysis at ECMWF, Q. J. Roy. Meteorol. Soc., 139, 1199–1213, doi:10.1002/qj.2023, 2013.

Dickinson, R.E., Oleson, K.W., Bonan, G., Hoffman, F., Thornton, P., Vertenstein, M., Yang, Z.L. and Zeng, X., 2006. The Community Land Model and its climate statistics as a component of the Community Climate System Model. *Journal of Climate*, *19*(11), pp.2302-2324.

Draper, C. S., J.-F. Mahfouf, and J. P. Walker (2009), An EKF assimilation of AMSR-E soil moisture into the ISBA landsurface scheme, J. Geophys. Res., 114, D20104, doi:10.1029/2008JD011650.

Friedl, M.A., McIver, D.K., Hodges, J.C., Zhang, X.Y., Muchoney, D., Strahler, A.H., Woodcock, C.E., Gopal, S., Schneider, A., Cooper, A. and Baccini, A., 2002. Global land cover mapping from MODIS: algorithms and early results. *Remote Sensing of Environment*, *83*(1-2), pp.287-302.

Lawrence, D.M., Oleson, K.W., Flanner, M.G., Thornton, P.E., Swenson, S.C., Lawrence, P.J., Zeng, X., Yang, Z.L., Levis, S., Sakaguchi, K. and Bonan, G.B., 2011. Parameterization improvements and functional and structural advances in version 4 of the Community Land Model. *Journal of Advances in Modeling Earth Systems*, *3*(1).

Li, M. and Ma, Z., 2015. Soil moisture drought detection and multi-temporal variability across China. *Science China Earth Sciences*, *58*(10), pp.1798-1813.

Li, M., Ma, Z. and Du, J., 2010. Regional soil moisture simulation for Shaanxi Province using SWAT model validation and trend analysis. *Science China Earth Sciences*, *53*(4), pp.575-590.

Li, H., Huang, M., Wigmosta, M.S., Ke, Y., Coleman, A.M., Leung, L.R., Wang, A. and Ricciuto, D.M., 2011. Evaluating runoff simulations from the Community Land Model

4.0 using observations from flux towers and a mountainous watershed. *Journal of Geophysical Research: Atmospheres*, *116*(D24).

Ni-Meister, W., P. R. Houser, and J. P. Walker (2006), Soil moisture initialization for climate prediction: Assimilation of scanning multifrequency microwave radiometer soil moisture data into a land surface model, J. Geophys. Res., 111, D20102, doi:10.1029/2006JD007190.

Niu, G.Y., Yang, Z.L., Mitchell, K.E., Chen, F., Ek, M.B., Barlage, M., Kumar, A., Manning, K., Niyogi, D., Rosero, E. and Tewari, M., 2011. The community Noah land surface model with multiparameterization options (Noah-MP): 1. Model description and evaluation with local-scale measurements. *Journal of Geophysical Research: Atmospheres*, *116*(D12).

Shrestha, P., Sulis, M., Masbou, M., Kollet, S. and Simmer, C., 2014. A scale-consistent terrestrial systems modeling platform based on COSMO, CLM, and ParFlow. *Monthly weather review*, *142*(9), pp.3466-3483.

Simmer, C., Adrian, G., Jones, S., Wirth, V., Göber, M., Hohenegger, C., Janjic, T., Keller, J., Ohlwein, C., Seifert, A. and Trömel, S., 2016. Herz: The german hans-ertel centre for weather research. *Bulletin of the American Meteorological Society*, *97*(6), pp.1057-1068.

Wahl, S., Bollmeyer, C., Crewell, S., Figura, C., Friederichs, P., Hense, A., Keller, J.D. and Ohlwein, C., 2017. A novel convective-scale regional reanalyses COSMO-REA2: Improving the representation of precipitation. *Meteorol. Z.*

---

## Author Comment (AC2) · 22 Jun 2018

**Response to comments by Anonymous Referee #2**

**General:**

The submitted manuscript of Naz et al. (2018) evaluates the effect of soil moisture assimilation (namely ESA-CCI product) into the CLM 3.5 over Europe during 2000-2006. The Ensemble Kalman Filter is used for the model analysis, the observations are sampled using 100 randomly located points across the entire domain, while the remaining locations are used for independent evaluation. The CLM model operates at 3km spatial resolution, while the assimilation product is available at coarser, approx. 25km (0.25degree) resolution. Additionally, the gridded (monthly) runoff product available at 0.5 degree is used for evaluation of the model runs. Results are presented for the open loop (OL) and data assimilation (DA) runs. Results are presented in terms of the RMSE and relative bias per nine PRUDENCE regions. I find this topic relevant for HESS, however, according to my opinion, the manuscript is not suitable for publication in the present form:

We would like to thank the anonymous reviewer for his/her comments and constructive suggestions, which we believe have led to an improved manuscript.

1. As correctly stated in Section 3.3, before applying any DA methods, the modeler should better parameterize and constrain the model parameters, reduce systematic biases etc. I am afraid you cannot apply DA after seeing those strong biases in the open loop estimates (Figures 6 or 10) at all. I encourage authors to pay attention to proper model calibration before DA analysis.

RESPONSE: Hydrologic states and fluxes simulated by Land Surface Models (LSM) are often biased due to, e.g., systematic biases in input forcings, uncertainty of input parameters, the use of parameterizations and uncertainty related to the (bio)physical processes representation in the models (e.g Yin et al., 2014; Han et al., 2014). Traditionally, the calibration of model parameters is performed using in situ data to resolve these biases. However, such in situ data – soil moisture in particular – are often unavailable at a large scale.  In this study, we have utilized the remotely sensed soil moisture for evaluating the performance of DA in reducing the model biases as previous studies have shown  (e.g. Moradkhani et al., 2005; Liu and Gupta, 2007; Kumar et al., 2008; Nie et al., 2011; Chen et al., 2013; Lievens et al., 2016; Lannoy and Reichle, 2016). Another motivation of this study is not only to improve the soil moisture through data assimilation approach, but also to evaluate the assimilations impacts on other land surface variables such as runoff. Model calibration prior to assimilation, e.g., with the runoff data set of Gudmundsson and Seneviratne (2016) would violate the independency criterion of calibration and validation data for the impact assessment of the data assimilation procedure.

Furthermore, in land surface modeling systematic differences between the model climatology and the observation data climatology are traditionally corrected before

assimilation, to ensure that data assimilation is applied under conditions of no systematic bias. However, different procedures to correct bias are used, like the estimation of a single constant bias value, seasonal dependent bias or CDF-matching (e.g. Reichle and Koster, 2004 and Drusch et al., 2005). The procedure has some important limitations: (i) the bias is only partially corrected or over-corrected; (ii) the bias in the DA-procedure is not assigned to the model or measurement data, but after the assimilation it is implicitly assumed that the systematic bias is related to the bias in the measurements (model states are not corrected for a systematic bias). We did therefore not perform any bias correction of the soil moisture observation by rescaling of the observations to model climatology. A further argument for not following this approach was that spatial patterns could be altered and thereby some of the independent information provided by the satellite may be removed; it is desirable to retain as much of the independent satellite information as possible.

The alternative procedure we followed here was to neglect systematic biases, but assimilate with a sufficient ensemble spread so that observations allow correcting model predictions. This approach results in a stronger corrective effect of measurement data. A further alternative is to attribute systematic biases to erroneous model parameter values, which is one of the main sources of error and uncertainty in land surface model predictions. We explore this option now in the revised version of the manuscript using a joint state-parameter estimation approach. Although this approach has also important limitations, related to the fact that we do not know well enough the relative importance of systematic model errors and systematic errors in the measurement data, an advantage is that we correct for possible systematic model bias by modifying soil texture parameters.

Therefore, we evaluated now the impact of soil texture properties like the percentages of clay and sand on the CLM model performance and jointly estimated model states and parameters in the data assimilation experiment. We plotted the results as mean monthly soil moisture for the 2000 – 2006 time period for CLM-OL, CLM-DA (only state update) and CLM-DA (joint state and parameter updates) as shown in Fig. R1. We see that in both cases (state updating alone, joint state-parameter updating) the DA-runs are quite close to the observed values. However, parameter updating is supposed to have corrected part of the systematic model bias and other fluxes like evapotranspiration and also discharge can show larger modifications as response to the different parameters.

[Figure]

Figure R1: Multi-ensemble spatially averaged mean monthly soil water content (2000 – 2006) simulated with CLM-OL (no assimilation; gray colour) and CLM-DA (with data assimilation) and compared with CCI-SM data over the PRUDENCE regions. The blue colour indicates CLM-DA-S (state updates only) and red colour indicates CLM-DA-SP (joint state-parameter update).

We have now discussed above points in the revised version of the paper and also include information from additional experiments for joint state-parameter updating in the revised manuscript.

2. This is not much of surprise when soil moisture gets assimilated into a model that model simulations at the analysis step get more close to the "observation- based" product, as much as the prescribed observation errors allow (given there exists spatial correlation between the 100 assimilated locations and remaining "withheld" observations). By assimilating the ESA-CCI product, the authors claim to improve the initial conditions of the model. That's all . . . I would welcome then the added value/implications of the improved initial model (wetness) conditions (e.g. with respect to some longer lead-times): OL vs. DA.

RESPONSE: Most land data assimilation studies only aim at improving the model initial conditions. These improved model initial conditions would impact other modeled fluxes like discharge (which is a focus of this paper), but also latent and sensible heat fluxes between the land and atmosphere, which could potentially improve weather predictions. The added value of our study is to apply a data assimilation modeling framework over Europe along the complete simulation time span to derive a longer-term and high spatial resolution land surface data assimilation product in order to increase monitoring accuracy for land surface moisture and water states and fluxes.

Beyond that, a continuous DA approach allows to monitor long-term changes of the

terrestrial water cycle and enhance our understanding of the role of land surface processes on, e.g. atmospheric boundary layer dynamics. Therefore, high quality soil moisture product is needed. Since point scale measurements from in situ sensors are only available at a limited number of locations, remotely sensed soil moisture products are promising for a more large scale improvement of soil moisture variations. However, sparse data coverage in satellite observations still limits their ability to provide spatially and temporally consistent time series of water balance estimates. To overcome this limitation, in our study, we used data assimilation to merge coarse resolution satellite observations with a land surface model to generate higher horizontal resolution i.e. downscaled estimates of the full soil moisture profile with complete spatio-temporal coverage.

We now discussed the added value of our data assimilation experiment in the introduction and conclusions section of the revised manuscript.

3. Additionally, using EnKF the authors modify the internal model states and thus introduce some numerical instability (against internal physical constraints for the model), which was not discussed at all. How do you handle this issue after the analysis step?

RESPONSE: In the DA scheme we ensure that updated states (soil moisture) are kept in reasonable physical ranges (between residual soil moisture and porosity) to yield physically consistent estimates of fluxes and maintained the water budget. Additionally, numerical stability is not significant compared to, e.g., groundwater models/ atmospheric models where PDE's are solved with iterative methods. In CLM (and generally in land surface models) the equations/algorithms for deriving mass/energy budgets/transfer are simpler and more robust (numerically) than for full PDE-based systems.

We have now discussed these points in the revised manuscript.

4. Hardly any discussion for (OL and DA) results is done with respect to other SM data assimilation/modeling studies over Europe ... which could be used as a benchmark(?)

RESPONSE: We appreciate the suggestion. We added several references of other assimilation/modeling studies over Europe in the introduction section of the revised manuscript. For instance, many studies have explored the role of soil moisture assimilation in different modeling frameworks over Europe (e.g. Albergel et al., 2017; de Rosnay et al., 2013; Brocca et al., 2010; Draper et al., 2009; Ni-Meister et al., 2006). Albergel et al. (2017) applied a global land data assimilation system at 0.5° over the Europe and Mediterranean domain to sequentially assimilate ESA CCI satellite-derived soil moisture data and leaf-area index product into the ISBA (Interactions between Soil, Biosphere and Atmosphere) land surface model. They found more improvements in the surface soil moisture and particularly in the summer and autumn than in the winter and spring but found little improvements to the discharge when compared to the open loop

(i.e. no assimilation) simulations.

Applying CLM over EU is indeed challenging, but there are other models already able to simulate SM and the choice of CLM is not well described either.

(Below is the same response to Reviewer 1, comment 1)

RESPONSE: We selected CLM because it is one of the most complete land surface model at the moment and part of a large community effort (Community Earth System Model; http://www.cesm.ucar.edu/). It has been widely applied at continental and global scales to understand how land processes and anthropogenic impact on land states affect climate (e.g. Bonan et al., 2002; Dickinson et al., 2006). The CLM model parameterizes most of the land surface processes (such as infiltration, evaporation, surface runoff, subsurface drainage, canopy and snow processes) using the water and energy balance equations. In addition, CLM was designed for coupling with climate models and is also part of the fully coupled Terrestrial Systems Model Platform (TerrSysMP; Shrestha et al., 2014) that simulates the full terrestrial hydrologic cycle including feedbacks between atmosphere, land-surface and subsurface compartments of the water cycle. For upcoming studies, it is planned to use TerrSysMP including the parallel data assimilation framework (PDAF) to assess the impacts of satellite soil moisture assimilation on other water cycle variables across the soil-vegetation-atmosphere system and its effects on the accuracy of model simulations at the continental scale can be explored. Moreover, the CLM model can efficiently run for large model domains and at high spatial resolution. Since we performed our simulations at high spatial resolution at continental scale, we selected the TerrSysMP -PDAF modeling framework which can be run on high performance computational infrastructure and can efficiently cope with the high computational burden of ensemble-based data assimilation framework.

The authors have "high-resolution" in their title. I strongly encourage them to eliminate this from the title, especially if they use such coarse scale data to as-similate.

RESPONSE: As discussed in the response to comment 2, in this study we used data assimilation approach to highlight the added value of merging coarse resolution satellite observations with a land surface model to generate higher spatial resolution, downscaled estimates of the surface soil moisture profile with complete spatio-temporal coverage and with a higher accuracy than that of the open loop model estimates. Many applications (such as drought, flood, irrigation management) require observations of the complete soil moisture profile and with finer spatial and temporal resolutions than those of remotely sensed products. To the best of our knowledge, this is the first study of its kind to provide a downscaled daily soil moisture product at 3km resolution over Europe. In order to accommodate the reviewer comment, we replaced "high resolution" in the title with "3km".

5. Why the authors did not use "high-resolution" discharge data for independent model evaluation? There are thousands of gauges with daily time step over Europe, if the routing would be enabled. I am afraid that using monthly gridded runoff is not sufficient for a "high-resolution" study.

RESPONSE: We agree that discharge data from many gauging stations are available and can be used for independent evaluation. However, in the current version of the CLM model, the routing scheme is based on simple linear storage outflow relationships, in which a prescribed channel velocity field without temporal variability is used. This simplified assumption can lead to an offset even in the monthly peak streamflow in large catchments (Li et al., 2011). In addition, in the CLM 3.5, the river routing module is implemented at 0.5° where the discretization of river routing elements is based on a grid method in which the grid for river routing is independent of the grid for runoff simulation. Therefore, a coarse spatial resolution, such as 0.5° can lead to unrealistic flow accumulation paths and cannot be used to evaluate discharge at small catchments. Due to the aforementioned points, adequate validation of the results is therefore not possible.

In this study, we instead used the E-RUN runoff product which combines observed river flow with gridded estimates of precipitation and temperature using machine learning. Therefore, this gridded runoff dataset is solely derived from observations and does not rely on any modeling assumptions. We believe that using an observation-based non-routed gridded runoff product has distinct advantage to evaluate the impact of soil moisture assimilation on runoff at every grid cell within a spatial domain. Using gridded runoff is also useful to evaluate model structure errors in representation of runoff generation in the model. In the land surface models such as CLM, the representation of runoff is often simplistic and conceptual and many previous studies have shown that performance of the CLM model in simulating hydrological processes varies based on regions. This might be attributed to the fact that assumptions to estimate surface and subsurface runoff in the model might be valid in some regions but not in other regions (e.g. humid vs. dry regions). We also noted similar behavior of CLM in our study where the assimilation of soil moisture helps to improve runoff in some regions but degraded in other regions.

In the CLM 3.5 runoff scheme, runoff is partitioned into surface flow and subsurface flow and basic simulation element of runoff is the grid cell. In the revised version of the manuscript, the performances of the surface flow and base flow are evaluated separately in order to identify the dominant factor in total runoff generation as a result of soil moisture assimilation.

6. The authors could have easily run their model at the resolution of the data and save their larger efforts in computer resources.

RESPONSE: The goal and added value of the study was to produce a high-resolution, downscaled land surface hydrology over Europe. It is true that (a large number of) 1D soil moisture DA-experiments could have been conducted at the measurement locations (where we assimilated ESA CCI soil moisture in this experiment), but an essential component of this work is that we updated soil moisture contents at other locations (at the European scale), based on spatial correlations, and investigated whether soil moisture characterization between measurement locations could also be improved, and if this improves also runoff characterization at the European scale. As a conclusion, we respectfully disagree with this point of the reviewer.

It should also be noted that we made an extensive effort to collect and organize high resolution land surface data and atmospheric forcing datasets to implement the CLM model at 3-km resolution over Europe. Particularly, the COSMO-REA6 is available at a high 6 km spatial resolution. The organization of COSMO-REA6 hourly record for the entire Europe is not a trivial task. The applicability of COSMO-REA6 for a land surface model simulation over the EURO-CORDEX domain has never been shown previously.

In addition to forcings, this study also uses high-resolution land surface information in order to better represent the effects of land surface heterogeneity and provide climate information at the scales needed for impact assessment. In the current version of the CLM model, the officially released land surface datasets are provided at 0.5° by 0.5° or coarser resolutions. For example, in our study, PFTs fractional cover data were derived using high resolution MODIS data. Such a data-intensive effort is unprecedented in the previous studies, and hence the new resource will be valuable because it will prompt future hydro-climate studies.

7. Another limitation of this study is the limited ensemble size. 12 members are way too low (this number is stated on p. 7, l. 23). Also, the ad-hoc construction of the perturbations needs better reasoning and clarifications!

RESPONSE: While we agree with this comment, the number of ensemble member, however, was set to 12 members in our study, due to the large number of grid cells and required computational resources. From previous literatures (e.g., Kumar et al., 2008; Pan and Wood, 2010; De Lannoy et al., 2012; Yin et al., 2015), it is clear that the performance of EnKF relies on the ensemble size. For example, (Kumar et al., 2008; Yin et al., 2015) indicated that when the ensemble size is close to 12, it may lead to efficient DA updating process, while (Pan and Wood, 2010; De Lannoy et al., 2012) suggested 20 ensemble members.

We now evaluated the impact of the number of ensembles numbers on the performance of EnKF by increasing the ensemble size to 20 and run the model for one test year (i.e. 2006). We plotted the results as an ensemble mean of spatially averaged daily soil moisture for the year 2006 for CLM-DA (12 vs. 20 ensemble members) and compared with the daily ESA CCI soil moisture values over PRUDENCE regions as shown in Fig. R2.

[Figure]

Figure R2: Spatially averaged ensemble daily soil water content (SWC) simulated with CLM-DA (ensemble mean of 12 and 20 ensemble members) and compared with CCI-SM data for year 2006 over the PRUDENCE regions. The R2 values in each panel are r-squared calculated for both CLM-DA with 12 and 20 ensemble members.

While, we see some improvements in the simulated soil moisture as results of using 20 ensemble members, particularly in the winter season and in some regions such as, SC, Al and EA when compared to observations, in both cases the simulated soil moisture from the DA-runs with 12 and 20 ensemble members are quite close to the observed values. We will now include this sensitivity analysis in the discussion section of revised manuscript.

In addition, please note that using an increased number of ensemble members is a big challenge for large-scale high-resolution model because of needed computation memory and storage, and to a lesser degree also because of the computational burden. One year of model run with 20 ensemble members requires 680GB of computer storage per output variable (i.e. equivalent to 5TB of storage for 7 years of simulations per variable at daily time scale) and has resulted in the use of 76,800 CPU core-hours (compare to 46,000 core-hours with 12 ensemble members).

We clearly noted the limitations of our study in the manuscript. In future, with improved availability of computing resources, larger ensemble sizes will be possible.

8. The uncertainty in the time series figures is for the 12 ensemble members?

Correct. We included this information in the figure captions.

9. Numerous papers mentioned in the text are not included in the reference list!!!

Thanks you for pointing this out. We added missing references in the reference list.

Technical:

Spell-out ESA-CCI in the abstract.

This modification has been made

p.1,line 14: remove "and the . . . due to"

This modification has been made

p.5, line 28: remove "In their study"

This modification has been made.

p. 6: "this product" => which product you refer to here?

We referred to gridded runoff product E-RUN version 1.1 (Gudmundsson and Seneviratne, 2016). We modified text for clarity.

p. 6, line 19: missing space after parenthesis

We removed the space after parenthesis.

p.10, l. 13: "UK" => "BI"

This modification has been made

p.12, line 9: runoff => "monthly runoff"

This modification has been made.

Figs. 5 and 9, caption: "a,c" => "a,b"

The figure caption has been modified for clarity.

---

## Author Comment (AC3) · 22 Jun 2018

**Response to comments by Anonymous Referee #3**

**General comments:**
The objective of this work is to assimilate ESA-CCI satellite-derived soil moisture estimates in CLM over Europe, from 2000 to 2006. The content of the paper is disappointing. The data assimilation experiment boils down to a sensitivity study illustrating possible model biases. Independent validation is missing. No justification is given for the choice of the 2000-2006 time period. Most recent satellites are missing in this period. It is not clear whether or not interannual variability of the vegetation is accounted for in the model. The authors do not use the scores usually used in hydrology. The improvement of the assimilation on water discharge is not convincing at all. The paper is poorly organized (no Discussion section). This work cannot be published in the present form.
Recommendation: reject.

RESPONSE: We thank the reviewer for his/her efforts. Unfortunately, some of the statements above are not correct. The DA-experiments do not boil down to a sensitivity study, there is independent validation in our opinion and we use scores which are traditionally used in land surface DA studies (it is true that we do not use scores traditionally used in conceptual rainfall-runoff modeling studies). Other issues which are highlighted above have been corrected for in the revised version of the manuscript.

The data assimilation experiment boils down to a sensitivity study illustrating possible model biases.
RESPONSE: In this study we used data assimilation approach to merge coarse resolution satellite observations with a land surface model, to generate higher resolution, downscaled estimates of the surface soil moisture profile with complete spatio-temporal coverage. The added value of our study is to apply a data assimilation modeling framework over Europe to derive a longer-term and high spatial resolution land surface data product in order to increase monitoring accuracy for land surface soil moisture and water states and fluxes.

We now discuss the added value of our data assimilation experiment in the introduction section of the revised manuscript to make our objectives clearer to the readers.

Independent validation is missing.
RESPONSE: We do not agree with this comment. Please note that we randomly selected 100 locations to assimilate the ESA CCI data into the CLM model, while the remaining data was used for independent validation. Additionally, another independent gridded observation-based runoff product was used to evaluate the performance of the soil moisture assimilation in the CLM model in improving indirectly additional hydrological variables such as runoff.

No justification is given for the choice of the 2000-2006 time period. Most recent satellites are missing in this period.

RESPONSE: In this study we used a high-resolution reanalysis system COSMO-REA6 from Hans-Ertel Centre for Weather Research (HErZ; Simmer et al., 2016) dataset, which is only now publicly available for longer time period of 1995 – 2015. However, this recent dataset was not fully publicly available at the beginning of our study. We preferred to use this data over other datasets, because of its high spatial resolution in comparison to other commonly used forcing datasets such as E-OBS and ERA-Interim at 25 and 80km resolution, respectively. In addition, the COSMO-REA6 dataset was produced through the assimilation of observational meteorological data and showed good performance particularly for precipitation when compared to observations (Wahl et al., 2017).

We agree that recent satellites such as the Soil Moisture and Ocean Salinity (SMOS) Mission, Soil Moisture Active Passive (SMAP) Mission could be another option, however, we selected the ESA CCI soil moisture product for assimilation because of its availability at longer time scales. This provides the opportunity to assess the potential impact of longer-term soil moisture observations on hydrologic simulations for climate change studies.

We clarified these points in the revised manuscript.

It is not clear whether or not interannual variability of the vegetation is accounted for in the model.

RESPONSE: In the current study we do not account for interannual variability of the vegetation. Instead we prescribed vegetation as the fractional coverage of different vegetation types according to MODIS land cover dataset for year 2001, which contains 21 land cover types defined by the International Geosphere-Biosphere Program (IGBP) (Friedl et al., 2002). This information is now added in the revised manuscript.
In the revised version of the manuscript, we will replace the CLM default PFT-specific annual LAI-cycles with prescribed LAI from MODIS to consider the inter-annual variability of the vegetation.

The authors do not use the scores usually used in hydrology.
RESPONSE: The main objective of the study is to evaluate the performance of data assimilation of remotely sensed data into a land surface model in simulating surface soil moisture. We evaluate the performance in terms of RMSE and absolute bias, which is a common practice in most land surface data assimilation studies. To address the reviewer's concern, we now evaluated the performance of the model in simulating daily soil moisture and monthly runoff for 2000 – 2006 time period over PRUDENCE regions using suggested scores as shown below in Table R1.

Table R1: Calculated hydrologic scores to evaluate model performance of daily soil moisture and monthly runoff simulated by CLM-OL and CLM-DA for PRUDENCE regions.

| | BI | IP | FR | ME | SC | AL | MD | EA | EU |
|---|---|---|---|---|---|---|---|---|---|
| **Soil Moisture** | | | | | | | | | |
| **CLM-OL** | | | | | | | | | |
| NSE | -4.56 | -0.17 | -1.49 | -1.52 | -1.66 | -1.71 | -1.32 | -1.81 | -1.96 |
| KGE | 0.30 | 0.70 | 0.35 | 0.25 | 0.02 | 0.26 | 0.69 | 0.16 | 0.27 |
| RMSE* | 0.06 | 0.05 | 0.07 | 0.07 | 0.05 | 0.06 | 0.06 | 0.08 | 0.05 |
| %BIAS | 23.80 | 24.50 | 27.40 | 24.90 | 18.60 | 21.90 | 27.50 | 32.10 | 25.00 |
| **CLM-DA** | | | | | | | | | |
| NSE | -0.14 | 0.60 | 0.27 | 0.16 | -0.29 | 0.19 | 0.51 | 0.08 | 0.15 |
| KGE | 0.60 | 0.71 | 0.71 | 0.63 | 0.23 | 0.61 | 0.61 | 0.56 | 0.59 |
| RMSE | 0.03 | 0.03 | 0.04 | 0.04 | 0.04 | 0.04 | 0.03 | 0.04 | 0.03 |
| %BIAS | 5.80 | -3.00 | 7.90 | 6.90 | 6.30 | 6.80 | 5.40 | 9.20 | 5.40 |
| **Runoff** | | | | | | | | | |
| **CLM-OL** | | | | | | | | | |
| NSE | -3.7 | -0.3 | -9.5 | -10.2 | -19.3 | -18.3 | -9.2 | -102.6 | -47.4 |
| KGE | -0.4 | 0.0 | -1.8 | -1.3 | -1.9 | -1.7 | -1.6 | -5.9 | -2.9 |
| RMSE | 1.2 | 0.4 | 1.1 | 0.8 | 1.4 | 1.7 | 0.9 | 1.1 | 0.9 |
| %BIAS | 115.9 | 67.9 | 221.3 | 163.3 | 182.5 | 149.2 | 192.8 | 359.6 | 186.8 |
| **CLM-DA** | | | | | | | | | |
| NSE | -1.46 | -0.64 | -0.55 | -0.35 | -0.65 | -1.91 | -0.52 | -0.63 | -1.66 |
| KGE | -0.09 | -0.33 | -0.11 | 0.33 | 0.26 | 0.20 | 0.17 | 0.13 | 0.17 |
| RMSE | 0.85 | 0.41 | 0.42 | 0.29 | 0.40 | 0.66 | 0.34 | 0.14 | 0.20 |
| %BIAS | -76.50 | -81.10 | -69.70 | -51.30 | -8.30 | -47.90 | -49.70 | -12.10 | -33.30 |

* units for RMSE are mm³/mm³ and mm/day for soil moisture and runoff, respectively.

While we see the value of including more scores for evaluating model performance, we note that including other scores does not change our conclusions. With respect to NSE and KGE, we clearly see a significant improvement in simulated soil moisture for CLM-DA in comparison to CLM-OL over all PRUDENCE analysis regions, except for region Scandinavia. For runoff, overall we see relatively less improvements in terms of NSE and KGE. In the land surface models such as CLM, the representation of runoff is often simplistic and conceptual and many previous studies have shown that performance of CLM model in simulating hydrological processes varies based on regions. This might be attributed to the fact that assumptions to estimate surface and subsurface runoff in the model might be valid in some regions but not in other regions (e.g. humid vs. dry regions). We also noted similar behavior of CLM in our study where the assimilation of soil moisture helps to improve runoff in some regions but degraded in other regions. Our results agree well with other data assimilation studies (e.g. Albergel et al., 2017; Parajka

et al., 2006; Crow et al., 2006) which showed that assimilation is more effective in modifying the surface soil moisture but found little improvements to the discharge as a result of the remotely sensed soil moisture assimilations. This also highlights the need to jointly use soil moisture and discharge observations to improve global and continental hydrological and/or rainfall-runoff models, but this is beyond the scope of the current manuscript.

The paper is poorly organized (no Discussion section).

RESPONSE: A discussion section is added in the revised manuscript.

Specific comments:
- P. 1, L. 21 (independent CCI-SSM observations): do you mean observations independent from CCI-SM? They should be!
RESPONSE: In this study, we randomly selected 100 locations to assimilate the ESA CCI data into the CLM model, while the remaining data was used for independent validation. In the manuscript we state "The soil moisture validation of the CLM-DA and CLM-OL simulations used all the available CCI-SM data in the time period of 2000 to 2006. This approach also allowed us to independently cross-validate the SM values over grid cells that were not used in the data assimilation."

- P. 3, L. 3: 10**0km?
RESPONSE: $10^0$km

- P. 4, L. 3 (CLM): how is vegetation represented in this version of the model?
RESPONSE: We prescribed vegetation as the fractional coverage of different vegetation types according to MODIS land cover dataset for year 2001, which contains 21 land cover types defined by the International Geosphere-Biosphere Program (IGBP) (Friedl et al., 2002). For each land cover, average monthly leaf and stem area index (LAI and SAI) values are based on the global CLM3.5 surface dataset, which was created using the multiyear MODIS land surface data products (Oleson et al., 2008). This is now explained in the revised manuscript.

- P. 5, L. 5 (LAI): does this mean that LAI for a given month is the same from one year to another? Given the marked impact of LAI on evapotranspiration, this might introduce marked soil moisture biases.
RESPONSE: As mentioned in the previous comment, in the current modeling setup we prescribed static vegetation cover as the fractional coverage of different land cover type and for each type, we specify the average monthly LAI values based on global CLM 3.5 surface dataset (Oleson et al., 2008). In the revised manuscript, we will replace the CLM default PFT-specific annual LAI-cycles with prescribed LAI from MODIS to account for the inter-annual variability of the vegetation.

- P. 5, L. 20 (6 km): it is written in the Abstract and in Section 2.3.2 that the assimilation was made at a spatial resolution of 3 km. Why such a mismatch with the spatial resolution of atmospheric forcing?
RESPONSE: In our study, we used the high-resolution reanalysis system COSMO-REA6 from Hans-Ertel Centre for Weather Research (HErZ; Simmer et al., 2016) dataset, which is available at 6km resolution. However, we performed model simulations over the EURO-CORDEX domain at 3km resolution, which is inscribed into the official EUR-11 grid at 0.11° spatial resolution. To match the model resolution, the 6km COSMO-REA6 was interpolated to 3km resolution.

- P. 7, L. 4: E-OBS was not defined before.
RESPONSE: Thanks you for pointing this out. We have now defined the E-OBS.

- P. 11, L. 13 (this study demonstrates): I am not convinced, there is no demonstration.

RESPONSE: The original statement has been revised for clarity.

- P. 11, L. 32 (soil texture): Absolute CCI-SM values depend on pedotransfer functions and texture of the NOAH model. They are not "observed" and they should not be taken for granted. This is not a good way of doing data assimilation.

RESPONSE: We can't completely agree with this comment. According to Dorigo et al., 2017, "the European Space Agency CCI soil moisture product is the first multi-decadal, global satellite-observed soil moisture (SM) dataset as part of its Climate Change Initiative (CCI) program. This product, named ESA CCI SM, combines various single-sensor active and passive microwave soil moisture products into three harmonised products: a merged ACTIVE, a merged PASSIVE, and a COMBINED active + passive microwave product." It is true that the soil moisture values in the passive microwave product is not entirely model-independent, for the combined product (i.e. active + passive microwave product), the systematic differences between ACTIVE and PASSIVE are corrected by matching the CDF of each pixel against long-term LSM-based soil moisture, which is provided by GLDAS-Noah.
Several authors (e.g. Albergel et al., 2013, 2017; Dorigo et al., 2017; McNally et al., 2016; Wagner et al., 2012) have highlighted the quality and stability of the product. For example Albergel et al., 2017 assimilated the CCI soil moisture data into the land surface model over the Euro-Mediterranean region for the time period of 2000 – 2012.
In addition, in this study we use an observation uncertainty during assimilation, as stated already in the manuscript: "In this study, we assumed a spatially uniform observational error for CCI-SM (i.e. 0.02 mm3/mm3) in the CLM-PDAF setup." Therefore, the experimental design follows a usual and adequate way of doing data assimilation.

- P. 22, Table 1: what about other key hydrological scores such as KGE or NSC? Given what can be seen in Fig. 10, I doubt these scores are improved by the assimilation.

RESPONSE: See our previous response and Table R1 above to the comment on hydrological scores.

**References:**

Albergel, C., Dorigo, W., Balsamo, G., Muñoz-Sabater, J., de Rosnay, P., Isaksen, L., Brocca, L., de Jeu, R. and Wagner, W., 2013. Monitoring multi-decadal satellite earth observation of soil moisture products through land surface reanalyses. Remote Sensing of Environment, 138, pp.77-89.

Albergel, C., Munier, S., Leroux, D. J., Dewaele, H., Fairbairn, D., Barbu, A. L., Gelati, E., Dorigo, W., Faroux, S., Meurey, C., Le Moigne, P., Decharme, B., Mahfouf, J.-F., and Calvet, J.-C.: Sequential assimilation of satellite-derived vegetation and soil moisture products using SURFEX_v8.0: LDAS-Monde assessment over the Euro-Mediterranean area, Geosci. Model Dev., 10, 3889-3912, https://doi.org/10.5194/gmd-10-3889-2017, 2017.

Crow, W.T., Bindlish, R. and Jackson, T.J., 2005. The added value of spaceborne passive microwave soil moisture retrievals for forecasting rainfall-runoff partitioning. Geophysical Research Letters, 32(18).

Dorigo, W., Wagner, W., Albergel, C., Albrecht, F., Balsamo, G., Brocca, L., Chung, D., Ertl, M., Forkel, M., Gruber, A. and Haas, E., 2017. ESA CCI Soil Moisture for improved Earth system understanding: state-of-the art and future directions. Remote Sensing of Environment, 203, pp.185-215.

Friedl, M.A., McIver, D.K., Hodges, J.C., Zhang, X.Y., Muchoney, D., Strahler, A.H., Woodcock, C.E., Gopal, S., Schneider, A., Cooper, A. and Baccini, A., 2002. Global land cover mapping from MODIS: algorithms and early results. Remote Sensing of Environment, 83(1-2), pp.287-302.

McNally, A., Shukla, S., Arsenault, K.R., Wang, S., Peters-Lidard, C.D. and Verdin, J.P., 2016. Evaluating ESA CCI soil moisture in East Africa. *International journal of applied earth observation and geoinformation*, *48*, pp.96-109.

Oleson, K.W., Niu, G.Y., Yang, Z.L., Lawrence, D.M., Thornton, P.E., Lawrence, P.J., Stöckli, R., Dickinson, R.E., Bonan, G.B., Levis, S. and Dai, A., 2008. Improvements to the Community Land Model and their impact on the hydrological cycle. Journal of Geophysical Research: Biogeosciences, 113(G1).

Parajka, J., Naeimi, V., Blöschl, G., Wagner, W., Merz, R. and Scipal, K., 2006. Assimilating scatterometer soil moisture data into conceptual hydrologic models at the regional scale. Hydrology and Earth System Sciences Discussions, 10(3), pp.353-368.

Simmer, C., Adrian, G., Jones, S., Wirth, V., Göber, M., Hohenegger, C., Janjic, T., Keller, J., Ohlwein, C., Seifert, A. and Trömel, S., 2016. Herz: The german hans-ertel

centre for weather research. Bulletin of the American Meteorological Society, 97(6), pp.1057-1068.

Wagner, W., Dorigo, W., de Jeu, R., Fernandez, D., Benveniste, J., Haas, E. and Ertl, M., 2012. Fusion of active and passive microwave observations to create an essential climate variable data record on soil moisture. *ISPRS Annals of the Photogrammetry, Remote Sensing and Spatial Information Sciences (ISPRS Annals)*, 7, pp.315-321.

Wahl, S., Bollmeyer, C., Crewell, S., Figura, C., Friederichs, P., Hense, A., Keller, J.D. and Ohlwein, C., 2017. A novel convective-scale regional reanalyses COSMO-REA2: Improving the representation of precipitation. *Meteorol. Z.*

---

## Author Response (AR1)

Dear Editor,

We appreciate the valuable and constructive comments from the Editors and the Reviewers, which helped in the improvement of the manuscript, which underwent major revisions. All comments and recommendations were addressed in great detail and incorporated in the revised manuscript. Our efforts in addressing these comments include the following:

1. **Conduct joint state-parameter updates to account for uncertainties in model parameters:** The question of uncertainties in model parameters was raised to reduce the bias of the forward model. In order to examine the impact of parameter estimation on the overall quality of the assimilation, we performed additional joint state-parameter estimation in our data assimilation experiment to correct for possible systematic model bias by modifying parameters. In addition, we revisited our methods to generate ensembles in the previous version of the manuscript, which was modified resulting in improved open loop simulations relative to the deterministic model run as pointed out by Reviewer 2. (see for example Figure 6 and 9 in the revised manuscript compared to Figure 6 and 10 in the HESSD version).
Please see [R2.1] below for further clarification.

2. **Increase size of ensemble members:** A point of criticism is the small ensemble size, which may have a negative impact on the result. Our selection of ensemble size is due to the large data amounts produced in the forward run of the 3km resolution model. For example, one year of model run with 20 ensemble members requires 680GB of computer storage per output variable (i.e. equivalent to 5TB of storage for 7 years of simulations per variable at daily time scale). We have increased the ensemble size to 20 in the revised manuscript and updated our results accordingly. We also added new assimilation experiments in the supplementary material to evaluate the impact of ensemble size on the performance of EnKF by increasing the number of members to 20 relative to 12 that we used in the HESSD version.
Please see our response to [R2.9] below for detailed discussions.

3. **Replace CLM-default vegetation parameters with MODIS based data:** In the revised manuscript, we replaced the CLM default Plant Functional Types (PFTs) annual monthly leaf area index (LAI) cycles with prescribed LAI from MODIS to consider the inter-annual variability of the vegetation. This modification resulted in spatially distributed and temporally continuous LAI data within each PFT for the considered time period of 2000-2006 in the model. To account for annual variability in LAI, yearly model runs were performed where the LAI information was updated at the start of each year run.
Please see our response to [R3.4] for detailed discussions.

4. **Conduct further evaluation of observation data rescaling to match the model resolution:** In order to evaluate whether the rescaling of the ESA CCI SM data/E-RUN runoff data to model resolution may introduce any bias in the data, we

compared the original 0.25° against 0.0275° ESA CCI soil moisture. Only small differences between the two resolutions were visible. We added a figure (Figure S5) in the supplementary material to show the effects of resolution and interpolation on the results.
Please see our response to [R1.3] for f detailed discussion.

5.  **Provide justification for using gridded observation runoff data for model validation.** Point of concern was the use of non-routed gridded runoff observation (E-RUN runoff data product Gudmundsson and Seneviratne, 2016) for model validation, instead of gauge station discharge observations. In the revised manuscript, we provided further clarification to justify our preference of using this dataset for the model validation. We conducted further evaluation to compare aggregated runoff using E-RUN data for few watersheds in Europe with monthly discharge data measured at gauge stations obtained from Global Runoff Data Center. Our comparison showed a good agreement of aggregated E-RUN runoff data with observed discharge (Fig. S1). This provided further confidence of using the E-RUN runoff data for model validation. Please see our response to [R2.7] for detailed discussion.

6.  **Clarifying concerns regarding the added value of high resolution data assimilation experiment.** One of the main concerns raised by the reviewers is regarding the added value of the high resolution data assimilation experiment using the coarse-resolution remotely sensed data. In our opinion, the unique contribution of this study is to demonstrate the added value of merging coarse resolution satellite observations through data assimilation with a land surface model to generate higher spatial resolution, downscaled estimates of the surface soil moisture at very high resolution with complete spatio-temporal coverage and with improved accuracy compared to open loop model estimates. To the best of our knowledge, this is the first study to implement a high resolution modeling framework to provide a long term downscaled daily soil moisture product at 3 km resolution over Europe. The soil moisture estimates with improved spatial resolution from the assimilation offer a new tool for monitoring soil water content with distinct benefits over the original CCI-SM data. In addition, using this data assimilation modeling framework allowed us to assess the potential impact of assimilating longer-term soil moisture observations on hydrologic simulations and identified model limitations to correctly represent processes controlling the hydrologic fluxes and states and their dependence on antecedent soil moisture. Please see [R2.2], [R2.6], [R2.8] and [R3.1] for detailed discussions.

7.  **In addition to the major additional work performed in the revision,** a new discussions section (Sec. 4) was added in the revised manuscript and we included multiple references recommended by the reviewers, and revised the manuscript to clarify various issues identified by the reviewers.

Detailed responses are provided below. Once again, we thank the Editor and Reviewers for their time and effort that helped us to significantly improve the quality of this study.

Sincerely,

Bibi S. Naz on behalf of all Co-authors

**Response to comments by Anonymous Referee #1**

General comments: The manuscript aims to demonstrate that a high resolution data-assimilation modelling framework allows improving soil moisture and runoff simulations at a continental scale. Thus, it addresses a question within the scopes of the journal. Aims of the work are overall clearly outlined and supported by references.

We would like to thank the anonymous reviewer for his/her comments and constructive suggestions, which we believe have led to an improved manuscript.

I suggest to better justify the choices of models and datasets and temporal domain (2000-2006).

[R1.1] We selected CLM3.5 because it is one of the most complete land surface model at the moment and part of a large community effort (Community Earth System Model; http://www.cesm.ucar.edu/).  It has been widely applied at continental and global scales to understand how land processes and anthropogenic impacts affect climate (e.g. Dickinson et al., 2006). The CLM model parameterizes most of the land surface processes (such as infiltration, evaporation, surface runoff, subsurface drainage, canopy and snow processes) using the water and energy balance equations. In addition, CLM3.5 was designed for coupling with climate models and is also part of the fully coupled Terrestrial Systems Model Platform (TerrSysMP; Shrestha et al., 2014) that simulates the full terrestrial hydrologic cycle including feedbacks between atmosphere, land-surface and subsurface compartments of the water cycle. Moreover, the CLM model can efficiently run for large model domains and at high spatial resolution. Since we performed our simulations at high spatial resolution and at continental scale, we selected the TerrSysMP-PDAF modelling framework (Kurtz et al., 2016) which can run on high performance computational infrastructure and can efficiently cope with the high computational burden of ensemble-based data assimilation (Page 5, line 13).

With respect to the time span simulated and COSMOREA6 data, we preferred to use this data over other datasets, because of its high spatial resolution in comparison to other commonly used forcing datasets such as the European gridded data set (E-OBS) (Haylock et al., 2008) and Interim ECMWF Reanalysis (ERA-Interim; Dee et al., 2011) at 25 and 80 km resolution, respectively. COSMO-REA6 was produced through the assimilation of observational meteorological data using the existing nudging scheme in COSMO with boundary conditions from ERA-Interim data as discussed in Bollmeyer et al. (2015). We used data from 2000 – 2006, which were available at the beginning of this study. The COSMO-REA6 is only now publicly available for a longer time period of 1995 – 2015.

We now added additional motivation for the model selection, time period and forcings datasets in the revised manuscript (Page 8, line 10).

Data-assimilation results are compared to open-loop simulations to quantitatively assess this improvement basing on root mean square error and mean bias error estimates with respect to CCI SM data. Overall results are well supported by figures and graphs. However, I would suggest the Authors to give a more detailed explanation for differences in overestimate and underestimate between the regions and between the seasons (particularly in par. 3.1.1). It seems that this part has been deepened more for runoff than for soil moisture.

[R1.2] We included two new Tables (Table R1 and Table R2) (Table 1 and 2 in the revised manuscript) and modified Fig. R1 (Figure 3 in the revised manuscript) as shown below to explain the differences in both magnitude and spatial patterns and also an extended discussion of the soil moisture differences in different regions and between seasons (Page 13, Sec. 3.1.1).

Table R1. Difference in CLM-OL and CLM-DA simulated mean seasonal SWC (mm3/mm3) with CCI-SM data for all PRUDENCE regions and all seasons, i.e. winter (DJF), spring (MAM), summer (JJA) and autumn (SON).

| Regions | CLM-OL minus CCI-SM | | | | CLM-DA minus CCI-SM | | | |
| --- | --- | --- | --- | --- | --- | --- | --- | --- |
| | Winter | Spring | Summer | Autumn | Winter | Spring | Summer | Autumn |
| BI | 0.03 | 0.05 | 0.02 | 0.04 | 0.01 | 0.02 | -0.02 | 0.00 |
| IP | 0.07 | 0.09 | 0.11 | 0.09 | 0.04 | 0.06 | 0.07 | 0.05 |
| FR | 0.03 | 0.06 | 0.06 | 0.05 | 0.01 | 0.03 | 0.02 | 0.02 |
| ME | 0.04 | 0.07 | 0.06 | 0.05 | 0.02 | 0.04 | 0.01 | 0.02 |
| SC | 0.07 | 0.08 | 0.02 | 0.03 | 0.04 | 0.05 | -0.02 | 0.00 |
| AL | 0.03 | 0.05 | 0.04 | 0.02 | 0.01 | 0.02 | 0.00 | -0.01 |
| MD | 0.05 | 0.07 | 0.08 | 0.06 | 0.03 | 0.04 | 0.04 | 0.03 |
| EA | 0.07 | 0.08 | 0.07 | 0.05 | 0.05 | 0.05 | 0.03 | 0.02 |
| EU | 0.09 | 0.11 | 0.09 | 0.09 | 0.07 | 0.08 | 0.05 | 0.06 |

Table R2. Evaluation performance criteria for comparing CLM-OL and CLM-DA with CCI-SM (spatially averaged SWC, surface layer, EU and PRUDENCE regions)

| | Soil Moisture (CLM-OL) | | | | | | | | |
| --- | --- | --- | --- | --- | --- | --- | --- | --- | --- |
| | EU | BI | IP | FR | ME | SC | AL | MD | EA |
| PBIAS (%) | 54.1 | 16.4 | 50.7 | 24.7 | 25.6 | 23.0 | 16.4 | 33.4 | 34.8 |
| RMSE (mm$^3$/mm$^3$) | 0.10 | 0.05 | 0.10 | 0.07 | 0.07 | 0.06 | 0.05 | 0.07 | 0.08 |
| MAE (mm$^3$/mm$^3$) | 0.09 | 0.04 | 0.09 | 0.06 | 0.06 | 0.05 | 0.04 | 0.07 | 0.07 |
| R | 0.48 | 0.41 | 0.75 | 0.60 | 0.51 | -0.14 | 0.55 | 0.80 | 0.40 |
| | Soil Moisture (CLM-DA) | | | | | | | | |
| | EU | BI | IP | FR | ME | SC | AL | MD | EA |
| PBIAS (%) | 36.4 | 3.1 | 33.2 | 10.1 | 11.0 | 9.0 | 3.0 | 18.1 | 19.0 |

| | | | | | | | | |
|---|---|---|---|---|---|---|---|---|
| **RMSE (mm³/mm³)** | 0.07 | 0.03 | 0.07 | 0.04 | 0.04 | 0.05 | 0.03 | 0.04 | 0.06 |
| **MAE (mm³/mm³)** | 0.06 | 0.03 | 0.06 | 0.03 | 0.03 | 0.04 | 0.03 | 0.04 | 0.05 |
| **R** | 0.51 | 0.40 | 0.76 | 0.61 | 0.54 | -0.12 | 0.56 | 0.80 | 0.42 |

[Figure]

Fig. R1. Temporally averaged soil moisture (mm3/mm3) content over the 2000 – 2006 period for a) CCI-SM, b) CLM-OL c) CLM-DA, and difference between d) CLM-OL and CCI-SM and d) CLM-DA and CCI-SM for DJF (December, January and February), MAM (March, April, May), JJA (June, July and August) and SON (September, October, November) seasons.

Besides I would suggest the Authors to better explain if and how results can be affected by spatial resolution differences in the data.

[R1.3] See our response and Figure R2 (Fig. S5 in the supplementary material) below to

the specific comment 6 (R1.9) on resolution and interpolation effects (Page 17, line 6).

The advancements described in this study would benefit from a quantitative or qualitative comparison with other studies claiming the use of data assimilation for improving soil moisture and runoff simulations, to assess if the obtained results are satisfying. My recommendation is to accept the manuscript with minor review.

Specific comments

Page 2 Line 32 and Page 6 Line 2: I would suggest to add spatial resolution in km, as done is other sections of the text

[R1.4] It has been modified in the revised manuscript (Page 3, line 12).

Page 2 Lines 18-27: I think sentences here are a bit contradictory. I would suggest the Authors to better clarify what is commonly done in the state of the art and what is rarely done (and eventually why it is rarely done), in order to better highlight element of novelties of this work.

[R1.5] We appreciate the suggestion. We added several references of other assimilation/modeling studies over Europe in the introduction section of the revised manuscript. We added following text at Page 3, line 14 in the revised manuscript.

For instance, several studies have explored the role of soil moisture assimilation over Europe, in different modeling frameworks (e.g. Albergel et al., 2017; Brocca et al., 2010; De Rosnay et al., 2013; Draper et al., 2009; Ni-Meister et al., 2006). Using NASA's global Catchment Land Model (CLSM), Ni-Meister et al. (2006) improved simulated soil moisture over small Eurasia catchments through assimilation of near surface soil moisture derived from Scanning Multichannel Microwave Radiometer (SMMR). Using Extended Kalman Filter (EKF), Draper et al. (2009) demonstrated the usefulness of assimilating near-surface soil moisture observations from C-band Advanced Microwave Scanning Radiometer (AMSR-E) in the land surface model ISBA (Interactions between Soil, Biosphere and Atmosphere) at 9 km resolution over continental Europe. More recently, Albergel et al. (2017) showed the potential of using the satellite-derived soil moisture data from European Space Agency (ESA) Climate Change Initiative (CCI) over Europe and Mediterranean domain to sequentially assimilate soil moisture and leaf-area index product into the ISBA (Interactions between Soil, Biosphere and Atmosphere) land surface model at 0.5° (~50 km) resolution. They found significant improvements in the surface soil moisture but little improvements of discharge estimates when compared to the open loop (i.e. no assimilation) simulations.

In the revised manuscript, we also highlighted the added value of merging coarse resolution satellite observations through data assimilation with a land surface model to generate higher spatial resolution, downscaled estimates of the surface soil moisture profile with complete spatio-temporal coverage and with a higher accuracy than that of the open loop model estimates. To the best of our knowledge, this is the first study of its kind to provide a downscaled daily soil moisture product at 3km resolution over Europe.

We discussed the added value of our data assimilation experiment in the introduction section (Sec. 1) of the revised manuscript to clarify the objectives of the study.

Figure 1b: I would suggest to use a discretized legend as it represents different classes

[R1.6] We appreciate this comment. The Fig. 1 has been modified in the revised manuscript.

Page 4 Lines 2-3: the Authors state that "CLM3.5 was used in this study, instead of its most recent version, to keep the modelling framework consistent to Kurtz et al. (2016)." Would it be possible to hypothesize some advantages or disadvantages in using CLM most recent versions?

[R1.7] We added the following text to highlight the differences in different versions of the CLM model:

Compared to CLM3.0, Oleson et al.,(2008) showed that CLM3.5 exhibits more realistic partitioning of ET into its components (i.e. transpiration, ground evaporation, and canopy evaporation), which resulted in overall improvements in the representation of the annual cycle of total water storage. Previous studies also showed that CLM 3.5 produces too high soil moisture with too lower variability compared to root zone soil moisture modelled by later CLM versions (4.0 and 4.5) (e.g. Lawrence et al., 2011 and Niu et al., 2011). In order to reduce these biases, Li and Ma, (2015) introduced a factor to describe soil porosity and increase recharge water from the soil column to the aquifer in the newer CLM versions which offer improved solutions related to soil moisture and biogeochemical processes. However, Lawrence et al. (2011) showed that the differences between CLM3.5 and new versions of CLM with respect to soil moisture variability remain low when compared to observations.

This information is added in the revised manuscript (Page 5, line 13).

Page 5 Line 9: I would suggest to briefly explain how this conversion is done or at least provide a reference?

[R1.8] The land cover information for each PFT is based on the Moderate Resolution Imaging Spectroradiometer (MODIS) MCD12Q1 (version 5) land cover product (Friedl et al., 2002), which contains a classification of the dominant land cover. The dominant land cover information was first aggregated to the model resolution, computing the percentage of all 500m pixels per 3 km grid cell. The aggregated land cover information was then transferred to the CLM-prescribed Plant Functional Types (PFT) on the basis of WorldClim climate data (Hijmans et al., 2005). This information is added in the revised manuscript (Page 7, line 15).

Page 6 Lines 12-15: Would it be possible to perform the inverse resampling (from 0.0275˚ to 0.25˚) and compare results with the one from 0.0275˚ to 0.25˚? The same could be said for runoff rates (paragraph 2.3.3)

[R1.9] We compared the original 0.25° against 0.0275° ESA CCI soil moisture. Only

small differences between the two resolutions were visible particular for the time period of 2003 – 2006 (Fig. R2). We found some differences in the first two years (2000 and 2001) and in the regions where the temporal coverage of the ESA CCI data is less than 30%. However, for the time period and regions with a good coverage of ESA CCI soil moisture data, the differences in the resolution were not significant. Similarly, for the model outputs, we don't see any significant difference (e.g. R-squared value in both cases is 0.56) in our results between 0.0275° and 0.25° when compared to the original CCI data at 0.25° resolution as shown in Fig. R3 as an example for one region.

This information is discussed in the discussion section of the revised manuscript and Figure R2 is included in the supplementary material (Fig. S5) (Page 17, line 6 in the revised manuscript)

[Figure]

Figure R2. Comparison of spatially averaged 2000 – 2006 daily ESA CCI soil moisture data between original 0.25° (black) and interpolated 0.0275° (red) over PRUDENCE regions.

[Figure]

Figure R3. Scatter plot showing the comparison of spatially averaged daily soil moisture between ESA CCI (0.25°) and CLM at 0.25° and 0.0275° resolution over the PRUDENCE region France.

Page 7 Line 27: Is it possible that Authors refer to another figure?

[R1.10] This has been now corrected in the revised manuscript (Page 11, line 22).

Page 10 Line 3: "CLM-DA reduces the runoff bias compared to CLM-OL". Evidence of this is in Figure 9, not in Figure 7. It would suggest the author to better clarify this point or remove it (as it is already stated in line 20) I suggest to add a table similar to Table 1 also for soil moisture.

[R1.11] We appreciate this comment. We removed this sentence and added a new table (Table 1 in the revised manuscript) for soil moisture (similar to Table 2 in the revised manuscript)

Technical corrections

Page 1 Line 31: Western et al., 2002 has to be changed with Western et al., 2004 according to references

[R1.12] We appreciate this comment. This has been now corrected in the revised manuscript (Page 2, line 7).

Page 2 Line 16: bracket missing after ". . .Clark et al., 2011)"

[R1.13] This has been addressed in the revised manuscript.

Page 2 Line 25: López et al., 2016 is missing in the references and there is a repetition of the name in the text

[R1.14] We appreciate this comment. This has been corrected in the revised manuscript (Page 3, line 8).

Page 2 Line 28: Rains et al., 2017 is missing in the references

[R1.15] The missing reference has been added in the revised manuscript.

Page 3 Line 28: remove comma before brackets Page 4 Line 9: remove the semicolon before bracket

[R1.16] This has been corrected in the revised manuscript (Page 6, line 11).

Page 4 Lines 12-27: I think this sentence is too long, thus I would split it into different sentences maybe one for each formula

[R1.17] This has been modified in the revised manuscript for clarity (Page 6 , line 13).

Page 5 Lines 30-31: Wahl et al. 2017 is missing in the references

[R1.18] The missing reference has been added in the revised manuscript.

Page 6 Line 13: Jones, 1999 is missing in the references

[R1.19] The missing reference has been added in the revised manuscript.

Page 6 Line 19: remove full stop after brackets

[R1.20] This has been corrected in the revised manuscript (Page 9, line 14).

Page 9 Line 28: replace Decker and Zeng, 2009 with Zeng and Decker, 2009, as for references

[R1.21] This has been corrected in the revised manuscript (Page 15, line 1).

Page 11 Lines 32-33: I think that Authors mean "it is preferable to account for additional model parameter uncertainties that shows a high sensitivity towards runoff" instead of "it is preferable to account for additional model parameter uncertainties towards runoff that shows a high sensitivity"

[R1.22] We appreciate this comment. This has been modified in the revised manuscript for clarity (Page 18, line 1).

Page 12 Line 12: Remove "This study showed that", as it is a repetition of the above line

[R1.23] This has been modified in the revised manuscript for clarity (Page 19, line 12).

Page 12 Line 30: "The improvement in peak runoff could be OF particular importance in the management of extreme events such as flooding"

[R1.24] This has been modified in the revised manuscript for clarity.

Page 14 Line 7: reference missing in the text

[R1.25] The missing reference has been corrected in the revised manuscript.

Page 15 Line 9: reference missing in the text

[R1.26] The missing reference has been corrected in the revised manuscript.

Page 17 Line 18: reference missing in the text

[R1.27] The missing reference has been corrected in the revised manuscript.

Page 19 Line 31: replace 998 with 1998

[R1.28] We appreciate this comment. This has been corrected in the revised manuscript.

Page 20 Line 34: year of publication should be after doi, as for the other references

[R1.29] This has been corrected in the revised manuscript.

Page 21 Line 8: reference missing in the text

[R1.30] The missing reference has been corrected in the revised manuscript.

Figure 2c: better if months initials are in English

[R1.31] Figure2c has been modified in the revised manuscript.

**Response to comments by Anonymous Referee #2**

**General:**

The submitted manuscript of Naz et al. (2018) evaluates the effect of soil moisture assimilation (namely ESA-CCI product) into the CLM 3.5 over Europe during 2000-2006. The Ensemble Kalman Filter is used for the model analysis, the observations are sampled using 100 randomly located points across the entire domain, while the remaining locations are used for independent evaluation. The CLM model operates at 3km spatial resolution, while the assimilation product is available at coarser, approx. 25km (0.25degree) resolution. Additionally, the gridded (monthly) runoff product available at 0.5 degree is used for evaluation of the model runs. Results are presented for the open loop (OL) and data assimilation (DA) runs. Results are presented in terms of the RMSE and relative bias per nine PRUDENCE regions. I find this topic relevant for HESS, however, according to my opinion, the manuscript is not suitable for publication in the present form:

We would like to thank the anonymous reviewer for his/her comments and constructive suggestions, which we believe have led to an improved manuscript.

1. As correctly stated in Section 3.3, before applying any DA methods, the modeler should better parameterize and constrain the model parameters, reduce systematic biases etc. I am afraid you cannot apply DA after seeing those strong biases in the open loop estimates (Figures 6 or 10) at all. I encourage authors to pay attention to proper model calibration before DA analysis.

[R2.1] Hydrologic states and fluxes simulated by Land Surface Models (LSM) are often biased due to, e.g., systematic biases in input forcings, uncertainty of input parameters, the use of parameterizations and uncertainty related to the (bio)physical processes representation in the models (e.g Yin et al., 2014; Han et al., 2014). Traditionally, the calibration of model parameters is performed using in situ data to resolve these biases. However, such in situ data – soil moisture in particular – are often unavailable at a large scale. In this study, we have utilized the remotely sensed soil moisture for evaluating the performance of DA in reducing the model biases as previous studies have shown (e.g. Moradkhani et al., 2005; Liu and Gupta, 2007; Kumar et al., 2008; Nie et al., 2011; Chen et al., 2013; Lievens et al., 2016; Lannoy and Reichle, 2016). Another motivation of this study was not only to improve the soil moisture through data assimilation approach, but also to evaluate the assimilations impacts on other land surface variables such as runoff. Model calibration prior to assimilation, e.g., with the runoff data set of Gudmundsson and Seneviratne (2016) would violate the independency criterion of calibration and validation data for the impact assessment of the data assimilation procedure (Page 2, line 26 in the revised manuscript).

Furthermore, in land surface modeling systematic differences between the model climatology and the observation data climatology are traditionally corrected before

assimilation, to ensure that data assimilation is applied under conditions of no systematic bias. However, different procedures to correct bias are used, like the estimation of a single constant bias value, seasonal dependent bias or CDF-matching (e.g. Reichle and Koster, 2004 and Drusch et al., 2005). The procedure has some important limitations: (i) the bias is only partially corrected or over-corrected; (ii) the bias in the DA-procedure is not assigned to the model or measurement data, but after the assimilation it is implicitly assumed that the systematic bias is related to the bias in the measurements (model states are not corrected for a systematic bias). We did therefore not perform any bias correction of the soil moisture observation by rescaling of the observations to model climatology. A further argument for not following this approach was that spatial patterns could be altered and thereby some of the independent information provided by the satellite may be removed; it is desirable to retain as much of the independent satellite information as possible. This information is now discussed (Page 9, line 23 in the revised manuscript).

The alternative procedure we followed here was to neglect systematic biases, but assimilate with a sufficient ensemble spread so that observations allow correcting model predictions. This approach results in a stronger corrective effect of measurement data. A further alternative is to attribute systematic biases to erroneous model parameter values, which is one of the main sources of error and uncertainty in land surface model predictions. We explored this option now in the revised version of the manuscript using a joint state-parameter estimation approach. Although this approach has also important limitations, related to the fact that we do not know well enough the relative importance of systematic model errors and systematic errors in the measurement data, an advantage is that we correct for possible systematic model bias by modifying soil texture parameters (Page 6, line 13 in the revised manuscript).

Therefore, we evaluated now the impact of soil texture properties like the percentages of clay and sand on the CLM model performance and jointly estimated model states and parameters in the data assimilation experiment. We plotted the results as mean monthly soil moisture for the 2000 – 2006 time period for CLM-OL, CLM-DA (only state update) and CLM-DA (joint state and parameter updates) as shown in Fig. R4. We found that in both cases (state updating alone, joint state-parameter updating) the DA-runs were quite close to the observed values. However, parameter updating is expected to correct part of the systematic model bias and other fluxes like evapotranspiration and also discharge can show larger modifications as response to the different parameters.

[Figure]

Figure R4: Multi-ensemble spatially averaged mean monthly soil water content (2000 – 2006) simulated with CLM-OL (no assimilation; gray colour) and CLM-DA (with data assimilation) and compared with CCI-SM data over the PRUDENCE regions. The blue colour indicates CLM-DA-S (state updates only) and red colour indicates CLM-DA-SP (joint state-parameter update).

In the revised manuscript, we performed additional joint state-parameter estimation in our data assimilation experiment to correct for possible systematic model bias by modifying parameters. In addition, we revisited our methods to generate ensembles in the previous version of the manuscript, which was modified resulting in improved open loop simulations relative to the deterministic model run as pointed out by Reviewer 2. (see for example Figure 6 and 9 in the revised manuscript compared to Figure 6 and 10 in the HESSD version).

2. This is not much of surprise when soil moisture gets assimilated into a model that model simulations at the analysis step get more close to the "observation- based" product, as much as the prescribed observation errors allow (given there exists spatial correlation between the 100 assimilated locations and remaining "withheld" observations). By assimilating the ESA-CCI product, the authors claim to improve the initial conditions of the model. That's all . . . I would welcome then the added value/implications of the improved initial model (wetness) conditions (e.g. with respect to some longer lead-times): OL vs. DA.

[R2.2] Most land data assimilation studies only aim at improving the model initial conditions. These improved model initial conditions would impact other modeled fluxes like discharge (which is a focus of this paper), but also latent and sensible heat fluxes between the land and atmosphere, which could potentially improve weather predictions. The added value of our study is to apply a data assimilation modeling framework over Europe along the complete simulation time span to derive a longer-term and high spatial

resolution land surface data assimilation product in order to increase monitoring accuracy for land surface moisture and water states and fluxes.

Beyond that, a continuous DA approach allows to monitor long-term changes of the terrestrial water cycle and enhance our understanding of the role of land surface processes on, e.g. atmospheric boundary layer dynamics. Therefore, high quality soil moisture product is needed. Since point scale measurements from in situ sensors are only available at a limited number of locations, remotely sensed soil moisture products are promising for a more large scale improvement of soil moisture variations. However, sparse data coverage in satellite observations still limits their ability to provide spatially and temporally consistent time series of water balance estimates. To overcome this limitation, in our study, we used data assimilation to merge coarse resolution satellite observations with a land surface model to generate higher horizontal resolution i.e. downscaled estimates of the full soil moisture profile with complete spatio-temporal coverage.

We now discussed the added value of our data assimilation experiment in the introduction section of the revised manuscript (Page 3, line 31 and Page 4, line 21 in the revised manuscript).

3. Additionally, using EnKF the authors modify the internal model states and thus introduce some numerical instability (against internal physical constraints for the model), which was not discussed at all. How do you handle this issue after the analysis step?

[R2.3] In the DA scheme we ensured that updated states (soil moisture) were kept in reasonable physical ranges (between residual soil moisture and porosity) to yield physically consistent estimates of fluxes and maintained the water budget. For the soil moisture update, the values of the updated soil moisture were restricted to values between zero and saturated soil water content. For the soil texture update, a value within 1% is assigned to sand and clay percentages in case the updated values are less than zero. In case the updated sum of the sand and clay are greater than 100%, the values were constrained to the normalized sum of updated soil and clay percentages. Other soil parameters such as the soil hydraulic and thermal parameters were adjusted after soil texture update using pedo-transfer functions.

Additionally, numerical stability is not significant compared to, e.g., groundwater models/ atmospheric models where PDE's are solved with iterative methods. In CLM (and generally in land surface models) the equations/algorithms for deriving mass/energy budgets/transfer are simpler and more robust (numerically) than for full PDE-based systems.

We now discussed these points in the revised manuscript (Page 7, line 4 in the revised manuscript).

4. Hardly any discussion for (OL and DA) results is done with respect to other SM data assimilation/modeling studies over Europe ... which could be used as a benchmark(?)

[R2.4] We appreciate the suggestion. We added several references of other assimilation/modeling studies over Europe in the introduction section of the revised manuscript. For instance, several studies have explored the role of soil moisture assimilation over Europe, in different modeling frameworks (e.g. Albergel et al., 2017; Brocca et al., 2010; De Rosnay et al., 2013; Draper et al., 2009; Ni-Meister et al., 2006). Using NASA's global Catchment Land Model (CLSM), Ni-Meister et al. (2006) improved simulated soil moisture over small Eurasia catchments through assimilation of near surface soil moisture derived from Scanning Multichannel Microwave Radiometer (SMMR). Using Extended Kalman Filter (EKF), Draper et al. (2009) demonstrated the usefulness of assimilating near-surface soil moisture observations from C-band Advanced Microwave Scanning Radiometer (AMSR-E) in the land surface model ISBA (Interactions between Soil, Biosphere and Atmosphere) at 9 km resolution over continental Europe. More recently, Albergel et al. (2017) showed the potential of using the satellite-derived soil moisture data from European Space Agency (ESA) Climate Change Initiative (CCI) over Europe and Mediterranean domain to sequentially assimilate soil moisture and leaf-area index product into the ISBA land surface model at 0.5°(~50km) resolution. They found significant improvements in the surface soil moisture but little improvements of discharge estimates when compared to the open loop (i.e. no assimilation) simulations (Page 3, line 15 in the revised manuscript).

Applying CLM over EU is indeed challenging, but there are other models already able to simulate SM and the choice of CLM is not well described either.

(Below is the same response to Reviewer 1(R1.1))

[R2.5] We selected CLM3.5 because it is one of the most complete land surface model at the moment and part of a large community effort (Community Earth System Model; http://www.cesm.ucar.edu/). It has been widely applied at continental and global scales to understand how land processes and anthropogenic impacts affect climate (e.g. Dickinson et al., 2006). The CLM model parameterizes most of the land surface processes (such as infiltration, evaporation, surface runoff, subsurface drainage, canopy and snow processes) using the water and energy balance equations. In addition, CLM3.5 was designed for coupling with climate models and is also part of the fully coupled Terrestrial Systems Model Platform (TerrSysMP; Shrestha et al., 2014) that simulates the full terrestrial hydrologic cycle including feedbacks between atmosphere, land-surface and subsurface compartments of the water cycle. Moreover, the CLM model can efficiently run for large model domains and at high spatial resolution. Since we performed our simulations at high spatial resolution and at continental scale, we selected the TerrSysMP-PDAF modelling framework (Kurtz et al., 2016) which can run on high performance computational infrastructure and can efficiently cope with the high computational burden of ensemble-based data assimilation (Page 5, line 13 in the revised manuscript).

The authors have "high-resolution" in their title. I strongly encourage them to eliminate this from the title, especially if they use such coarse scale data to as-similate.

[R2.6] As discussed in the response to comment 2, in this study we used data assimilation approach to highlight the added value of merging coarse resolution satellite observations with a land surface model to generate higher spatial resolution, downscaled estimates of the surface soil moisture profile with complete spatio-temporal coverage and with a higher accuracy than that of the open loop model estimates. Many applications (such as drought, flood, irrigation management) require observations of the complete soil moisture profile and with finer spatial and temporal resolutions than those of remotely sensed products. To the best of our knowledge, this is the first study of its kind to provide a downscaled daily soil moisture product at 3 km resolution over Europe. In order to accommodate the reviewer comment, we replaced "high resolution" in the title with "3 km".

5. Why the authors did not use "high-resolution" discharge data for independent model evaluation? There are thousands of gauges with daily time step over Europe, if the routing would be enabled. I am afraid that using monthly gridded runoff is not sufficient for a "high-resolution" study.

[R2.7] We agree that discharge data from many gauging stations are available and can be used for independent evaluation. However, in the current version of the CLM model, the routing scheme is based on simple linear storage outflow relationships, in which a prescribed channel velocity field without temporal variability is used. This simplified assumption can lead to an offset even in the monthly peak streamflow in large catchments (Li et al., 2011). In addition, in the CLM 3.5, the river routing module is implemented at 0.5° where the discretization of river routing elements is based on a grid method in which the grid for river routing is independent of the grid for runoff simulation. Therefore, a coarse spatial resolution, such as 0.5° can lead to unrealistic flow accumulation paths and cannot be used to evaluate discharge at small catchments. Due to the aforementioned points, adequate validation of the results is therefore not possible. However, our comparison of aggregated runoff using E-RUN data for few watersheds with monthly discharge measured at gauge stations and obtained from Global Runoff Data Center (GRDC; Global Runoff Data Center, 2011) in Europe showed a good agreement with observed discharge (Fig. S1 in the revise manuscript) as show below in Fig. R5.

[Figure]

Figure R5: Comparison of monthly GRDC discharge data with E-RUN runoff product for the Oder, Vienne, Mosselle and Danube river basins. The non-routed runoff rates (mm/day) from the E-RUN data were converted to flow volumes by aggregating the data for each grid cell within basin and multiplying with the drainage area.

In this study, we instead used the E-RUN runoff product which combines observed river flow with gridded estimates of precipitation and temperature using machine learning. Therefore, this gridded runoff dataset is solely derived from observations and does not rely on any modeling assumptions. We believe that using an observation-based non-routed gridded runoff product has distinct advantage to evaluate the impact of soil moisture assimilation on runoff at every grid cell within a spatial domain. Using gridded runoff is also useful to evaluate model structure errors in representation of runoff generation in the model. In the land surface models such as CLM, the representation of runoff is often simplistic and conceptual and many previous studies have shown that performance of the CLM model in simulating hydrological processes varies based on regions. This might be attributed to the fact that assumptions to estimate surface and subsurface runoff in the model might be valid in some regions but not in other regions (e.g. humid vs. dry regions). We also noted similar behavior of CLM in our study where the assimilation of soil moisture helped to improve runoff in some regions but degraded in other regions (Page 10, line 12 in the revised manuscript).

In the CLM 3.5 runoff scheme, runoff is partitioned into surface flow and subsurface flow and basic simulation element of runoff is the grid cell. In the revised version of the manuscript, the performances of the surface flow and base flow were evaluated separately in order to identify the dominant factor in total runoff generation as a result of soil moisture assimilation.  T show this evaluation, a new Fig. 11 is included in the revised paper as shown below (Fig. R6) (Page 18, line 13  in the revised manuscript).

[Figure]

Figure R6: Comparison of spatially-averaged monthly simulated runoff components (surface and subsurface runoff) between CLM-OL and CLM-DA over Europe and PRUDENCE regions for the time period of 2000 – 2006. Gray line represents 1:1 line.

6. The authors could have easily run their model at the resolution of the data and save their larger efforts in computer resources.

[R2.8] The goal and added value of the study was to produce a high-resolution, downscaled land surface hydrology over Europe. It is true that (a large number of) 1D soil moisture DA-experiments could have been conducted at the measurement locations (where we assimilated ESA CCI soil moisture in this experiment), but an essential component of this work is that we updated soil moisture contents at other locations (at the European scale), based on spatial correlations, and investigated whether soil moisture

characterization between measurement locations could also be improved, and if this improves also runoff characterization at the European scale. As a conclusion, we respectfully disagree with this point of the reviewer.

It should also be noted that we made an extensive effort to collect and organize high resolution land surface data and atmospheric forcing datasets to implement the CLM model at 3-km resolution over Europe. Particularly, the COSMO-REA6 is available at a high 6 km spatial resolution. The organization of COSMO-REA6 hourly record for the entire Europe is not a trivial task. The applicability of COSMO-REA6 for a land surface model simulation over the EURO-CORDEX domain has never been shown previously.

In addition to forcings, this study also used high-resolution land surface information in order to better represent the effects of land surface heterogeneity and provide climate information at the scales needed for impact assessment. In the current version of the CLM model, the officially released land surface datasets are provided at 0.5° by 0.5° or coarser resolutions. For example, in our study, PFTs fractional cover data were derived using high resolution MODIS data. Such a data-intensive effort is unprecedented in the previous studies, and hence the new resource will be valuable because it will prompt future hydro-climate studies.

These points are now discussed in the revised manuscript (Page 11, line 24 in the revised manuscript)

7. Another limitation of this study is the limited ensemble size. 12 members are way too low (this number is stated on p. 7, l. 23). Also, the ad-hoc construction of the perturbations needs better reasoning and clarifications!

[R2.9] While we agree with this comment, the number of ensemble member, however, was set to 12 members in our study, due to the large number of grid cells and required computational resources. From previous literatures (e.g., Kumar et al., 2008; Pan and Wood, 2010; De Lannoy et al., 2012; Yin et al., 2015), it is clear that the performance of EnKF relies on the ensemble size. For example, (Kumar et al., 2008; Yin et al., 2015) indicated that when the ensemble size is close to 12, it may lead to efficient DA updating process, while (Pan and Wood, 2010; De Lannoy et al., 2012) suggested 20 ensemble members (Page 11, line 9 in the revised manuscript).

We now evaluated the impact of the number of ensembles numbers on the performance of EnKF by increasing the ensemble size to 20 and run the model for one test year (i.e. 2006). We plotted the results as an ensemble mean of spatially averaged daily soil moisture for the year 2006 for CLM-DA (12 vs. 20 ensemble members) and compared with the daily ESA CCI soil moisture values over PRUDENCE regions as shown in Fig. R7 and Fig. S4 in the supplementary material of the revised manuscript.

[Figure]

Figure R7: Spatially averaged ensemble daily soil water content (SWC) simulated with CLM-DA (ensemble mean of 12 and 20 ensemble members) and compared with CCI-SM data for year 2006 over the PRUDENCE regions. The R2 values in each panel are r-squared calculated for both CLM-DA with 12 and 20 ensemble members.

While, we see some improvements in the simulated soil moisture as results of using 20 ensemble members, particularly in the winter season and in some regions such as, SC, Al and EA when compared to observations, in both cases the simulated soil moisture from the DA-runs with 12 and 20 ensemble members are quite close to the observed values. In addition, please note that using an increased number of ensemble members is a big challenge for large-scale high-resolution model because of needed computation memory and storage, and to a lesser degree also because of the computational burden. One year of model run with 20 ensemble members requires 680GB of computer storage per output variable (i.e. equivalent to 5TB of storage for 7 years of simulations per variable at daily time scale) and has resulted in the use of 76,800 CPU core-hours (compare to 46,000 core-hours with 12 ensemble members). These points are now discussed in the discussion section (Sec. 4) (Page 17, line 24 in the revised manuscript).

In the revised manuscript, we performed our data assimilation experiment with 20 ensemble members and revised all our results accordingly.

8. The uncertainty in the time series figures is for the 12 ensemble members?

[R2.10] Correct. We included this information in the figure captions (Fig. 6 and Fig 9 in the revised manuscript).

9. Numerous papers mentioned in the text are not included in the reference list!!!

[R2.11] Thanks you for pointing this out. We added missing references in the reference list.

Technical:

Spell-out ESA-CCI in the abstract.

[R2.12] This modification has been made (Page 1, line 18  in the revised manuscript).

 p.1,line 14: remove "and the . . . due to"

[R2.13] This modification has been made (Page 1, line 15  in the revised manuscript).

p.5, line 28: remove "In their study"

[R2.14] This modification has been made (Page 8, line 25  in the revised manuscript.

 p. 6: "this product" => which product you refer to here?

[R2.15] We referred to gridded runoff product E-RUN version 1.1 (Gudmundsson and Seneviratne, 2016). We modified text for clarity (Page 10, line 5  in the revised manuscript).

p. 6, line 19: missing space after parenthesis

[R2.16] This has been corrected in the revised manuscript.

 p.10, l. 13: "UK" => "BI"

[R2.17] This modification has been made (Page 19, line 31 in the revised manuscript).

p.12, line 9: runoff => "monthly runoff"

[R2.18] This modification has been made (Page 19, line 9 in the revised manuscript).

Figs. 5 and 9, caption: "a,c" => "a,b"

[R2.19] The figure captions has been modified for clarity (Fig 5 and Fig. 10 in the revised manuscript)

**Response to comments by Anonymous Referee #3**

**General comments:**
The objective of this work is to assimilate ESA-CCI satellite-derived soil moisture estimates in CLM over Europe, from 2000 to 2006. The content of the paper is disappointing. The data assimilation experiment boils down to a sensitivity study illustrating possible model biases. Independent validation is missing. No justification is given for the choice of the 2000-2006 time period. Most recent satellites are missing in this period. It is not clear whether or not interannual variability of the vegetation is accounted for in the model. The authors do not use the scores usually used in hydrology. The improvement of the assimilation on water discharge is not convincing at all. The paper is poorly organized (no Discussion section). This work cannot be published in the present form.
Recommendation: reject.

RESPONSE: We thank the reviewer for his/her efforts. Unfortunately, some of the statements above are not correct. The DA-experiments do not boil down to a sensitivity study, there is independent validation in our opinion and we use scores which are traditionally used in land surface DA studies (it is true that we do not use scores traditionally used in conceptual rainfall-runoff modeling studies). Other issues which are highlighted above have been corrected in the revised version of the manuscript.

The data assimilation experiment boils down to a sensitivity study illustrating possible model biases.
[R3.1] In this study we used data assimilation approach to merge coarse resolution satellite observations with a land surface model, to generate higher resolution, downscaled estimates of the surface soil moisture profile with complete spatio-temporal coverage. The added value of our study is to apply a data assimilation modeling framework over Europe to derive a longer-term and high spatial resolution land surface data product in order to increase monitoring accuracy for land surface soil moisture and water states and fluxes.

We now discussed the added value of data assimilation in the introduction section of the revised manuscript to make our objectives clearer to the readers (Page 3, line 31 and Page 4, line 21 in the revised manuscript).

Independent validation is missing.
[R3.2] We do not agree with this comment. Please note that we randomly selected 100 locations to assimilate the ESA CCI data into the CLM model, while the remaining data was used for independent validation. Additionally, another independent gridded observation-based runoff product was used to evaluate the performance of the soil moisture assimilation in the CLM model in improving indirectly additional hydrological variables such as runoff.

No justification is given for the choice of the 2000-2006 time period. Most recent satellites are missing in this period.

[R3.3] In this study, we used a high-resolution reanalysis system COSMO-REA6 from Hans-Ertel Centre for Weather Research (HErZ; Simmer et al., 2016) dataset which is a high spatial resolution in comparison to other commonly used forcing datasets such as the European gridded data set (E-OBS) (Haylock et al., 2008) and Interim ECMWF Reanalysis (ERA-Interim; Dee et al., 2011) at 25 and 80km resolution, respectively. COSMO-REA6 was produced through the assimilation of observational meteorological data using the existing nudging scheme in COSMO with boundary conditions from ERA-Interim data as discussed in Bollmeyer et al. (2015). We used data from 2000 – 2006, which were available at the beginning of this study. The COSMO-REA6 is only now publicly available for a longer time period of 1995 – 2015.

We agree that recent satellites such as the Soil Moisture and Ocean Salinity (SMOS) Mission, Soil Moisture Active Passive (SMAP) Mission could be another option, however, we selected the ESA CCI soil moisture product for assimilation because of its availability at longer time scales. This provides the opportunity to assess the potential impact of longer-term soil moisture observations on hydrologic simulations for climate change studies.

We clarified these points in the revised manuscript (Page 8, line 10 and Page 4, line 8 in the revised manuscript).

It is not clear whether or not interannual variability of the vegetation is accounted for in the model.

[R3.4] In the revised manuscript, we replaced the CLM default Plant Functional Types (PFTs) annual monthly leaf area index (LAI) cycles with prescribed LAI from MODIS to consider the inter-annual variability of the vegetation. Monthly LAI values for each PFT were computed based on the 1-km Global Land Surface Satellite (GLASS) Leaf Area Index (LAI) product (1981-2012). GLASS contains of 1 km x 1 km global maps of LAI provided every 8 days. The product is derived from time-series of MODIS (MOD09A1) and AVHRR reflectance data using general regression neural network method (Xiao et al., 2014). To derive a monthly climatology over the assimilation period (2000 – 2006), the 1 km 8-day improved GLASS LAI for each year was used to calculate a mean monthly LAI that was then aggregated to the model resolution. The monthly LAI for each PFT values were then determined by mapping the 3 km pixels to the 3 km aggregated PFT values within each grid cell. This approach provides the spatially distributed and temporally continuous LAI data within each PFT for the considered time period of 2000 - 2006. To account for annual variability in LAI, yearly model runs were performed where the LAI information was updated at the start of each year run. It should be noted that CLM3.5 only allows specifying monthly average LAI values for each FTP (Page 7, line 24 in the revised manuscript).

Comparison of new LAI parameters with the default CLM parameters showed that the new LAI parameters have significantly higher LAI for most PFTs as shown below in Fig. R8.

[Figure]

Figure R8: Comparison of new LAI parameters (dashed line) with the default CLM LAI values (solid lines) for different PFT over the time period of 2000 – 2006.

The authors do not use the scores usually used in hydrology.

[R3.5] The main objective of the study was to evaluate the performance of data assimilation of remotely sensed data into a land surface model in simulating surface soil moisture. We evaluated the performance in terms of RMSE, which is a common practice in most land surface data assimilation studies. To address the reviewer's concern, we now evaluated the performance of the model in simulating daily soil moisture and monthly runoff for 2000 – 2006 time period over PRUDENCE regions using suggested scores (mean absolute error (MAE), the root mean square error (RMSE), percentage bias (PBIAS) and correlation coefficient (R), Nash–Sutcliffe coefficient of efficiency (NSE) and Kling–Gupta efficiency (KGE) indices) as shown below in Table R3 and Table R4 (Table 2 and Table 4 in the revised manuscript).

Table R3: Evaluation performance criteria for comparing CLM-OL and CLM-DA with CCI-SM (spatially averaged SWC, surface layer, Europe and PRUDENCE regions).

| Soil Moisture (CLM-OL) | | | | | | | | | |
|---|---|---|---|---|---|---|---|---|---|
| | EU | BI | IP | FR | ME | SC | AL | MD | EA |
| PBIAS (%) | 54.1 | 16.4 | 50.7 | 24.7 | 25.6 | 23.0 | 16.4 | 33.4 | 34.8 |
| RMSE (mm$^3$/mm$^3$) | 0.10 | 0.05 | 0.10 | 0.07 | 0.07 | 0.06 | 0.05 | 0.07 | 0.08 |

| | | | | | | | | | |
|---|---|---|---|---|---|---|---|---|---|
| MAE (mm³/mm³) | 0.09 | 0.04 | 0.09 | 0.06 | 0.06 | 0.05 | 0.04 | 0.07 | 0.07 |
| R | 0.48 | 0.41 | 0.75 | 0.60 | 0.51 | -0.14 | 0.55 | 0.80 | 0.40 |

**Soil Moisture (CLM-DA)**

| | EU | BI | IP | FR | ME | SC | AL | MD | EA |
|---|---|---|---|---|---|---|---|---|---|
| PBIAS (%) | 36.4 | 3.1 | 33.2 | 10.1 | 11.0 | 9.0 | 3.0 | 18.1 | 19.0 |
| RMSE (mm³/mm³) | 0.07 | 0.03 | 0.07 | 0.04 | 0.04 | 0.05 | 0.03 | 0.04 | 0.06 |
| MAE (mm³/mm³) | 0.06 | 0.03 | 0.06 | 0.03 | 0.03 | 0.04 | 0.03 | 0.04 | 0.05 |
| R | 0.51 | 0.40 | 0.76 | 0.61 | 0.54 | -0.12 | 0.56 | 0.80 | 0.42 |

Table R4: Evaluation performance criteria for comparing CLM-OL and CLM-DA with E-RUN (spatially averaged runoff, EU and PRUDENCE regions).

**Total Runoff (CLM-OL)**

| | EU | BI | IP | FR | ME | SC | AL | MD | EA |
|---|---|---|---|---|---|---|---|---|---|
| PBIAS | 59.7 | -38.6 | 130.8 | 34.2 | 46.1 | -30.9 | -37.4 | 87.5 | 130.3 |
| RMSE (mm/day) | 0.5 | 1.3 | 0.8 | 0.7 | 0.6 | 0.9 | 0.9 | 0.7 | 0.7 |
| MAE (mm/day) | 0.4 | 0.9 | 0.7 | 0.6 | 0.5 | 0.8 | 0.8 | 0.6 | 0.7 |
| NSE | -4.0 | -0.2 | -1.8 | 0.3 | -0.3 | -0.4 | -0.4 | -1.6 | -10.2 |
| KGE | -0.1 | 0.1 | -0.4 | 0.3 | 0.4 | 0.1 | 0.3 | 0.0 | -0.6 |
| R | 0.9 | 0.5 | 0.6 | 0.7 | 0.6 | 0.3 | 0.7 | 0.5 | 0.6 |

**Total Runoff (CLM-DA)**

| | EU | BI | IP | FR | ME | SC | AL | MD | EA |
|---|---|---|---|---|---|---|---|---|---|
| PBIAS | 4.4 | -60.2 | 50.5 | -13.3 | -5.1 | -54.1 | -58.9 | 21.8 | 48.9 |
| RMSE (mm/day) | 0.3 | 1.6 | 0.6 | 0.7 | 0.5 | 1.2 | 1.3 | 0.5 | 0.4 |
| MAE (mm/day) | 0.2 | 1.2 | 0.5 | 0.5 | 0.4 | 1.0 | 1.2 | 0.5 | 0.3 |
| NSE | -0.7 | -0.9 | -0.6 | 0.1 | -0.1 | -1.2 | -1.8 | -0.6 | -2.6 |
| KGE | 0.2 | -0.1 | 0.0 | 0.2 | 0.3 | 0.1 | 0.1 | 0.1 | 0.0 |
| R | 0.6 | 0.4 | 0.2 | 0.4 | 0.3 | 0.5 | 0.6 | 0.1 | 0.4 |

While we see the value of including more scores for evaluating model performance, we noted that including other scores does not change our conclusions. With respect to RMSE and MEA, we clearly see a significant improvement in simulated soil moisture for CLM-DA in comparison to CLM-OL over all PRUDENCE analysis regions, except for region Scandinavia. For runoff, overall we see relatively less improvements in terms of NSE and KGE. In the land surface models such as CLM, the representation of runoff is often simplistic and conceptual and many previous studies have shown that performance of CLM model in simulating hydrological processes varies based on regions. This might be attributed to the fact that assumptions to estimate surface and subsurface runoff in the model might be valid in some regions but not in other regions (e.g. humid vs. dry regions). We also noted similar behavior of CLM in our study where the assimilation of soil moisture helps to improve runoff in some regions but degraded in other regions. Our results agree well with other data assimilation studies (e.g. Albergel et al., 2017; Parajka et al., 2006; Crow et al., 2006) which showed that assimilation is more effective in modifying the surface soil moisture but found little improvements to the discharge as a

result of the remotely sensed soil moisture assimilations. This also highlights the need to jointly use soil moisture and discharge observations to improve global and continental hydrological and/or rainfall-runoff models, but this is beyond the scope of the current manuscript. We now discussed the above points in the discussion section (Page 16, line 9 and Page 18, line 11 in the revised manuscript).

The paper is poorly organized (no Discussion section).

[R3.6] A discussion section (Sec. 4) is added in the revised manuscript.

Specific comments:
- P. 1, L. 21 (independent CCI-SSM observations): do you mean observations independent from CCI-SM? They should be!
[R3.7] In this study, we randomly selected 100 locations to assimilate the ESA CCI data into the CLM model, while the remaining data was used for independent validation. This approach not only allowed us to independently cross-validate the SM values over grid cells that were not used in the data assimilation, but also to produce updated soil moisture contents at other locations (at the European scale), based on spatial correlations, and to investigate whether soil moisture characterization between measurement locations could also be improved, and its impacts on runoff characterization (Page 11, line 24 in the revised manuscript).

- P. 3, L. 3: 10**0km?
[R3.8] $10^0$km

- P. 4, L. 3 (CLM): how is vegetation represented in this version of the model?
[R3.9] In CLM, each grid cell consists of five landunits (i.e. vegetation, wetland, lakes, glaciers and urban) covering a certain percentage of the total grid cell area. The vegetation landunit is further subdivided into Plant Functional Types (PFTs) defined by fractional areas with respect to the entire grid cell (Bonan et al., 2002). In the current study, the land cover information for each PFT is based on the Moderate Resolution Imaging Spectroradiometer (MODIS) MCD12Q1 (version 5) land cover product (Friedl et al., 2002), which contains a classification of the dominant land cover (Figure 1b). The dominant land cover information from MODIS was first aggregated to the model resolution, calculating the percentage of all 500 m pixels per 3 km grid cell. The aggregated land cover information was then transferred to the CLM-prescribed PFTs on the basis of WorldClim climate data (Hijmans et al., 2005).
Monthly leaf area index (LAI) values for each PFT were computed based on the 1 km Global Land Surface Satellite (GLASS) Leaf Area Index (LAI) product (1981-2012). GLASS contains of 1 km x 1 km global maps of LAI provided every 8 days. The product is derived from time-series of MODIS (MOD09A1) and AVHRR reflectance data using general regression neural network method (Xiao et al., 2014). To derive a monthly climatology over the assimilation period (2000 – 2006), the 1 km 8-day improved GLASS LAI for each year was used to calculate a mean monthly LAI that was then aggregated to the model resolution. The monthly LAI for each PFT values were then

determined by mapping the 3 km pixels to the 3-km aggregated PFT values within each grid cell. This approach provides the spatially distributed and temporally continuous LAI data within each PFT for the considered time period of 2000-2006. To account for annual variability in LAI, yearly model runs were performed where the LAI information was updated at the start of each year run. It should be noted that CLM3.5 only allow to specify monthly average LAI values for each FTP (Page 7, line 15 in the revised manuscript).

- P. 5, L. 5 (LAI): does this mean that LAI for a given month is the same from one year to another? Given the marked impact of LAI on evapotranspiration, this might introduce marked soil moisture biases.

[R3.10] As mentioned in the previous comment, in tmanuscript, monthly leaf area index (LAI) values for each PFT were computed based on the 1-km Global Land Surface Satellite (GLASS) Leaf Area Index (LAI) product (1981-2012). GLASS contains of 1 km x 1 km global maps of LAI provided every 8 days. The product is derived from time-series of MODIS (MOD09A1) and AVHRR reflectance data using general regression neural network method (Xiao et al., 2014). To derive a monthly climatology over the assimilation period (2000 – 2006), the 1 km 8 day improved GLASS LAI for each year was used to calculate a mean monthly LAI that was then aggregated to the model resolution. The monthly LAI for each PFT values were then determined by mapping the 3 km pixels to the 3-km aggregated PFT values within each grid cell. This approach provides the spatially distributed and temporally continuous LAI data within each PFT for the considered time period of 2000-2006. To account for annual variability in LAI, yearly model runs were performed where the LAI information was updated at the start of each year run. It should be noted that CLM3.5 only allow to specify monthly average LAI values for each FTP (Page 7, line 24 in the revised manuscript).

- P. 5, L. 20 (6 km): it is written in the Abstract and in Section 2.3.2 that the assimilation was made at a spatial resolution of 3 km. Why such a mismatch with the spatial resolution of atmospheric forcing?

[R3.11] In our study, we used the high-resolution reanalysis system COSMO-REA6 from Hans-Ertel Centre for Weather Research (HErZ; Simmer et al., 2016) dataset, which is available at 6km resolution. However, we performed model simulations over the EURO-CORDEX domain at 3km resolution, which is inscribed into the official EUR-11 grid at 0.11° spatial resolution. To match the model resolution, the 6km COSMO-REA6 was interpolated to 3 km resolution.

- P. 7, L. 4: E-OBS was not defined before.

[R3.12] Thanks you for pointing this out. We now defined the E-OBS (Page 8, line 11 in the revised manuscript).

- P. 11, L. 13 (this study demonstrates): I am not convinced, there is no demonstration.

[R3.13] The original statement has been revised for clarity.

- P. 11, L. 32 (soil texture): Absolute CCI-SM values depend on pedotransfer functions and texture of the NOAH model. They are not "observed" and they should not be taken for granted. This is not a good way of doing data assimilation.

[R3.14] We can't completely agree with this comment. According to Dorigo et al., 2017, "the European Space Agency CCI soil moisture product is the first multi-decadal, global satellite-observed soil moisture (SM) dataset as part of its Climate Change Initiative (CCI) program. This product, named ESA CCI SM, combines various single-sensor active and passive microwave soil moisture products into three harmonised products: a merged ACTIVE, a merged PASSIVE, and a COMBINED active + passive microwave product." It is true that the soil moisture values in the passive microwave product is not entirely model-independent, for the combined product (i.e. active + passive microwave product), the systematic differences between ACTIVE and PASSIVE are corrected by matching the CDF of each pixel against long-term LSM-based soil moisture, which is provided by GLDAS-Noah.
Several authors (e.g. Albergel et al., 2013, 2017; Dorigo et al., 2017; McNally et al., 2016; Wagner et al., 2012) have highlighted the quality and stability of the product. For example Albergel et al., 2017 assimilated the CCI soil moisture data into the land surface model over the Euro-Mediterranean region for the time period of 2000 – 2012.
In addition, in this study we use an observation uncertainty during assimilation, as stated already in the manuscript: "In this study, we assumed a spatially uniform observational error for CCI-SM (i.e. 0.02 mm3/mm3) in the CLM-PDAF setup." Therefore, the experimental design follows a usual and adequate way of doing data assimilation.

- P. 22, Table 1: what about other key hydrological scores such as KGE or NSC? Given what can be seen in Fig. 10, I doubt these scores are improved by the assimilation.

[R3.15] See our previous response above and Table R2 and Table R3 to the comment ([R3.5]) on hydrological scores.

[revised manuscript text omitted]

---

## Author Response (AR2)

Dear Editor,

We appreciate the valuable and constructive comments from the Editor and the Reviewers, which helped to improve the manuscript. Following the constructive criticism from the Reviewers in the 1st round, the manuscript underwent major revisions including additional data assimilation experiments with larger ensemble sizes and new input data sets. A major effort has been undertaken to address all comments and suggestions. Based on these revisions, Reviewer 1 and 2 accept the manuscript. Reviewer 3 raised additional concerns in the 2nd round of reviews. We believe that comments by Reviewer 3 were not precise, yet suggest that additional clarification of the text is required. To clarify these additional concerns raised by Reviewer 3, we revised the manuscript again and furthermore added additional information in the manuscript. Our detailed responses (in blue) to the reviewer's comments are provided below. Changes are also made in the manuscript accordingly (marked in red color).

We look forward to your decision.

On behalf of all co-authors,
Yours sincerely,
Bibi S Naz

**Response to Reviewers comments:**

**Anonymous Referee #1:**

All the comments issued have been addressed by the authors and clearly integrated within the text.

**Response:** We thank the reviewer for his/her valuable comments which helped us to improve the quality of our manuscript.

Comment: One minor correction at page 3 line 28: I would suggest the authors to add resolution also in km for easier comparison with stated resolutions in the rest of the paragraph.
Response: This has been corrected in the revised manuscript.

Comment: I would suggest the authors to check for typos in the text. As an example, double bracket at page 4 line 13 and missed full stop at page 17 line 13.
**Response:** We checked for all typos in the manuscript and have been corrected in the revised manuscript.

Comment: After these small corrections, my recommendation is to accept the manuscript after these technical corrections.
**Response:** We thank the reviewer for this positive assessment.

**Anonymous Referee #2**
accepted as is
**Response:** We thank the reviewer for his/her valuable comments which helped us to improve the quality of our manuscript.

**Anonymous Referee #3**

General comments:
Comment: Is assimilating ESA-CCI SM useful? This is a key question. Unfortunately, this work does not correctly address this issue.
**Response:** The main goal of this study is twofold: Firstly, it investigates the value of coarse-resolution remotely sensed soil moisture data in improving soil moisture and runoff modeling and to provide higher spatial resolution, downscaled estimates of the surface soil moisture profile and hydrological fluxes with complete spatio-temporal coverage over Europe. Secondly, it aims to explore the feasibility of long-term remotely sensed product such as ESA-CCI SM to derive a soil moisture reanalysis product over Europe at higher resolution. The soil moisture estimates in this study with improved spatial resolution from the assimilation offer a new product for monitoring soil water content with distinct benefits over the original CCI-SM data. This is has been clarified in the revised manuscript (Page 4, lines 21-25 and Page 20, lines 19-21).

Comment: The content of the revised paper remains disappointing. The authors have not properly addressed (understood?) my comments.
**Response:** In our opinion, main concerned raised by Reviewer 3 in the first round of review, were fully addressed which included (1) replacing the default LAI parameters with MODIS-based LAI values to account for interannual variability in vegetation, (2) assessing the results by including additional scores

such as NSE and KGE as suggested by the reviewer, (3) clarifying in the revised manuscript regarding the independent validation of the results (i.e. using runoff observation and cross-validation of soil moisture data which were not used in the data assimilation), and (4) separate results and discussion sections. However, the reviewer did not mentioned specifically which specific comments were not addressed (understood?) well and why.

Comment: Their responses to the other reviewers confirm that this work has major shortcomings.
**Response:** We do not agree with this comment as the other two reviewers recommend publication. This also justifies our concern about the reviewer's overall understanding of our previous responses.

Comment: The authors claim that "overcorrecting" assimilated SM limits the usefulness of the assimilation of SM data. I can agree with this statement. However, doing a bias correction is needed because absolute soil moisture values in m3m-3 are model-dependent. The authors of the ESA-CCI SM product have chosen to rescale their initially dimensionless product using soil parameters used in the NOAH model, not those parameters used in the CLM model. The bias observed by the authors of this paper is due to this coincidence and the impact of the assimilation observed by the authors is governed by this fake bias in SM values.
**Response:** In the data assimilation experiments, we are always dealing with systematic biases between model and data. For example, in meteorological models there are systematic biases for modelled precipitation (for example orographically induced precipitation). Nevertheless, there is no prior bias correction in data assimilation studies with atmospheric models. A priori bias correction is a specific approach taken in land surface data assimilation in case of large mismatches between modeled and measured values, for example, when the observations are located outside of the ensemble spread. We argue that for this dataset, we see systematic biases between model and data, yet these are small enough. In addition, data assimilation is able to remove biases besides the random component. In addition, data assimilation can resolve such biases besides the random component.
We have added some of this discussion in the revised manuscript (Page 10, lines 9-12).

Furthermore, we argue that soil-specific hydraulic parameters in most land surface models are derived using similar approaches and datasets representing reality. For example, the soil type–specific hydraulic parameters in Noah are obtained from the pedotransfer function (PTF) provided in Cosby et al. (1984), which was also adopted by other LSMs, such as the Community Land Model (CLM). The underlying soil classification in our setup is based on data from FAO soil map (Batjes, 1997), which was the basis for the GLDAS derived soil parameters used in GLDAS-Noah and employed to derive the ESA CCI-SM product (e.g. Dorigo et al., 2012 and https://ldas.gsfc.nasa.gov/gldas/GLDASsoils.php). The setup and the parameterizations of Noah and our CLM model should hence be fairly consistent. We have added some of this discussion in the revised manuscript (Page 9, lines 10-15 in the revises manuscript).

Comment: Moreover, the validation of the analysis relies only on river discharge observations. Why not using other independent datasets such as in situ SM observations or pre-existing evapotranspiration datasets in addition to river discharge?
Response: Unfortunately, the reviewer did not read our previous response letter carefully, in which we already addressed this issue. We noted already that only sparse observation networks are available, and in the period 2000-2006 even less in situ observations were available than nowadays. Furthermore, it can be difficult to compare the point-based observation with the average value of coarse resolution model grid cell particularly in terms of calculating bias.

Comment: The river discharge scores (NSE, KGE) NOW given by the authors are quite poor. Much better scores can be found in the literature (including in HESS publications) using the same kind of large scale hydrological models, not assimilating any satellite observation.
Response: We would like to ask the reviewer to provide more precise information. Which studies for the EUROCORDEX domain the reviewer is referring to, where only in situ observations are assimilated? What were the scores for NSE and KGE in those studies, for which variables etc? This comment remains unclear to us.
Considering traditional hydrologic modeling studies, we agree with the reviewer that highs score of NSE and KGE are required, if one aims to improve the forecast reliability and accuracy of the hydrological model, particularly for streamflow. However, this is mostly done by calibrating the uncertain parameters in the

hydrological models using discharge observations at guage locations for different catchments. Because of unavailability of spatially consistent *in-situ* observations at larger scale, calibrating hydrological model is not a trivial task at the continental scale.

In this study, we implemented a data assimilation framework with the aim to improve the prediction of not only soil moisture but also runoff at larger scale. Therefore, we investigate the potential of long-term remotely sensed products such as ESA-CCI SM for downscaling of soil moisture to high spatial resolution at the continental scale via data assimilation. In addition, the study also interrogates whether assimilation of satellite-derived surface soil moisture will improve the skill of the simulated discharge, in gauged and ungauged regions. Our results demonstrate that our data assimilation framework improves the simulation of runoff also for regions with limited availability of in-situ observations (e.g. in the Eastern Europe; Figure 10 in the revised manuscript). Therefore, assimilating satellite-derived information into land surface models may have an important added value for regions where in situ measurements are not available.

We have added some of this discussion in the revised manuscript (Page 4, lines 27-30).

Comment:  The assimilation does not improve the scores overall. Figure 9 shows that CLM-OL is often much better than CLM-DA.

**Response:** In our opinion the reviewer should consider both soil moisture and discharge and not use the provided evaluation selectively.

Comment: Finally, the used ESA-CCI time series has severe limitations. While quality tends to improve in 2007 with the ASCAT data, the authors use the 2000-2006 time period.

**Response:** We agree that with the merging of ASCAT data from 2007 onwards may improve the quality of the CCI-SM data, however we argue that data from sensors SSM/I, TMI, AMSR-E, Windsat and ERS 1/2 SCAT constitute an adequate observation basis in order to generate a long-term soil moisture data set. Especially by the integration of C-band data from AMSR-E, Windsat and ERS 1/2 SCAT in the 2000-2006 time frame the general quality of CCI SM data is good enough to use in the data assimilation studies. Several authors (e.g. Albergel et al., 2013, 2017; Dorigo et al., 2017; McNally et al., 2016; Wagner et al., 2012) have highlighted the quality and stability of the product. For example, Albergel et

al., 2017 assimilated the ESA-CCI soil moisture data into the land surface model over the Euro-Mediterranean region for the time period of 2000 – 2012. In addition, in this study we account for observation uncertainty during assimilation using a spatially uniform observational error for CCI-SM (i.e. 0.02 mm3/mm3) in the CLM-PDAF setup.

Additionally, as stated in the revised manuscript, that we selected ESA CCI-SM data because of its availability at longer time scales, which also provides a basis for evaluating the feasibility to derive a climatological land surface reanalysis for Europe. Such a reanalysis furthermore allows us to assess the potential impact of assimilating longer-term soil moisture observations on hydrologic simulations. It is true that a number of soil moisture retrievals from other missions such as the Soil Moisture and Ocean Salinity (SMOS, launched in 2009) and SMAP mission (Soil Moisture Active Passive, launched in 2015) have been used in assimilation studies (e.g. Lievens et al., 2015, 2016). However, these recently developed data products are only available for the recent past and cannot be used to in a land surface reanalysis for extended time periods in the near future.

In addition, we used a high-resolution atmospheric reanalysis dataset (COSMO-REA6 from Hans-Ertel Centre for Weather Research; Simmer et al., 2016; Bollmeyer et al., 2015) which was only available for 2000-2006 at the beginning of our study. The main advantage of this datasets over 
[revised manuscript text omitted]